# Controllable Molecule Generation via Sparse Representation Editing: An Interpretability-Driven Perspective

**Zhuoran Li** [1]   **Xu Sun** [1]   **Chang Wen Chen** [1]   **Wanyu Lin** [1]

## Abstract

Controllable molecule generation is crucial for diverse scientific applications, such as drug discovery and materials design. While large language models (LLMs) show great promise, their dense and entangled representations impede precise control over the generation of molecules with bespoke substructures or properties. To address this, we propose Sparse Representation Editing (SpaRE), an interpretability-driven framework for fine-grained and precise control in LLM-based molecule generation. The crux of SpaRE is to learn an overcomplete sparse feature space that disentangles LLM representations into a compact set of latent features corresponding to chemically meaningful concepts. Within this space, we can directly manipulate these concept-aligned latent features to achieve (1) local control, by generating target atoms and functional groups at specified positions; and (2) global control, by customizing the overall structural and physicochemical properties within defined ranges. In this way, our framework advances interpretability from post-hoc analysis to actionable generative control. Experiments show that SpaRE can generate chemically desirable molecules under complex constraints in real-world scenarios, while offering mechanistic insights for quantitative structure–property analysis. The code and demo are available at `https://github.com/WanyuGroup/ICML2026_SpaRE`.

## 1. Introduction

Designing molecules with specified substructures and properties is central to scientific discovery, spanning areas such as drug development, materials engineering, and chemical synthesis (Fan et al., 2022; Yi et al., 2024; Zhao et al., 2025). Recent breakthroughs in LLMs have revolutionized *de novo* molecule generation, leading to improved performance and broader applicability (Feng et al., 2024; Pei et al., 2025; Fu et al., 2025). By learning expressive, high-dimensional representations, LLMs can effectively capture meaningful patterns embedded within molecular sequences (Zhang et al., 2025). However, these rich representations are a double-edged sword: their complexity obscures the semantic disentanglement of underlying features, making it significantly challenging to identify and manipulate molecular characteristics in response to human instructions.

Motivated by this limitation, recent works have explored LLM-based controllable molecule generation to better align molecular outputs with human-specified objectives. Representative techniques include retrieval-based search (Wang et al., 2023), geometry-aware property tokenization (Li et al., 2025), cross-modal hierarchical alignment (Zhang et al., 2025), and trigger-query-based multimodal control (Liu et al., 2025a). Despite offering some controllability, these solutions are coarse-grained, restricting control only to the presence of specific properties or fragments, without precisely adjusting property levels or editing sites. However, real-world molecular design often necessitates site-specific edits on predefined scaffolds to optimize biological activity or desired properties under multiple constraints, without incurring disruptive structural changes (Kennedy et al., 2021; Jurczyk et al., 2022). Therefore, granular control is indispensable for practical molecular discovery.

To this end, we develop a lightweight approach, dubbed SpaRE, to realize fine-grained control over molecule generation at inference time, without altering model parameters. Technically, SpaRE operates directly in the model's representation space, where it leverages the sparse autoencoder (SAE) (Olshausen & Field, 1997; Huben et al., 2024; Bricken et al., 2023; Templeton et al., 2024; Gao et al., 2025) to learn a suite of sparsely activated and semantically interpretable features from the model's activations. These features align with key concepts that govern molecular structure and property formation in chemical space. Building upon this, we develop two complementary strategies to modulate the strength of these concepts, selectively amplifying or suppressing their influence on generated molecules. Specifi-

---

[1]Department of Computing, The Hong Kong Polytechnic University, Hong Kong SAR, China. Correspondence to: Wanyu Lin <wan-yu.lin@polyu.edu.hk>.

*Proceedings of the 43rd International Conference on Machine Learning*, Seoul, South Korea. PMLR 306, 2026. Copyright 2026 by the author(s).

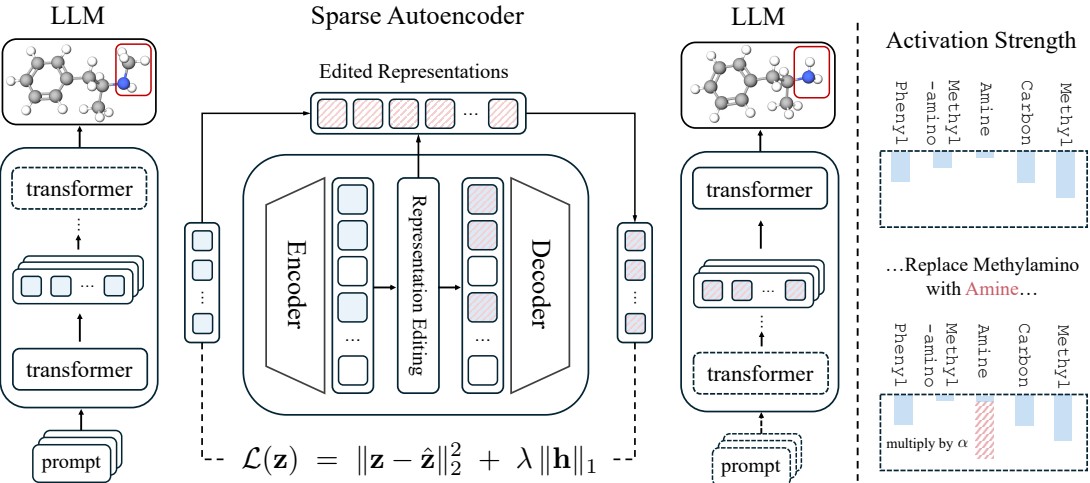

*Figure 1.* SpaRE learns concept-specific activation patterns that connect dense LLM representations to chemically meaningful concepts in latent space. By modulating concept activation strengths, SpaRE steers the corresponding LLM representations toward molecules with desired substructures and properties (dashed boxes denote skipped computations).

cally, *local control* achieves targeted generation of substructures by retrieving activations of specific atoms or functional groups with a forward hook. In contrast, *global control* adjusts overall molecular properties by deriving property-associated activations from contrastive guiding samples (*i.e.,* positive vs. negative samples). During inference, the edited activations are inserted back into their original layer to steer the generated molecules toward user-specified targets.

To verify its effectiveness, we evaluate SpaRE on a variety of real-world tasks, including high-throughput screening for drug design and optimization (*e.g.,* on the ChEBI-20 dataset (Edwards et al., 2021)). In summary, SpaRE supports: (1) molecule generation with specific atoms or functional groups at designated sites; (2) structure- and property-controlled generation, allowing precise control over characteristics such as aromaticity, synthetic accessibility, and ring systems within specified ranges; (3) planning and optimization of intricate molecular editing routes; and (4) molecular optimization under complex constraints. Empirical results indicate that SpaRE efficiently generates valid, synthesizable molecules with a high control success rate. More crucially, it yields attributable explanations of how specific edits impact molecular properties. These insights facilitate rigorous quantitative analysis and substantially expedite scientific discovery. Our main contributions include:

- We use SAE to disentangle LLM representations into chemically meaningful concepts, offering an interpretable lens for controllable molecule generation.

- We introduce a unified representation editing framework that enables customizable molecule generation with fine-grained control over both local and global molecular characteristics.

- We evaluate SpaRE on multiple molecular design and optimization tasks, where it showcases excellent generation quality, precise control, and high efficiency.

## 2. Related Work

**Molecule Generation with LLMs.** The rapid innovation of LLMs has propelled advances in sequential data generation (Radford et al., 2018; Achiam et al., 2023; Touvron et al., 2023) and expanded their use to scientific tasks such as molecular discovery (Guo et al., 2023; Bran & Schwaller, 2024). By representing molecules as strings (Weininger, 1988; Krenn et al., 2020), LLMs can process them as sequences of chemical tokens. Recent studies show that LLMs excel at molecule generation by learning chemical semantics from large corpora, supporting flexible generation beyond conventional generative models (Bagal et al., 2021; Edwards et al., 2022; Liu et al., 2023b; Li et al., 2024; Feng et al., 2024). However, fine-grained control remains challenging because the entangled, high-dimensional LLM representations hinder precise adherence to human guidance. To address this, we disentangle and modulate concept representations within the latent space of LLMs to achieve customized generation of molecular substructures and properties.

**Controllable Molecule Generation.** Controllable molecule generation seeks to align latent representations of generated molecules with human intentions (Kang & Cho, 2018; Reidenbach et al., 2023). Building on this principle, a range of strategies have been developed to improve the controllability (Rothchild et al., 2021; Li et al., 2023; Wang et al., 2023; Roy et al., 2023; Fang et al., 2024; Li et al., 2025; Zhang et al., 2025; Liu et al., 2025a). Yet, these approaches offer only coarse-grained control. Moreover, they typically rely on auxiliary conditioning modules or demand considerable

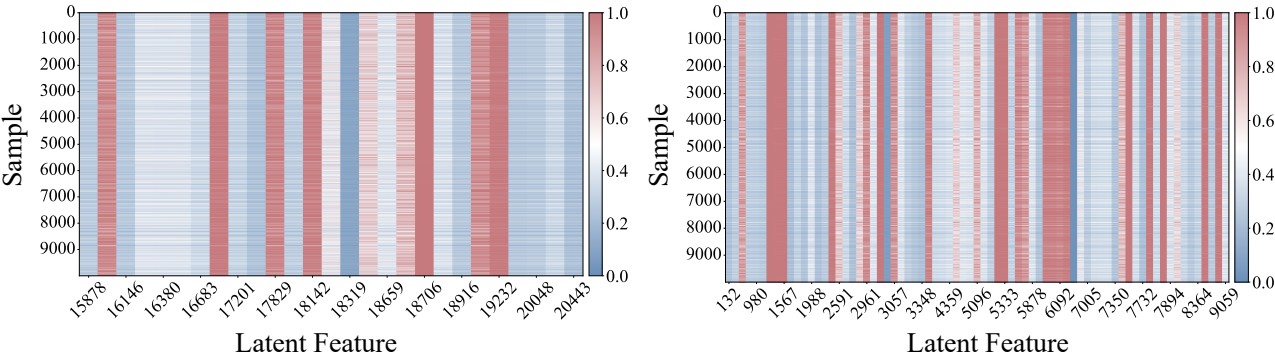

*Figure 2.* Activation patterns for concepts are derived using two schemes: (**Left**) local substructure control of carbon atom and (**Right**) global property control of solubility. For each concept, we select only latent features that are consistently activated (*i.e.,* activation value > 0.5) in every sample of that concept and concatenate them to form the concept representation used for its control.

training or fine-tuning. In this work, we present an efficient representation editing approach for pretrained LLMs, providing precise control over molecular substructures and properties. SpaRE delivers enhanced controllability and generation quality with reduced computational overhead.

## 3. Method

As shown in Figure 1, SpaRE establishes a concept-aligned subspace in which targeted activation edits yield desired molecules. Particularly, we extract activations from each LLM layer to train an SAE that projects them into a latent space where few active dimensions correspond to interpretable concepts (detailed in Section 3.1). We then modulate the strength of these concepts, either amplifying or suppressing them, resulting in new activation vectors to control them. The modified activations are re-injected into their source layers, enabling controllable generation at both local and global levels during inference (detailed in Section 3.2).

### 3.1. Disentangling Latent Representations into Concepts

To achieve controllable molecule generation, we first transform the dense hidden states of LLMs into a structured, sparse latent space, where each basis dimension corresponds to a distinct concept (Elhage et al., 2022). We achieve this using an SAE, which disentangles neuron activations into multiple sparsely activated latent features via sparsity constraints, thereby reducing feature redundancy (Kreutz-Delgado et al., 2003; Lee et al., 2006). Formally, let $\mathbf{z} \in \mathbb{R}^d$ be the activation vector extracted from an LLM layer for a given input prompt, where $d$ is the activation dimension. We define a latent space of dimension $m \gg d$ to construct a large, overcomplete basis, which offers a rich vocabulary for decomposing complex activations into highly disentangled and semantically interpretable features. The SAE comprises an encoder $\mathbf{h}$ and a decoder $\hat{\mathbf{z}}$, denoted as:

$$\mathbf{h} = \mathrm{ReLU}\big(\mathbf{W}_{\mathrm{enc}}\mathbf{z} + \mathbf{b}_{\mathrm{enc}}\big), \qquad \hat{\mathbf{z}} = \mathbf{W}_{\mathrm{dec}}\mathbf{h} + \mathbf{b}_{\mathrm{dec}},$$

where $\mathbf{W}_{\mathrm{enc}} \in \mathbb{R}^{m \times d}$ and $\mathbf{W}_{\mathrm{dec}} \in \mathbb{R}^{d \times m}$ are the learnable encoder and decoder weights, and $\mathbf{b}_{\mathrm{enc}} \in \mathbb{R}^m$ and $\mathbf{b}_{\mathrm{dec}} \in \mathbb{R}^d$ are the corresponding bias terms. Empirically, $m$ is set to $d$ times a positive expansion factor to construct the overcomplete basis. The objective for training the SAE consists of two components, reconstruction error and sparsity regularization, defined as follows:

$$\mathcal{L}(\mathbf{z}) \;=\; \|\mathbf{z} - \hat{\mathbf{z}}\|_2^2 \;+\; \lambda \|\mathbf{h}\|_1, \tag{1}$$

where the reconstruction error $\|\mathbf{z} - \hat{\mathbf{z}}\|_2^2$ preserves essential chemical information, while the $L_1$ penalty encourages sparsity in $\mathbf{h}$. The hyperparameter $\lambda$ balances reconstruction fidelity and sparsity. Overcompleteness allows each activation to be represented as a sparse combination of latent features that are aligned with specific chemical concepts. Interestingly, we observe a representational hierarchy: shallow layers capture global molecular properties, while deeper layers encode local substructural fragments, enabling both global and local control in molecule generation. We detail SAE training hyperparameters in Appendix E.

### 3.2. Modifying the Strength of Concept Representations

SpaRE provides precise, interpretable control by modulating the strengths of disentangled concepts learned in Section 3.1, without extra modules or fine-tuning. Accordingly, we propose two schemes within a unified framework to enable both local, site-specific modifications and global property adjustments. Specifically, let $\mathbf{z}_t \in \mathbb{R}^d$ denote the intermediate activation at token position $t$ (the position to be controlled) at a chosen layer. The SAE is formulated as follows:

$$\mathbf{h}_t = \mathrm{ReLU}\big(\mathbf{W}_{\mathrm{enc}}\mathbf{z}_t + \mathbf{b}_{\mathrm{enc}}\big), \qquad \hat{\mathbf{z}}_t = \mathbf{W}_{\mathrm{dec}}\mathbf{h}_t + \mathbf{b}_{\mathrm{dec}}.$$

In general, we adjust concept strength at token position $t$ by adding an inference-time edit $\Delta\mathbf{z}_t$ to the original activation, producing a steering activation $\mathbf{z}_t^\star = \mathbf{z}_t + \Delta\mathbf{z}_t$. This drives the model from its original hidden state $\mathbf{z}_t$ toward a target state $\mathbf{z}_t^\star$, which is used for subsequent inference to generate the desired output. We develop two schemes for controlling

concepts at both local and global levels as follows.

**Scheme 1: Local Substructure Control via Hook Retrieval.** For the target atom/group at generative step $t = t_0$ that we aim to control, we attach a forward hook to read out its decoded activation vector. The hook is a lightweight callback executed during the forward pass that (i) extracts the activation $\mathbf{z}_{t_0} \in \mathbb{R}^d$ used to produce the token logits and (ii) can replace it before the model proceeds. Concretely, by intercepting inference at the step when the token feeds into the tokenizer head, the hook fetches token-level activations for the atom or functional group to edit. Let the atom or functional group at position $t = t_0$ be the control target:

$$\mathbf{z}_t^* = \mathbf{z}_t + \Delta \mathbf{z}_t, \text{ i.e., } \Delta \mathbf{z}_t = \alpha \cdot \texttt{forward\_hook}(\mathbf{z}_{t_0}).$$

Here, the local concept representation can be obtained by $\texttt{forward\_hook}(\mathbf{z}_{t_0})$. At the target generative step $t = t_0$, forward hook extracts the token-specific activation, which is then adjusted toward the desired atom or functional group. The adjustment is scaled by a factor $\alpha \in [0, 1]$, where suppression corresponds to $\alpha \approx 0$ and amplification to $\alpha \approx 1$ ($|\alpha - 1| \leq \varepsilon$ for small $\varepsilon > 0$). For all other steps (when $t \neq t_0$), activations are left unchanged, ensuring that only the local substructure $t_0$ is modified while preserving the rest of the molecular structure. In practice, once we compute concept representations for each atom and functional group in the dataset, we can reuse them for local concept control during inference indefinitely. The justification of this scheme is shown in Appendix A.

**Scheme 2: Global Property Control via Contrastive Guidance.** For global structural or physicochemical targets, no single token independently determines the overall molecular property. Instead, the relevant semantics are distributed collectively across multiple tokens. To address this, we build positive and negative exemplar sets (molecules with and without the target property) and contrast their average concept representations. The resulting difference vector captures the property-associated activations in representation space, allowing targeted guidance during generation. Given contrastive samples $\mathcal{X}^+$ and $\mathcal{X}^-$, we aggregate and average the activations $\mathbf{h}_t(x)$ across tokens and samples. Then we have $\bar{\mathbf{h}}^+ = \frac{1}{|\mathcal{X}^+|} \sum_{x \in \mathcal{X}^+} \frac{1}{T_x} \sum_{t=1}^{T_x} \mathbf{h}_t(x)$ and $\bar{\mathbf{h}}^- = \frac{1}{|\mathcal{X}^-|} \sum_{x \in \mathcal{X}^-} \frac{1}{T_x} \sum_{t=1}^{T_x} \mathbf{h}_t(x)$, where $x$ is the LLM input and $T_x$ is its token length. We define the global concept representation as $(\bar{\mathbf{h}}^+ - \bar{\mathbf{h}}^-)$ and modulate its strength by $\alpha$. Global control is applied at every step $t$ as:

$$\mathbf{z}_t^* = \mathbf{z}_t + \Delta \mathbf{z}_t, \text{ i.e., } \Delta \mathbf{z}_t = \alpha \cdot \mathbf{W}_{\text{dec}}(\bar{\mathbf{h}}^+ - \bar{\mathbf{h}}^-).$$

We construct contrastive exemplar sets $(\mathcal{X}^+, \mathcal{X}^-)$ and estimate a global direction in representation space that encodes the target property. During inference, we apply edits at every generative step so their effects accumulate, gradually steering the generation toward specified global properties.

The justification of this scheme is shown in Appendix B.

**Case Study.** Figure 2 presents activations for the concepts *carbon atom* (local) and *solubility* (global), derived using two different schemes. For the carbon atom (*i.e.,* to generate a carbon atom at a specific position), we extract the concept representation by hooking the activated latent features when the LLM outputs the carbon atom token. We retain only those features with activation values above 0.5 in all samples, and concatenate them to form the concept representation. We apply the edit only at the step immediately before the carbon atom is generated. For solubility (*i.e.,* to generate molecules with high solubility), we construct contrastive samples (lipophilic and hydrophilic molecules) and compare their activation patterns. We retain latent features with activations above 0.5 in all positives and below 0.5 in all negatives, then average their activations over the entire dataset to form the concept representation. In contrast to the carbon atom, we apply edits at every step during the molecule generation. In summary, the local scheme retrieves activations via a forward hook at the selected position as the LLM generates the target atom or functional group, and uses these activations as the concept representation. The global scheme contrasts average activations from positive and negative samples across all token positions, isolating a shared concept representation for the global property. The implementation details are provided in Appendix C.

## 4. Experiments

We evaluate SpaRE on three benchmarks: ChEBI-20 (Edwards et al., 2021), GEOM-DRUGS, and GEOM-QM9 (Axelrod & Gomez-Bombarelli, 2022), against 13 baselines, including MolXPT (Liu et al., 2023b), BioT5 (Pei et al., 2023), BioT5+ (Pei et al., 2024), LDMol (Chang & Ye, 2025), NExT-Mol (Liu et al., 2025b), TGM-DLM (Gong et al., 2024), Atomas (Zhang et al., 2025), CDGS (Huang et al., 2023a), JODO (Huang et al., 2023b), RetMol (Wang et al., 2023), MARS (Xie et al., 2021), MolEvol (Chen et al., 2021), and Llamole (Liu et al., 2025a). We use comprehensive metrics covering generation quality, synthetic feasibility, controllability, efficiency, and molecular properties. Experimental details are in Appendix D.

### 4.1. Direct Control Over Local and Global Concepts

**Local Control.** Table 1 summarizes local control results. To ensure fair comparison, we include baselines with site-specific constrained generation capabilities. We evaluate local control by iterating over all molecular positions and, at each site, generating a target atom or functional group from a library of about 150 candidates to replace the original one. SpaRE performs structure-aware control: when modifying a local site, the model adaptively edits its neighbors to ensure

*Table 1.* Site-specific molecule generation on the ChEBI-20 dataset (Edwards et al., 2021). Quality and controllability are percentages, while synthesizability and efficiency are reported as numerical values. **Best** and second-best results are marked in bold and underlined.

| | QUALITY (↑) | | | | | CONTROL(↑) | SYNTHESIS(↓) | EFFICIENCY (↓) |
|---|---|---|---|---|---|---|---|---|
| MODEL | VALID | UNIQUENESS | NOVELTY | ATOM STA | COMPLETENESS | SUCCESS RATE | SA SCORE | TIME |
| *GNN-Based* | | | | | | | | |
| MARS | 87.24 | 82.53 | 84.62 | 89.82 | 98.24 | 50.37 | 4.12 | 384.24 |
| MolEvol | 88.73 | 80.96 | 84.82 | 94.17 | 98.92 | 37.25 | 4.26 | 231.27 |
| *Diffusion-Based* | | | | | | | | |
| LDMol | 90.23 | 80.42 | 80.32 | 89.57 | 98.48 | 26.36 | 3.96 | 290.52 |
| TGM-DLM | 87.33 | 74.67 | 80.17 | 92.68 | 98.71 | 30.77 | 4.69 | 349.68 |
| CDGS | 89.51 | 80.32 | 84.98 | 93.62 | 98.16 | 35.89 | 3.93 | 332.35 |
| JODO | 88.42 | 75.36 | 87.13 | 92.59 | 97.46 | 31.71 | 4.52 | 325.57 |
| *Autoregressive-Based* | | | | | | | | |
| MolXPT | 87.72 | 80.25 | 88.56 | 95.46 | **99.71** | 37.15 | 4.85 | 25.23 |
| BioT5 | **100.00** | **83.48** | 86.12 | 92.32 | 99.54 | 28.79 | 5.72 | 29.06 |
| BioT5+ | **100.00** | 72.68 | 89.63 | 93.71 | 98.97 | 32.91 | 4.47 | 26.47 |
| NExT-Mol | 85.14 | 71.89 | 86.54 | 95.86 | 99.42 | 35.97 | 4.73 | 298.34 |
| Atomas | 86.65 | 73.61 | 90.91 | 93.21 | 98.81 | 30.48 | 4.72 | 317.98 |
| RetMol | 94.84 | 82.65 | 81.24 | 92.86 | 94.82 | 62.58 | 3.85 | 308.62 |
| Llamole | 92.37 | 80.24 | 81.27 | 89.97 | 97.29 | 46.26 | 4.54 | 192.35 |
| SpaRE (**Ours**) | **100.00** | 81.60 | **92.10** | **97.24** | 99.66 | **98.92** | **3.78** | **12.19** |

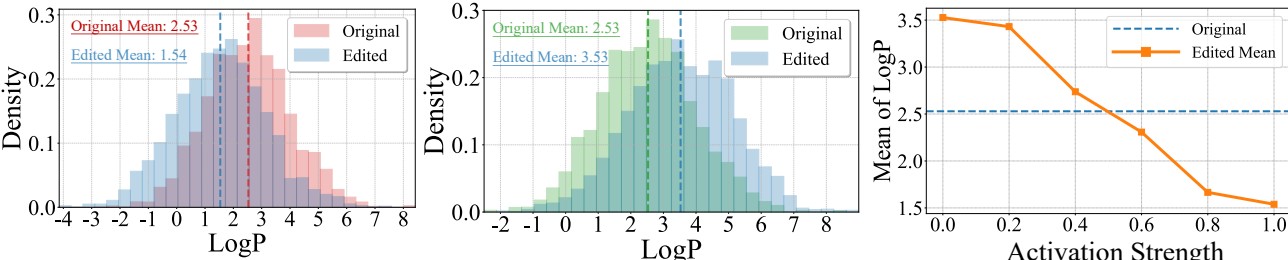

*Figure 3.* Distribution of molecules generated under global control of solubility: (**Left**) amplification, (**Middle**) suppression, and (**Right**) controllable tuning by varying activation strength.

validity, generating globally coherent molecules. For comparison, we also include a naive string substitution approach in Table 9. Despite performing well on some metrics, it ignores molecular structure and produces syntactically valid but chemically implausible molecules, which highlights the need for chemically meaningful generation. In summary, SpaRE offers fine-grained control with a high success rate, chemical consistency, and synthesizability, all at much lower cost. See Appendix F for more details.

**Global Control.** Figure 3 demonstrates global control over the concept of solubility. By constructing contrastive samples based on solubility, we can precisely manipulate this property. SpaRE shifts the distribution of the solubility metric, LogP, in both directions: the left panel shows increased solubility (lower LogP), the middle panel shows decreased solubility, and the right panel illustrates smooth, tunable solubility changes as activation strength varies. These results confirm the feasibility of globally controlling complex molecular characteristics. Additional results for aromaticity, hydrogen bonding, ring systems, and ortho-disubstituted positions are provided in Appendix G.

### 4.2. Editing Routes Planning and Optimization

**Bioisosteric Editing.** Bioisosteric editing optimizes properties by substituting atoms or groups with structurally and electronically similar surrogates while preserving the core scaffold. Unlike prior methods that disrupt structure or raise synthetic complexity, SpaRE enables targeted edits with high scaffold and pharmacophore similarity at each step. In this study, we utilize an expert-curated library of bioisosteric pairs and design four multi-step editing routes for a lipophilicity-oriented optimization task relevant to membrane permeability. As shown in the upper panel of Figure 4, each route consistently increases LogP and reduces TPSA (more lipophilic), enhancing lipophilicity while maintaining structural and biological relevance. Notably, two routes retain favorable drug-like properties throughout (Figure 22, Figure 23). Scaffold and pharmacophore similarity remain above 60% at each step, ensuring structural consistency. Empirically, certain edits, such as replacing sulfonamide with trifluoromethylsulfone (see Step 3, Route 4 in Figure 24), are especially effective, offering practical guidance for molecular design. These results show that SpaRE

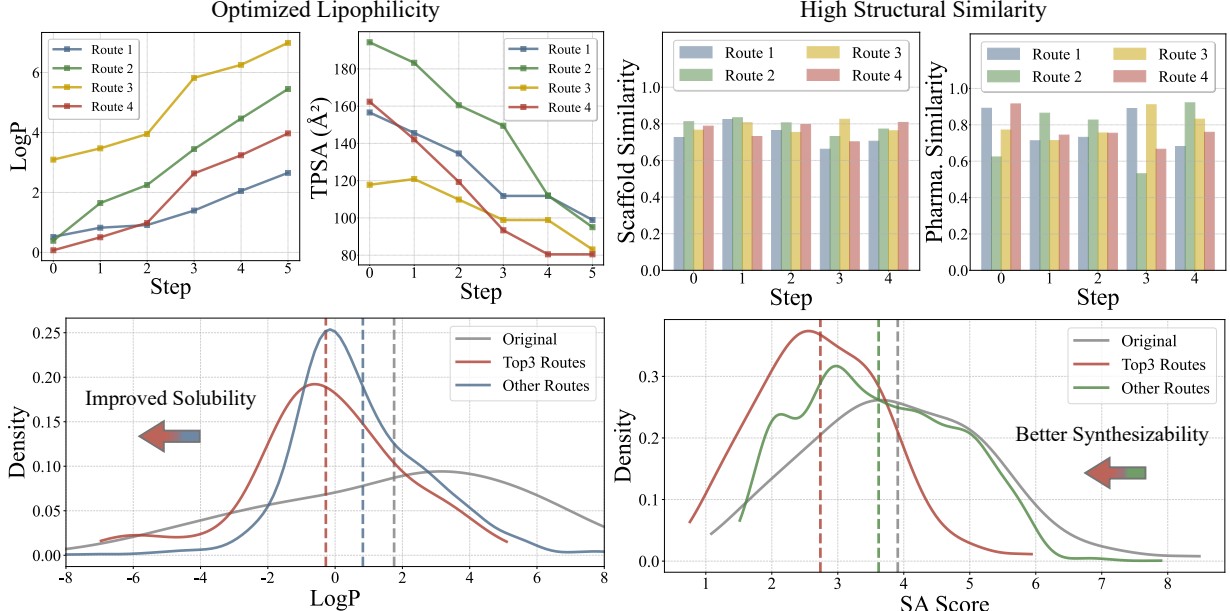

*Figure 4.* (**Top**) The edited molecules optimize lipophilicity while maintaining high scaffold and pharmacophore similarity. (**Bottom**) After optimizing the editing routes, the generated molecules exhibit improved water solubility and synthetic accessibility.

*Table 2.* Generation quality (%) for molecules optimized for QED, SA score, and predicted binding affinities to GSK3$\beta$ and JNK3, as estimated by the pretrained models (Jin et al., 2020).

| METHOD | SUCCESS | VALID | NOVELTY | DIVERSITY |
|---|---|---|---|---|
| MolEvol | 88.9 | 95.1 | 81.4 | 67.9 |
| RetMol | 93.8 | 98.7 | 87.2 | 71.6 |
| TGM-DLM | 94.5 | 99.7 | 78.1 | 67.4 |
| Atomas | 95.2 | 97.6 | 85.3 | 72.2 |
| Llamole | 90.3 | 97.3 | 90.6 | 68.2 |
| SpaRE (**Ours**) | **98.1** | **100.0** | **96.2** | **74.9** |

enables fine-grained, attributable property tuning under similarity constraints. It thus supports molecular optimization for lead development, scaffold hopping, and toxicity reduction. Further results are provided in Appendix H.

**Complex Editing Routes Planning.** Real-world molecular editing is hindered by two main challenges: limited controllability of precise edits and poor tractability under synthesizability constraints (Nicolaou et al., 2012). Building on recent advances in arene/heteroarene skeletal editing and skeletal-versus-peripheral editing of indoles with fluoroalkyl N-triftosylhydrazones (Liu et al., 2024; Cheng et al., 2024), we address these issues by combining SpaRE's controllable generation with a Monte Carlo Tree Search (MCTS)–based search strategy. In particular, we leverage MCTS to explore candidate atoms or functional groups, evaluating each edit for synthetic feasibility and chemical validity. This enables efficient identification of low-barrier pathways, typically within five steps, that convert accessible starting molecules into complex target compounds otherwise challenging to

design manually. Our results highlight SpaRE's potential to accelerate real-world laboratory synthesis pipelines. Further results are provided in Appendix I.

**Strategic Optimization of Editing Routes.** Optimizing molecular properties while maintaining chemical validity and synthetic feasibility remains a challenge in both laboratory and computational settings. Existing methods often either overly restrict exploration or expand the search space without ensuring feasibility, disrupting baseline properties or failing to fine-tune them within desired ranges. To tackle this problem, SpaRE performs site-specific molecular edits under chemical constraints. Instead of predefining edit sites, we first prompt an LLM to suggest functional groups relevant to the target property (*e.g.,* ten groups for water solubility improvement, such as $-OH$). All feasible editing sites on the scaffold are then enumerated, and each group is systematically introduced at up to two sites, subject to two synthetic accessibility constraints. Each edit is assessed for its effect on the target property, producing a ranked list of viable site–group combinations ($\sim 50$ routes in total). As shown in the bottom panel of Figure 4, SpaRE optimizes routes that improve water solubility while enhancing synthetic feasibility. This approach streamlines chemical space exploration, reduces experimental workload, and accelerates compound screening and discovery. It also elucidates how site-specific modifications affect properties (*e.g.,* introducing a carboxylate anion or nitro group at certain sites effectively enhances water solubility; see Figure 28), informing actionable insights for molecular design. Further results are presented in Appendix J.

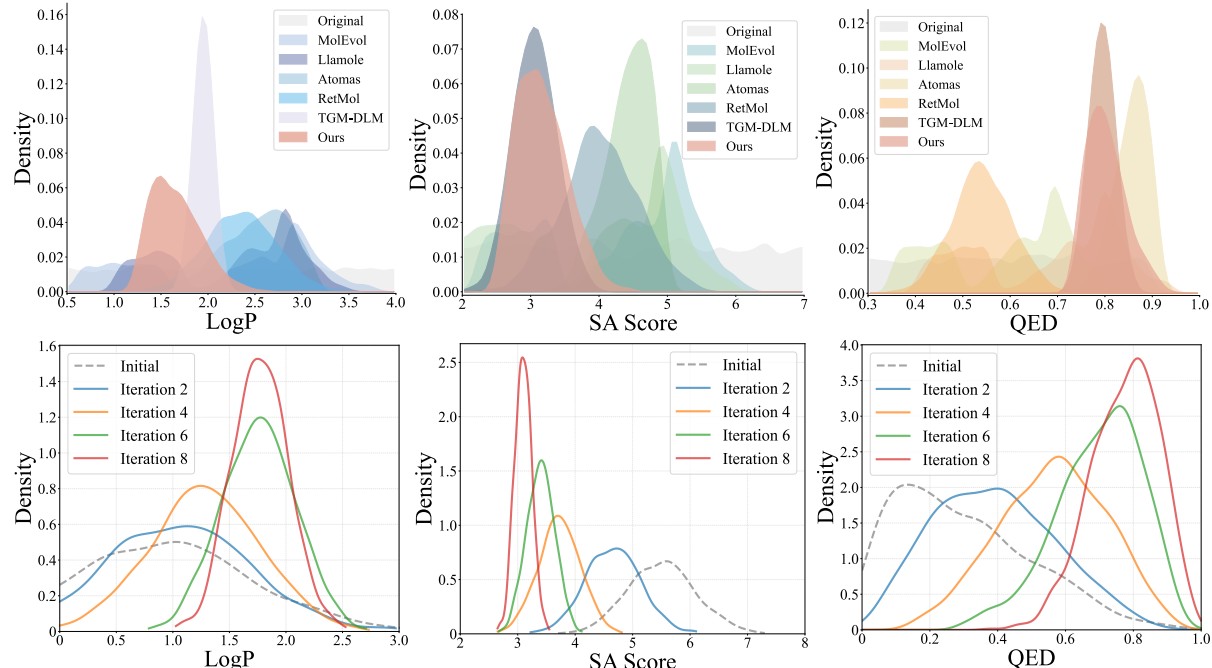

*Figure 5.* (**Top**) Distribution of generated molecules for oral drugs, showing that three key constraints are precisely optimized within their defined ranges, with SpaRE outperforming all baselines. (**Bottom**) The optimization objectives are achieved in only eight iterations.

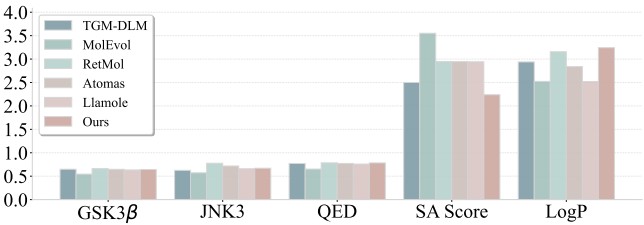

*Figure 6.* The property constraints employed for GSK3$\beta$ and JNK3 inhibitor discovery are precisely optimized within target ranges. SpaRE achieves stronger overall performance than the baselines.

*Figure 7.* Trajectories of iterative property optimization.

### 4.3. Constrained Multi-Objective Molecule Generation

**Multi-Target Drug Discovery.** GSK3$\beta$ (Glycogen synthase kinase-3 beta) and JNK3 (c-Jun N-terminal kinase 3) are serine/threonine kinases involved in metabolic signaling and CNS stress responses, both linked to neurodegenerative diseases such as Alzheimer's (Cohen & Goedert, 2004; Hooper et al., 2008). This underscores the need for dual inhibitors with favorable molecular and pharmacological properties. However, designing such compounds requires simultaneous optimization across multiple targets within a vast chemical space (Navia & Chaturvedi, 1996). To address this, we present a controllable generation framework integrated with MCTS for accurate property control throughout the molecule generation process. Following (Jin et al., 2020), we design drugs under four constraints: GSK3$\beta$ activity $\geq$ 0.5, JNK3 activity $\geq$ 0.5, drug-likeness (QED) > 0.6, and SA score < 4. Algorithmically, each molecule is treated as a state and each edit as an action, with rewards re-

flecting progress toward meeting all constraints. The search selectively expands promising nodes, estimates expected returns, and terminates when all constraints are satisfied or the step budget is reached. As shown in Table 2 and Figure 7, our approach reliably generates molecules that meet all property thresholds via targeted edits, outperforming state-of-the-art baselines in both generation quality and success rate. Furthermore, all properties are robustly optimized to levels well beyond the constraint thresholds (Figure 6), demonstrating the effectiveness of our granular controllability strategy for drug discovery. Additional analyses of Perindopril and Aripiprazole are provided in Appendix K.

**Complex Constraint Optimization for Oral Drug Design.** While oral administration is patient-friendly, optimization of oral drugs is complicated by stringent and multifaceted constraints (Khanna, 2012). Achieving high bioavailability requires balancing diverse molecular properties (Zhang & Wilkinson, 2007) and simultaneously satisfying multiple

*Table 3.* LLM-derived interpretations of latent features, summarizing representational commonalities and reporting each feature's category, concept, and detailed interpretation. More examples are presented in Table 17.

| Feature ID | Category | Concept | Detailed Interpretation |
|---|---|---|---|
| 2107, 5036, 10626, 14729 | Functional Group | N-oxide | Recognizes the N-oxide feature for tuning the electronics and solubility of heterocycles. |
| 10113, 14412 | Structural Class | Saturated carbocyclic systems | Generates saturated carbocyclic systems (*e.g.,* cyclohexane), focusing on sp$^3$-rich structures. |
| 2839, 2872, 13586 | Stereochemistry | Helical chirality | Models helical chirality, a stereochemical feature arising from a molecule's screw-shaped structure. |
| 14381, 14624 | Chemical Interaction | Hydrogen bond donor/acceptor arrays | Specifies a defined pattern of hydrogen bond donors/acceptors to guide intermolecular interactions. |
| 7, 77, 12777 | Topology | Branched vs linear chain isomerism | Differentiates between branched and linear topologies of an aliphatic chain. |
| 1601, 8808, 11765 | Conformation | Acyclic conformation: gauche/trans preference | Understands the energetic preference for gauche vs. trans conformations in acyclic chains. |

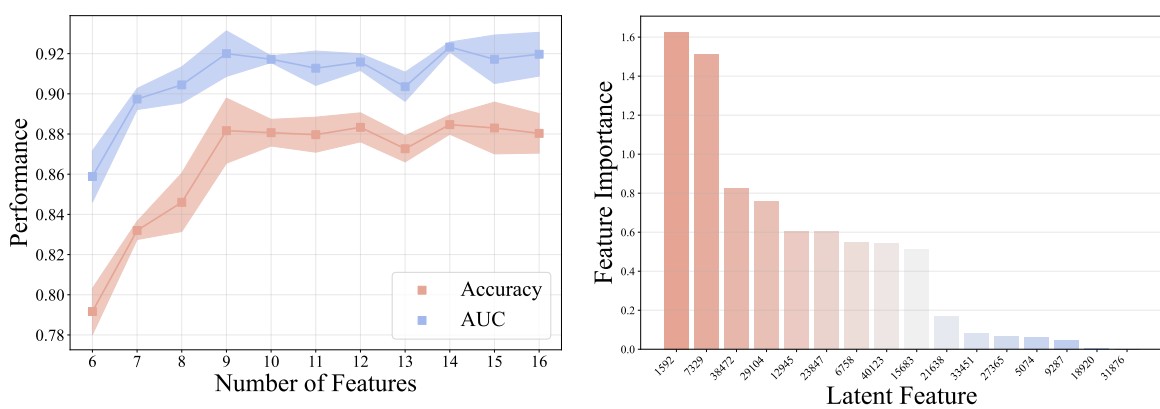

*Figure 8.* **The Left Panel:** Ablation study on the number of features. **The Right Panel:** Feature importance based on *p*-values.

pharmaceutical criteria, typically five to ten. However, most existing methods struggle to meet all requirements at once, limiting their effectiveness for complex constraint optimization. Here, we integrate SpaRE with MCTS to iteratively generate and refine candidate molecules under a comprehensive set of constraints. At each step, molecules are optimized to satisfy eight constraints: $1 < \text{LogP} < 3$, molecular weight (MW) $< 500$, topological polar surface area (TPSA) $\leq 140$, H-bond donors $\leq 5$ and acceptors $\leq 10$, aromatic rings $\leq 3$, SA score $< 5$, and QED $> 0.7$, all aligned with Lipinski-style heuristics and industry guidelines (Lipinski, 2004; 2016; Chen et al., 2020; Ritchie & Macdonald, 2009). Our approach accurately steers all criteria into the defined ranges, significantly advancing drug optimization. As shown in Figure 5, SpaRE generates molecules tightly clustered within the specified ranges after only eight optimization iterations, surpassing baselines that fail to meet complex requirements and validating its superior controllability. Additional results are provided in Appendix L.

### 4.4. Interpretable Molecule Generation

Section 4.2 demonstrates how editing routes are optimized under structural constraints and how specific edits affect molecular properties. In summary, introducing a carboxylate anion, nitro group, or primary amide at specific sites most effectively improves water solubility, while replacing a sulfonamide with a trifluoromethylsulfone best enhances

lipophilicity while maintaining structural similarity. These findings can directly inform molecular discovery in the laboratory. Motivated by this advantage, we further explore the mechanism of molecule generation in this section.

**LLM-Based Interpretation of Concept Representation.** SpaRE assumes the existence of representational patterns for concepts. We adopt an LLM-based view to explore whether such shared patterns exist in the latent space. Specifically, we prompt the LLM to reason about the activated latent features. For each feature, we collect the 500 samples with the highest activation values. These samples are analyzed by Gemini 2.5 Pro (Comanici et al., 2025), which summarizes their common patterns or semantic regularities. Through the process, Gemini distills abstract descriptions that capture the similarities underlying each latent feature, delivering human-understandable explanations for opaque latent features (see Table 3). For instance, latent feature 21 is defined as "Peroxide (R-O-O-R') bond" with the interpretation "Recognizes the peroxide (R-O-O-R') bond, a feature implying reactivity or instability". The results suggest that concepts have structured representations in LLMs' latent space, with instances of the same concept sharing common representational patterns. Additionally, we find that SpaRE learns semantically consistent concepts that generalize well to unseen data (see Appendix O for more results). These findings validate the premise for controllable generation that such latent patterns can be interpreted and thus controlled. The implementation

*Table 4.* Ablation study on expansion factor for site-specific molecule generation on the ChEBI-20 dataset (Edwards et al., 2021). The first six columns report percentages, and the last column reports numerical values.

| EXPANSION FACTOR | VALID | UNIQUENESS | NOVELTY | ATOM STA | COMPLETENESS | SUCCESS RATE | SA SCORE |
|---|---|---|---|---|---|---|---|
| 10 | 100.00 | 78.37 | 85.93 | 89.85 | 99.41 | 93.52 | 3.87 |
| 20 | 100.00 | 77.25 | 89.98 | 87.29 | 99.39 | 91.43 | 3.89 |
| 40 | 100.00 | 81.60 | 92.10 | 97.24 | 99.66 | 98.92 | 3.78 |
| 80 | 100.00 | 76.95 | 84.23 | 87.82 | 99.25 | 96.47 | 3.56 |
| 100 | 100.00 | 74.67 | 89.51 | 91.71 | 98.92 | 85.74 | 3.91 |

*Table 5.* Ablation study on LLM layer choice for site-specific molecule generation on the ChEBI-20 dataset (Edwards et al., 2021). The first six columns report percentages, and the last column reports numerical values.

| LAYER CHOICE | VALID | UNIQUENESS | NOVELTY | ATOM STA | COMPLETENESS | SUCCESS RATE | SA SCORE |
|---|---|---|---|---|---|---|---|
| 0 | 100.00 | 79.88 | 84.31 | 86.11 | 98.57 | 0.92 | 3.82 |
| 4 | 100.00 | 75.85 | 86.02 | 87.93 | 99.37 | 1.46 | 3.78 |
| 8 | 100.00 | 78.72 | 86.74 | 86.81 | 98.64 | 4.02 | 3.57 |
| 12 | 100.00 | 73.75 | 85.38 | 91.56 | 99.69 | 10.52 | 3.62 |
| 16 | 100.00 | 79.91 | 82.47 | 86.13 | 99.72 | 56.61 | 3.86 |
| 20 | 100.00 | 80.34 | 83.04 | 89.71 | 99.53 | 93.59 | 3.69 |
| 22 | 100.00 | 81.60 | 92.10 | 97.24 | 99.66 | 98.92 | 3.78 |

and additional results are given in Appendix N.

**Linear Probing for Interpretable Feature Analysis.** We seek to validate whether the learned concepts are applicable in real-world tasks. To this end, we use a linear probe (Alain & Bengio, 2016; Pagh et al., 2007) to fit a simple solubility prediction model using the activated latent features associated with solubility, as shown in Figure 2. Particularly, we select the top activated latent features for binary classification. As presented in Figure 8, the linear model achieves 88% accuracy and a 92% AUC, suggesting that as few as nine features can effectively capture chemically relevant determinants of solubility. Moreover, we use LLM-based interpretations to examine which features are most relevant to solubility (Figure 8). Notably, the most significant features (1592, 7329, and 38472) correspond to "Polarity", "Hydrophilic Groups", and "Ionic Nature", respectively. Finally, we run an upper-bound linear probe on the full feature set to estimate the ceiling of linearly accessible information. A small subset nearly matches this upper bound (88% vs. 93%), indicating a sparse representation with key information concentrated in a few features. This observation suggests that sparse latent features extracted from text descriptions may support efficient linear prediction of molecular properties. Details are provided in Appendix N, and additional results are given in Appendix P.

**Matching Input and Output Activation Patterns.** We inspect how input prompts to the LLM activate specific output molecular tokens within the representation space. Our method reveals the inputs that maximally activate particular output tokens, thereby elucidating the relationship between input semantics and token-level activations within

molecules. For example, generating 3-sulfolactic acid is strongly associated with latent activations for the prompt tokens acid, sulfo, and lactic. This explains the mechanisms through which concepts influence the generation of specific molecules. The visualization is shown in Figure 40.

### 4.5. Ablation Studies

We conduct ablation studies on the expansion factor and the LLM layer used to train the SAE. Specifically, we examine (1) how much to expand the overcomplete basis for concept completeness and (2) which LLM layer to use to train SAEs for concept control. Local control results for the expansion factor and LLM layers are reported in Table 4 and Table 5. Empirically, an expansion factor of 40 and layer 22 yield the best generative performance. For global control, we ablate the LLM layer for solubility control. Results for layers 4 and 16 are shown in Figure 9 and Figure 10. Layer 10 (see Figure 3) achieves the best controllability. In summary, we use layer 10 for global control, layer 22 for local control, and an expansion factor of 40 for the overcomplete basis.

## 5. Conclusion

We present SpaRE, an interpretability-driven framework that disentangles dense LLM representations into chemically meaningful concepts for controllable molecule generation. By modulating concept strength, SpaRE affords fine-grained control over both local and global molecular characteristics, surpassing existing methods in controllability. Experiments show SpaRE consistently generates chemically desirable molecules under complex constraints. Moreover, SpaRE is effective across diverse real-world molecular tasks, while offering actionable insights for quantitative analysis.

## Acknowledgments

This research was supported in part by the Hong Kong Polytechnic University Internal Research Fund (P0057774, P0063303 through RIAIoT) and the Research Grants Council of Hong Kong's General Research Fund (Ref. No. 15208725).

## Impact Statement

The proposed method advances controllable molecule generation by providing interpretable and granular control over various molecular characteristics, with potential benefits for drug discovery and materials design. As with other generative chemistry tools, it may have broader societal implications that include beneficial applications such as accelerating lead optimization and lowering barriers to molecular design, as well as risks such as enabling the discovery of unsafe or harmful compounds and encouraging over-reliance on automated suggestions without sufficient experimental validation. There are many potential societal consequences of our work, none of which we feel must be specifically highlighted here.

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

## A. Conditional Justification for Scheme 1: Local Concept Control

This section provides a heuristic justification for **Scheme 1** in Section 3.2, demonstrating that the proposed edit to a model's representation can controllably steer its generative output. Let $\mathbf{z}_{t_0} \in \mathbb{R}^d$ be the activation vector at a chosen layer for token position $t = t_0$. The logit, $\ell_j$, for the token $j$ in the vocabulary is computed via the affine transformation $\ell_j = \mathbf{u}_j^\top \mathbf{z}_{t_0} + b_j$, where $\mathbf{u}_j^\top$ is the $j$-th row of the model's unembedding matrix $\mathbf{W}_U$. Our objective is to increase the probability of generating a target token, $T_\mathcal{C}$, which represents a specific interpretable molecular concept $\mathcal{C}$ (*e.g.,* a desired atom or functional group). The SAE has identified a feature direction $\mathbf{v}_\mathcal{C} \in \mathbb{R}^d$ that corresponds to this concept, which is the token-specific activation $\texttt{forward\_hook}(\mathbf{z}_{t_0})$ at token position $t_0$ in this study. The core operation of Scheme 1 is to edit the activation as follows:

$$\mathbf{z}_t^\star = \mathbf{z}_t + \alpha\,\mathbf{v}_\mathcal{C} = \mathbf{z}_t + \alpha \cdot \texttt{forward\_hook}(\mathbf{z}_{t_0})$$

where $\alpha > 0$ is a scalar hyperparameter that determines the magnitude of the editing.

We now analyze the effect of this edit on the logit of the target token, $\ell_{T_\mathcal{C}}$. The new logit, $\ell_{T_\mathcal{C}}^\star$, is computed using the edited activation $\mathbf{z}_t^\star$:

$$\begin{aligned}
\ell_{T_\mathcal{C}}^\star &= \mathbf{u}_{T_\mathcal{C}}^\top \mathbf{z}_t^\star + b_{T_\mathcal{C}} \\
&= \mathbf{u}_{T_\mathcal{C}}^\top (\mathbf{z}_t + \alpha \mathbf{v}_\mathcal{C}) + b_{T_\mathcal{C}} \\
&= (\mathbf{u}_{T_\mathcal{C}}^\top \mathbf{z}_t + b_{T_\mathcal{C}}) + \alpha(\mathbf{u}_{T_\mathcal{C}}^\top \mathbf{v}_\mathcal{C}) \\
&= \ell_{T_\mathcal{C}} + \alpha(\mathbf{u}_{T_\mathcal{C}}^\top \mathbf{v}_\mathcal{C})
\end{aligned}$$

This derivation demonstrates that the change in the target logit, $\Delta\ell_{T_\mathcal{C}} = \ell_{T_\mathcal{C}}^\star - \ell_{T_\mathcal{C}}$, is equal to $\alpha(\mathbf{u}_{T_\mathcal{C}}^\top \mathbf{v}_\mathcal{C})$. The efficacy of the edit is therefore contingent upon the sign of the inner product $\mathbf{u}_{T_\mathcal{C}}^\top \mathbf{v}_\mathcal{C}$.

The principle of this method is that for a well-trained model, the unembedding vector $\mathbf{u}_{T_\mathcal{C}}$ and the concept representation $\mathbf{v}_\mathcal{C}$ are directionally aligned. This alignment emerges as a consequence of the respective training objectives. The LLM is optimized to produce activations $\mathbf{z}_t$ that align with $\mathbf{u}_{T_\mathcal{C}}$ to maximize the log-likelihood of token $T_\mathcal{C}$. Concurrently, the SAE is optimized to reconstruct these same activations from a group of sparsely activated latent features, meaning that when concept $\mathcal{C}$ is present, $\mathbf{z}_t$ can be well-approximated by a sparse linear combination of dictionary vectors dominated by $\mathbf{v}_\mathcal{C}$. For both conditions to hold, $\mathbf{u}_{T_\mathcal{C}}$ and $\mathbf{v}_\mathcal{C}$ must exhibit positive directional correlation, which implies their inner product is positive: $\mathbf{u}_{T_\mathcal{C}}^\top \mathbf{v}_\mathcal{C} > 0$. Given that $\alpha > 0$, the change in the logit, $\Delta\ell_{T_\mathcal{C}}$, is therefore strictly positive.

Finally, an increase in a token's logit leads to an increase in its generation probability. The probability of generating $T_\mathcal{C}$ is given by the softmax function as:

$$P(y_{t_0} = T_\mathcal{C}) = \frac{\exp(\ell_{T_\mathcal{C}})}{\sum_k \exp(\ell_k)}.$$

The partial derivative of this probability with respect to its logit, denoted as:

$$\frac{\partial P_{T_\mathcal{C}}}{\partial \ell_{T_\mathcal{C}}} = P_{T_\mathcal{C}}(1 - P_{T_\mathcal{C}}),$$

is strictly positive for any $P_{T_\mathcal{C}} \in (0, 1)$. Because the latent features in the SAE are disentangled, the concept vector $\mathbf{v}_\mathcal{C}$ is approximately orthogonal to the unembedding vectors of unrelated tokens. As a result, their logits are largely unaffected by the edit. The targeted increase in $\ell_{T_\mathcal{C}}$ thus robustly increases the probability of generating desired token $T_\mathcal{C}$. Under the alignment assumption above, this derivation provides heuristic support for local control via sparse representation editing. $\square$

# B. Conditional Justification for Scheme 2: Global Concept Control

This section provides a heuristic justification for **Scheme 2** in Section 3.2, which aims to control a global property $\mathcal{P}$ over the entire generated molecule. Unlike local control, which targets a single token, global control modifies the whole generative trajectory toward the molecular property by applying consecutive edits to the model's generative steps. Let $\mathbf{v}_{\mathcal{P}} \in \mathbb{R}^d$ be the concept representation corresponding to the global property $\mathcal{P}$, as identified by the SAE. The control mechanism is implemented by editing the activation $\mathbf{z}_t$ at every step $t$ of the autoregressive generation process:

$$\mathbf{z}_t^{\star} = \mathbf{z}_t + \alpha \mathbf{v}_{\mathcal{P}} \tag{2}$$

where $\alpha$ is a scalar hyperparameter. An $\alpha$ close to 1 is used to enhance the property, while an $\alpha$ close to 0 is used to suppress it. This method of applying a constant directional offset throughout generation is a form of representation engineering designed to steer model behavior.

The core of the proof lies in analyzing how this persistent edit influences the logit distribution at an arbitrary generative step $t$. For any token $j$ in the vocabulary, the original logit is $\ell_{t,j} = \mathbf{u}_j^{\top} \mathbf{z}_t + b_j$. The edited logit, $\ell_{t,j}^{\star}$, is computed using the modified activation $\mathbf{z}_t^{\star}$:

$$\begin{aligned} \ell_{t,j}^{\star} &= \mathbf{u}_j^{\top} \mathbf{z}_t^{\star} + b_j \\ &= \mathbf{u}_j^{\top} (\mathbf{z}_t + \alpha \mathbf{v}_{\mathcal{P}}) + b_j \\ &= (\mathbf{u}_j^{\top} \mathbf{z}_t + b_j) + \alpha(\mathbf{u}_j^{\top} \mathbf{v}_{\mathcal{P}}) \\ &= \ell_{t,j} + \alpha(\mathbf{u}_j^{\top} \mathbf{v}_{\mathcal{P}}) \end{aligned}$$

This result demonstrates that at every step, the logit of each token $j$ is shifted by an amount $\Delta \ell_{t,j} = \alpha(\mathbf{u}_j^{\top} \mathbf{v}_{\mathcal{P}})$. This shift is constant for a given token across all generative steps and is proportional to the alignment between the token's unembedding vector $\mathbf{u}_j$ and the global property vector $\mathbf{v}_{\mathcal{P}}$.

A global property $\mathcal{P}$ is determined not by any single token, but by the collective influence of the entire token sequence. Accordingly, the associated concept vector $\mathbf{v}_{\mathcal{P}}$ represents an abstract direction in activation space. For this vector to be meaningful, it should be directionally aligned with the unembedding vectors of tokens that amplify the property, and anti-aligned with those that suppress it. For example, if $\mathcal{P}$ corresponds to "high solubility," we expect $\mathbf{v}_{\mathcal{P}}$ to have a positive inner product with unembedding vectors for polar functional groups (*e.g.,* "O" for hydroxyls, "N" for amines), and a negative or near-zero inner product with tokens corresponding to long, non-polar alkyl chains. This alignment naturally arises from how LLMs compose abstract concepts from lower-level features. As a result, the edit systematically increases the logits of tokens positively correlated with property $\mathcal{P}$ and decreases the logits for those negatively correlated.

By applying this directional bias at every step, we do not force any individual token. Rather, we reshape the token distribution so that tokens aligned with property $\mathcal{P}$ become more likely. Under the assumption that $v_P$ captures a meaningful property direction, the cumulative effect of these step-wise logit shifts can bias the final molecule toward exhibiting property $\mathcal{P}$. This derivation therefore serves as heuristic support for global control via sparse representation editing. $\square$

# C. Implementation Details

Our method begins by constructing an adapted foundation model tailored for token-specific generation of atoms and functional groups. To achieve this, we utilize Group SELFIES (Cheng et al., 2023), a textual representation that encodes molecules as sequences of atoms and functional group units, such that each input token corresponds to either an atom or a functional group. We first fine-tune the molecular foundation model to improve its understanding of Group SELFIES. Based on MolXPT (Liu et al., 2023b), a molecular LLM based on the GPT-2 architecture (Brown et al., 2020), we introduce key modifications to the data preparation and training pipelines. Specifically, all fine-tuning data, including ChEBI-20 (Edwards et al., 2021), GEOM-DRUGS and GEOM-QM9 (Axelrod & Gomez-Bombarelli, 2022), are systematically converted from their original SMILES format to Group SELFIES, enhancing both structural fidelity and token consistency. For fine-tuning, we employ the AdamW optimizer with a learning rate of $5 \times 10^{-5}$, $\epsilon$ of $1 \times 10^{-8}$, and no weight decay. The learning rate is scheduled using a linear decay strategy without warmup. The model is then fine-tuned on the Group SELFIES-formatted dataset with the objective of minimizing the standard autoregressive language modeling loss: maximizing the log-likelihood of each subsequent token given the preceding sequence. For a token sequence $T = (t_1, t_2, \ldots, t_n)$, the fine-tuning loss is defined as:

$$\mathcal{L}_{\text{fine-tuning}} = -\sum_{i=1}^{n} \log P(t_i \mid t_1, \ldots, t_{i-1}; \theta),$$

where $\theta$ denotes model parameters. Fine-tuning is deliberately limited to 2 epochs to ensure sufficient training on Group SELFIES-based data while preventing overfitting.

After completing the fine-tuning, we introduce SAEs to analyze the model's internal representations. Independent SAEs are trained on activations from each transformer layer, mapping each 1,024-dimensional activation vector to a 40,960-dimensional overcomplete feature space. We select the expansion factor of $40\times$ because such a large, overcomplete basis provides sufficient vocabulary for the model to decompose complex, distributed activations into a set of highly disentangled and semantically interpretable sparse features.

To train the SAEs, we sample $10^5$ molecules from the three datasets used for fine-tuning and generate approximately $2.61 \times 10^6$ activation samples. Training is conducted with a batch size of 1024 for 8 epochs. We employ the Muon optimizer (Jordan et al., 2024), which exhibited faster convergence and better avoidance of local minima compared to AdamW. Notably, although the initial feature sparsity is about 50%, the trained models achieved extremely high sparsity exceeding 99.7%, demonstrating the effectiveness of the training strategy. The encoder consists of a linear layer followed by a ReLU activation, yielding non-negative sparse activations. For feature selection and visualization, we rescale activation magnitudes to a common range on a per-feature basis. The SAE itself is trained on the raw non-negative activations. As given in Equation (1), the SAE is trained to minimize a composite loss comprising mean squared error for reconstruction and an $L_1$ sparsity penalty, which promotes highly sparse features while preserving reconstruction fidelity.

For local control (**Scheme 1**, Section 3.2), we aim to control the generation of a specified atom or functional group at a particular token position $t_0$. Experiments demonstrated that activation vectors at layer 22 (close to model output) are most effective for such fine-grained, local control. The workflow for obtaining a token-specific concept representation is as follows: (1) assemble a set of activation vectors immediately before the generation of the target token at position $t = t_0$ via the hook retrieval; (2) use the SAE to extract activated latent features, retaining those with activation values above 0.5 in 100% of samples; and (3) derive the concept representation by averaging the activation vectors (*i.e.,* corresponding SAE dictionary vectors) of the identified latent features. For example, when the target token corresponds to a single atom or functional group, this process typically identifies approximately 15 latent features that are strongly associated with each token. In other words, a set of about 15 latent features collectively corresponds to a single atom or functional group, such as a carbon atom or a nitro group. Unlike the continuous modulation used for property control, this editing is discrete: the edited concept representation is injected into layer 22 only at the timestep immediately prior to the target token's generation (*i.e.,* token position $t = t_0$), with strength coefficient $\alpha$ ranging from 0 to 1. This increases the generation probability of the target token, thereby steering the molecule generation as desired.

A key tool used to obtain local concepts (atoms or functional groups) is the forward hook. The forward hook is a PyTorch mechanism (via `torch.nn.Module.register_forward_hook`) that lets users run a custom function immediately after a module's forward pass. To extract the activation at a specific token position $t_0$ from a transformer block, you can attach a forward hook to that block, receive its output tensor of shape $[B, T, d]$, and index `[:, t_0, :]` to capture the desired activation. Forward pre-hooks (`register_forward_pre_hook`) intercept or modify inputs before computation, and full backward hooks (`register_full_backward_hook`) observe gradients during backpropagation. In Transformer-based

workflows, hooks are widely used to fetch intermediate hidden states and attention maps, construct probes or visualizations, perform distillation, or intervene online. In this work, hooks are used for activation extraction and modification.

For global control (**Scheme 2**, Section 3.2), we target the generation of molecules with specific molecular properties. Empirical analysis revealed that activation vectors at layer 10 are optimal for controlling such global properties. The procedure for obtaining a concept representation is as follows: (1) define positive and negative sample sets by thresholding a quantitative chemical metric via the hook retrieval; (2) use the SAE trained on layer 10 to extract and compare activation patterns between these two sets; (3) retain latent features with activation values exceeding 0.5 in all positive samples and below 0.5 in all negative samples; and (4) construct the concept representation for the target property by averaging the activation vectors (*i.e.,* corresponding SAE dictionary vectors) over the entire dataset (both positives and negatives). For example, in lipophilicity control, we use LogP as the metric, assigning molecules with $\text{LogP} \geq 2$ as positive and $\text{LogP} < 2$ as negative. Typically, this process identifies approximately 30 highly activated latent features for the property of interest (as they exhibit greater complexity than local concepts). During generation, the resulting concept representation is continuously injected into layer 10 at every timestep $t$, modulated by a strength coefficient $\alpha$, thereby steering the generative process toward the desired global molecular structural and physicochemical properties. Notably, all local and global concepts are constructed strictly from the training split, with no overlap with validation or test data.

The model fine-tuning is conducted on $2\times$ NVIDIA H100 GPUs for 16 hours, and SAE training is performed on $1\times$ NVIDIA A100 GPU for 8 hours. Once trained, the SAEs can be reused for inference without retraining. We apply local control at a single layer (empirically, layer 22) and global control at another (*i.e.,* layer 10). The edit is a precomputed vector in activation space injected via a forward hook, requiring no SAE forward pass and consistent with our fastest runtime among baselines. The hyperparameters used for fine-tuning and SAE training are summarized in Table 6 and Table 7, respectively.

*Table 6.* Hyperparameters for Fine-Tuning

| Hyperparameter | Value |
|---|---|
| Base Model | MolXPT (Liu et al., 2023b) |
| Transformer Layers | 24 |
| Attention Heads | 16 |
| Hidden Dimension | 1024 |
| Data Format | Group SELFIES |
| Max Sequence Length | 1024 |
| Optimizer | AdamW (Loshchilov & Hutter, 2019) |
| Learning Rate | 5e-5 |
| LR Schedule | Linear decay (no warmup) |
| Weight Decay | 0.0 |
| Adam $\beta_1$ | 0.9 |
| Adam $\beta_2$ | 0.999 |
| Adam $\epsilon$ | 1e-8 |
| Batch Size | 64 |
| Training Epochs | 2 |

*Table 7.* Hyperparameters for SAE Training

| Hyperparameter | Value |
|---|---|
| Input Dimension | 1024 |
| Expansion Factor | 40× |
| Hidden (Sparse) Dimension | 40,960 |
| Encoder Activation | ReLU + $L_2$ norm |
| Reconstruction Loss | MSE |
| Sparsity Penalty | $L_1$ |
| $L_1$ Coef. ($\lambda$) | Tuned per layer (1e-5–1e-4) |
| Optimizer | Muon (Jordan et al., 2024) |
| Learning Rate | 1e-4 |
| LR Schedule | Cosine Annealing |
| Batch Size | 1024 |
| Training Epochs | 8 |
| Dataset | ∼2.61M activations |

## D. Experimental Setup

We provide further details about our experimental setup, including the dataset, baseline models, and evaluation metrics.

**Dataset**. We employ three widely used benchmark datasets to train SAEs and fine-tune the foundation model, as well as to conduct experimental evaluations. Specifically, ChEBI-20 (Edwards et al., 2021), GEOM-DRUGS and GEOM-QM9 (Axelrod & Gomez-Bombarelli, 2022) are used. In our preprocessing pipeline, all molecules are converted to Group SELFIES format, making them suitable for our text-based molecule generation tasks. The three datasets contain around 27K, 140K, and 80K molecules that meet our experimental requirements, respectively. The training, validation, and test sets are split in a 70:15:15 ratio. Detailed descriptions of each dataset are provided below.

1. **ChEBI-20**: ChEBI-20 is a curated subset of the Chemical Entities of Biological Interest (ChEBI) resource, focusing on approximately twenty major chemical classes relevant to bioactive small molecules. It provides standardized molecular structures with ontology-based class labels, supporting conditional and class-aware molecule generation and evaluation. Molecules are typically represented as SMILES, with quality-controlled annotations for property-conditioned generation, scaffold-aware sampling, and class-balanced benchmarking. The ontology grounding enables interpretable analysis of generative coverage across chemically meaningful categories.

2. **GEOM-DRUGS**: GEOM-DRUGS is part of the GEOM corpus of molecular geometries, targeting drug-like chemical space. It offers ensembles of low-energy conformers per molecule, computed using high-quality quantum-chemical and force-field pipelines, along with associated energies. This dataset emphasizes conformational diversity and realistic geometries, making it well-suited for generative modeling, conformation generation, and energy-aware sampling. It includes representations such as 3D atomic coordinates, bond graphs, and energetics.

3. **GEOM-QM9**: GEOM-QM9 extends the classic QM9 dataset by providing comprehensive 3D conformer ensembles for each molecule in the QM9 chemical space (C, H, O, N, F with up to 9 heavy atoms). It supplies multiple optimized conformations and relative energies per molecule, offering a richer view of the accessible conformational landscape compared to single-geometry datasets. This makes GEOM-QM9 a strong benchmark for generative models requiring both chemical validity and 3D variability, within a well-defined and widely used domain.

**Baselines**. We compare the performance of SpaRE against a comprehensive suite of baseline models, including MolXPT (Liu et al., 2023b), BioT5 (Pei et al., 2023), BioT5+ (Pei et al., 2024), LDMol (Chang & Ye, 2025), NExT-Mol (Liu et al., 2025b), TGM-DLM (Gong et al., 2024), Atomas (Zhang et al., 2025), CDGS (Huang et al., 2023a), JODO (Huang et al., 2023b), RetMol (Wang et al., 2023), MARS (Xie et al., 2021), MolEvol (Chen et al., 2021), and Llamole (Liu et al., 2025a). To ensure a thorough evaluation, we select baselines spanning GNN-, diffusion-, and autoregressive-based approaches. For property- or structure-controlled generation, we include RetMol (Wang et al., 2023), Llamole (Liu et al., 2025a), Atomas (Zhang et al., 2025), MolEvol (Chen et al., 2021), and TGM-DLM (Gong et al., 2024) as baselines, since these models are designed for controllable molecule generation tasks. Their generation can be controlled via text prompts or

conditional guidance, enabling fair comparison of controllable molecule generation across models. Particularly, GNN-based methods model molecules as graphs and iteratively edit them (*e.g.,* via evolutionary algorithms), offering direct structural control but at higher computational cost. Diffusion-based methods avoid discrete tokens, generating atom types and 3D coordinates via denoising to produce full conformers in a non-autoregressive manner. Autoregressive methods (including ours) tokenize molecules as strings and perform next-token prediction with LLM-style architectures. To ensure fairness, we used identical inputs, metrics, and constraint definitions across all baselines.

**Evaluation Metrics**. For **molecule generation quality**, our evaluation metrics cover multiple facets of molecule generation. Validity measures the fraction of molecules that are chemically parsable and valence-correct, ensuring adherence to basic chemical rules. Uniqueness quantifies the proportion of non-duplicate valid molecules, reflecting diversity and resistance to mode collapse. Novelty reports the share of valid molecules not present in the training set, indicating generalization beyond memorization. Atom stability (atom sta) assesses the rate of atoms with permissible valences and charges, capturing the chemical soundness. Completeness measures how often outputs fully satisfy required scaffolds or constraints without missing fragments, indicating adherence to specification. SA score (lower is better) estimates synthetic accessibility, approximating how feasible a molecule is to make in practice. Together, these metrics assess chemical correctness, diversity, and constraint satisfaction while accounting for real-world synthesizability, providing a comprehensive view of controllable molecule generation quality. **Notably, we repeat each experiment ten times and report the mean in our study**.

For **controllability**, we define the control success rate (CSR) as the proportion of generated molecules that successfully satisfy the specified control constraints, formulated as follows:

$$\text{CSR} = \frac{N_{\text{success}}}{N_{\text{total}}},$$

where $N_{\text{success}}$ denotes the number of molecules that successfully meet the defined constraints, and $N_{\text{total}}$ denotes the total number of generated molecules.

For **efficiency**, we use average computation time for generating a single molecule, measured in milliseconds (ms), as the metric. This reflects the computational speed of each approach.

For **molecular structural and physicochemical properties**, we leverage a set of commonly used descriptors. The octanol–water partition coefficient (LogP) quantifies lipophilicity, which modulates membrane partitioning and can be used to assess permeability and solubility. The topological polar surface area (TPSA) estimates the solvent-accessible polar surface contributed by heteroatoms and polar bonds and is widely used as a predictor of passive permeability and absorption (Prasanna & Doerksen, 2009). The quantitative estimate of drug-likeness (QED) aggregates multiple molecular descriptors (*e.g.,* MW, LogP, HBD/HBA, TPSA, rotatable bonds, aromatic ring count, and structural alerts) into a single interpretable score that reflects overall drug-likeness characteristics (Bickerton et al., 2012). To approximate synthetic tractability, we report both the synthetic accessibility (SA) score and the retrosynthetic accessibility (RA) score. The SA score provides a fast, heuristic estimate combining fragment/substructure frequency and molecular complexity penalties; lower SA score typically indicates easier synthesis (Ertl & Schuffenhauer, 2009). Complementarily, the RA score leverages machine learning over retrosynthetic analysis to estimate the practical ease of synthesis in planning-based settings (Thakkar et al., 2021). In addition, we use Tanimoto similarity (Bajusz et al., 2015) to compute both scaffold similarity and pharmacophore similarity. These metrics are used to compare the core molecular frameworks and the arrangement of essential interaction features relevant to target recognition, respectively. Aromaticity is evaluated using three complementary metrics. The harmonic oscillator model of aromaticity (HOMA) probes bond-length equalization as a geometric criterion (Kruszewski & Krygowski, 1972). The nucleus-independent chemical shift (NICS) assesses the magnetic response associated with diatropic ring currents. In this study, we report NICS values at standardized probe positions to improve discriminability (Chen et al., 2005). Aromatic stabilization energy (ASE) estimates the degree of resonance stabilization by comparing a given ring (or polycyclic system) to appropriate nonaromatic reference states. Since ASE is both method- and reference-dependent, its values are interpreted comparatively within a consistent computational protocol in this study (Schleyer & Pühlhofer, 2002).

# E. More Ablation Studies

We conduct ablation studies on the expansion factor and the LLM layer used to train the SAE. Specifically, we examine (1) how much to expand the overcomplete basis for concept completeness and (2) which LLM layer to use to train an SAE for concept control. Local-control results for the expansion factor and LLM layers are reported in Table 4 and Table 5. Empirically, an expansion factor of 40 and layer 22 yield the best generative performance. For global control, we ablate the LLM layer for solubility control. Results for layers 4 and 16 are shown in Figure 9 and Figure 10; layer 10 (Figure 3) achieves the best controllability. In summary, we use layer 10 for global control, layer 22 for local control, and an expansion factor of 40 for the overcomplete basis.

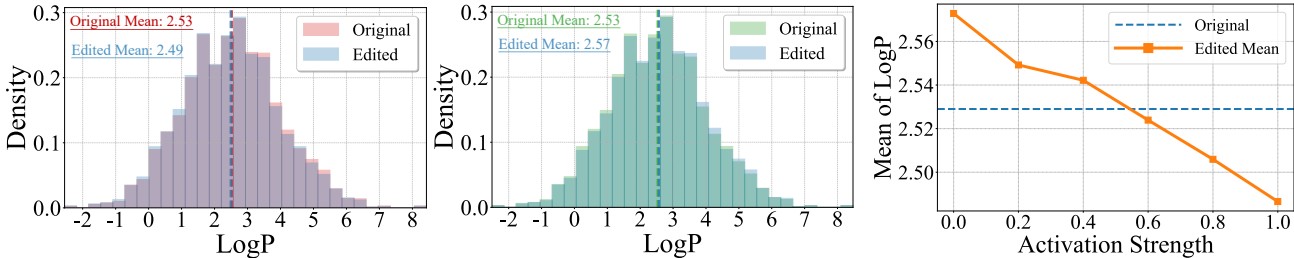

*Figure 9.* Distribution of molecules generated under solubility control with LLM layer 4: (**Left**) amplification, (**Middle**) suppression, and (**Right**) controllable tuning by varying activation strength.

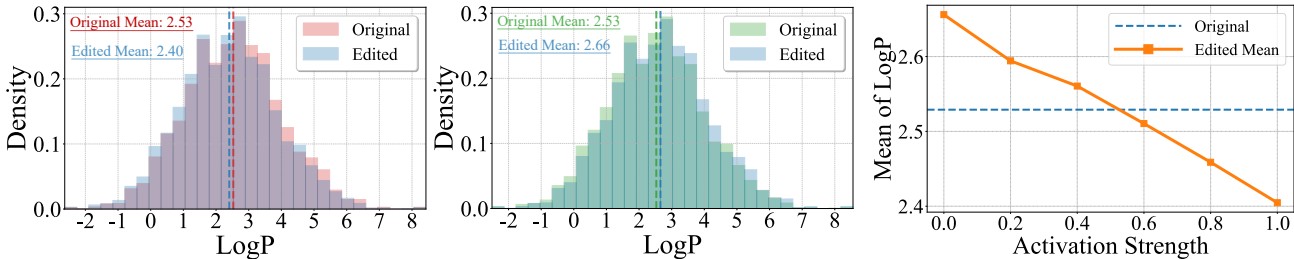

*Figure 10.* Distribution of molecules generated under solubility control with LLM layer 16: (**Left**) amplification, (**Middle**) suppression, and (**Right**) controllable tuning by varying activation strength.

# F. Local Concept Control

We further evaluate local, site-specific molecule generation on the GEOM-DRUGS dataset, as reported in Table 8. SpaRE sets a new state-of-the-art performance on this dataset: it matches the best validity and achieves the highest novelty, atom stability, and near-perfect completeness. Most notably, SpaRE attains the highest success rate, demonstrating its fine-grained controllability over site-specific edits rather than generic distributional matching among existing approaches. In addition to quality and control, SpaRE also delivers the best synthesizability and the fastest runtime, outperforming GNN-, diffusion-, and autoregressive-based baselines, which typically trade off control, generation quality, or synthesizability. This study indicates that SpaRE's improvements are robust and consistent: high novelty, validity, and atom stability show that SpaRE explores new chemical space while maintaining correct bonding and valence. Strong completeness further confirms that SpaRE reliably realizes the requested site-specific constraints without damaging molecular structures, rather than resorting to coarse or partial edits. The combination of high success rate and low SA score suggests edits that are both controllable

and synthetically tractable, aligning with real-world molecular design and optimization. In contrast, an MCTS-based naive string substitution (results shown in Table 9) attains high validity and novelty yet fails on atom stability and synthesizability, demonstrating that the performance gains are not attributable to the search itself but to SpaRE's chemistry-aware (*i.e.,* capturing global information in the molecular context) editing rather than local token operations. Furthermore, Table 10 reports the performance of general-purpose LLMs on local concept control.

*Table 8.* Local molecule generation on the GEOM-DRUGS dataset (Axelrod & Gomez-Bombarelli, 2022). Quality and controllability are reported as percentages, while synthesizability and efficiency are reported as numerical values. **Best** and second-best results are indicated in bold and underlined, respectively.

| | QUALITY (↑) | | | | | CONTROL(↑) | SYNTHESIS(↓) | EFFICIENCY (↓) |
|---|---|---|---|---|---|---|---|---|
| MODEL | VALID | UNIQUENESS | NOVELTY | ATOM STA | COMPLETENESS | SUCCESS RATE | SA SCORE | TIME |
| *GNN-Based* | | | | | | | | |
| MARS | 87.62 | 81.62 | 80.47 | 89.88 | 97.83 | 32.86 | 4.26 | 371.28 |
| MolEvol | 86.24 | 78.86 | 82.81 | 91.47 | 96.82 | 38.43 | 4.24 | 252.62 |
| *Diffusion-Based* | | | | | | | | |
| LDMol | 86.49 | 83.01 | 80.09 | 89.16 | 95.24 | 31.74 | 3.80 | 346.19 |
| TGM-DLM | 92.19 | 78.85 | 77.35 | 91.12 | 96.44 | 27.12 | 4.50 | 315.32 |
| CDGS | 90.26 | 83.47 | 81.07 | 91.37 | 97.13 | 31.05 | 3.84 | 308.64 |
| JODO | 86.85 | 78.87 | 85.72 | 90.49 | 92.89 | 37.29 | 4.39 | 345.81 |
| *Autoregressive-Based* | | | | | | | | |
| MolXPT | 87.05 | 83.74 | 86.76 | 91.27 | 96.65 | 21.09 | 3.96 | 21.36 |
| BioT5 | **100.00** | **85.03** | 82.53 | 89.99 | 95.87 | 28.17 | 4.83 | 29.58 |
| BioT5+ | **100.00** | 75.21 | 85.41 | 89.19 | 98.05 | 21.63 | 4.43 | 24.13 |
| NExT-Mol | 88.14 | 74.98 | 86.35 | 92.31 | 95.83 | 37.02 | 4.54 | 337.77 |
| Atomas | 87.69 | 75.85 | 87.74 | 90.00 | 94.56 | 33.41 | 5.35 | 332.75 |
| RetMol | 84.62 | 79.24 | 84.58 | 93.26 | **99.92** | 57.84 | 3.86 | 312.62 |
| Llamole | 91.84 | 79.26 | 82.47 | 89.69 | 98.68 | 39.87 | 4.01 | 203.49 |
| SpaRE (**Ours**) | **100.00** | 82.40 | **98.57** | **96.32** | 99.13 | **96.77** | **3.51** | **12.80** |

*Table 9.* The results of MCTS-based naive string substitution on two datasets. Quality and controllability are reported as percentages, while synthesizability and efficiency are reported as numerical values.

| | QUALITY (↑) | | | | | CONTROL(↑) | SYNTHESIS(↓) | EFFICIENCY (↓) |
|---|---|---|---|---|---|---|---|---|
| DATASET | VALIDITY | UNIQUENESS | NOVELTY | ATOM STABILITY | COMPLETENESS | SUCCESS RATE | SA SCORE | TIME |
| ChEBI-20 | 100.00 | 99.53 | 98.38 | 63.94 | 82.29 | - | 7.55 | 1.82 |
| GEOM-DRUGS | 100.00 | 97.29 | 96.97 | 41.51 | 92.35 | - | 7.85 | 1.56 |

*Table 10.* Evaluation of recent general-purpose LLMs on local control on the ChEBI-20 (Edwards et al., 2021). All models are prompted with the same task instructions and decoded under the same post-processing and evaluation pipeline.

| MODEL | VALID | UNIQUENESS | NOVELTY | ATOM STA | COMPLETENESS | SUCCESS | SA SCORE |
|---|---|---|---|---|---|---|---|
| GPT-5.4 | 88.60 | 80.83 | 83.79 | 92.11 | 95.41 | 56.98 | 3.97 |
| Claude Opus 4.6 | 91.02 | 79.60 | 84.77 | 89.74 | 95.46 | 66.82 | 4.02 |
| Gemini 3.0 Pro | 92.05 | 77.53 | 84.07 | 91.52 | 96.91 | 57.27 | 3.86 |
| SpaRE (**Ours**) | 100.00 | 81.60 | 92.10 | 97.24 | 99.66 | 98.92 | 3.78 |

# G. Global Concept Control

We extensively control the global concept of *aromaticity* (as presented in Figure 11, Figure 12, and Figure 13), *ring systems* (as presented in Figure 14), *hydrogen bonding* (as presented in Figure 15, Figure 16, and Figure 17), and *ortho-disubstituted positions* (as presented in Figure 18, Figure 19, and Figure 20). We apply both amplification and suppression of concepts using the global control scheme, and perform an ablation study to evaluate the effect of activation strength.

The results show that SpaRE can steer several global concepts in a tunable manner. For aromaticity, amplification increases ASE and HOMA while decreasing NICS, whereas reversing the edit direction produces the opposite trend. Ring-system control also shifts this distributed structural property in a graded way. Hydrogen-bonding control changes HBA/HBD counts in the expected directions, and ortho-disubstitution control modulates both counts and steric values. Overall, the clear separation between opposite edit directions and the smooth response to edit magnitude suggest that SpaRE captures chemically meaningful global features while largely preserving chemical validity.

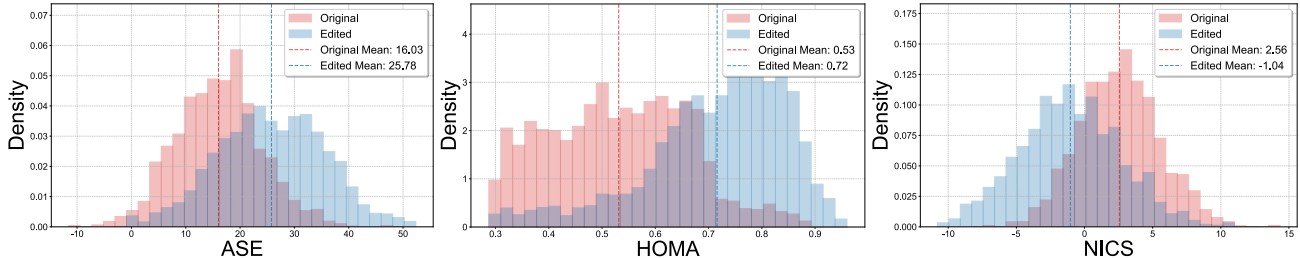

*Figure 11.* Distribution of molecules generated under aromaticity amplification. Aromaticity is evaluated by ASE (**Left**), HOMA (**Middle**), and NICS (**Right**): higher ASE and HOMA, and lower NICS, indicate increased aromaticity.

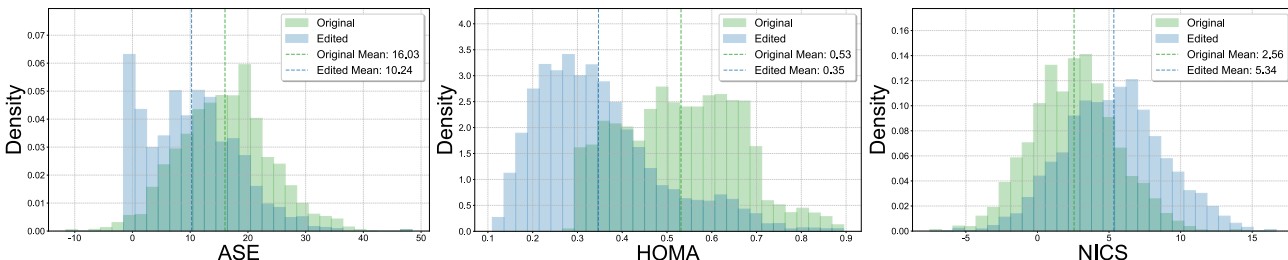

*Figure 12.* Distribution of molecules generated under aromaticity suppression. Aromaticity is evaluated by ASE (**Left**), HOMA (**Middle**), and NICS (**Right**): lower ASE and HOMA, and higher NICS, indicate reduced aromaticity.

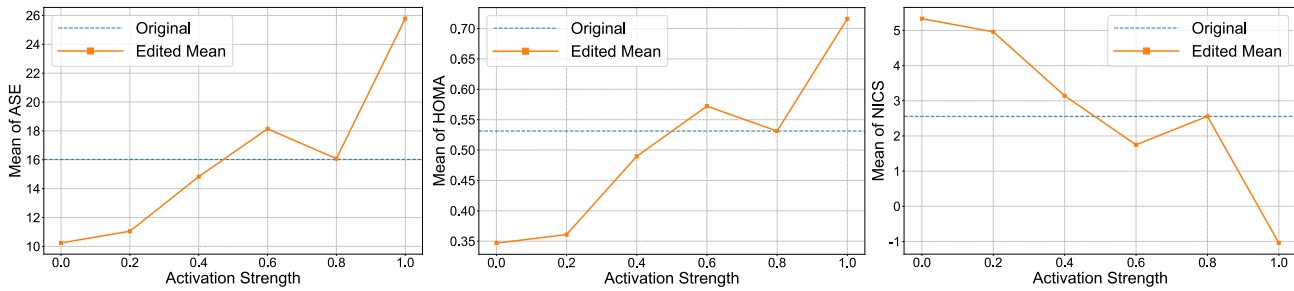

*Figure 13.* Tuning activation strength enables adjustment of aromaticity, as measured by ASE (**Left**), HOMA (**Middle**), and NICS (**Right**).

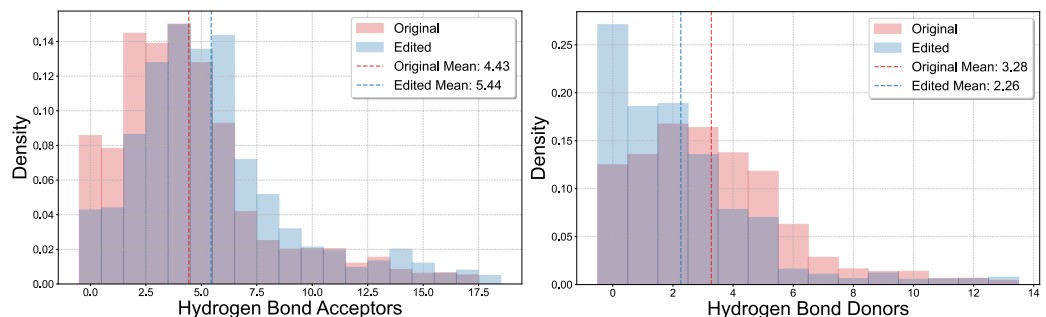

*Figure 14.* Distribution of molecules generated under global control of rings: amplification (**Left**) and suppression (**Middle**) with smooth tunability via changes in activation strength (**Right**).

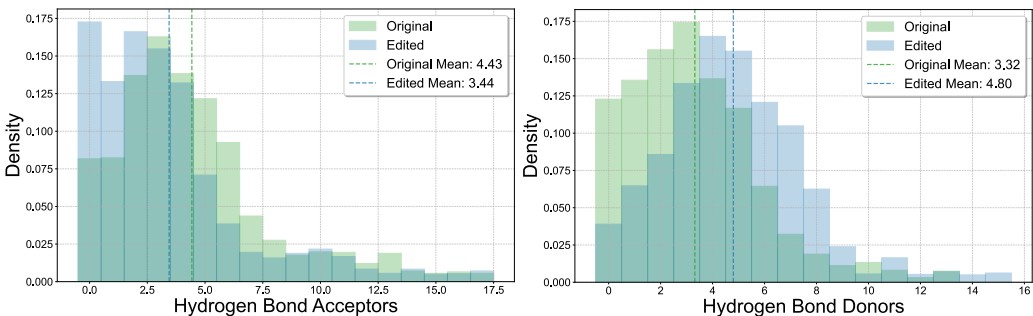

*Figure 15.* Distribution of molecules generated under hydrogen bond acceptor amplification. Hydrogen bonding is evaluated by the count of HBA (**Left**) and HBD (**Right**).

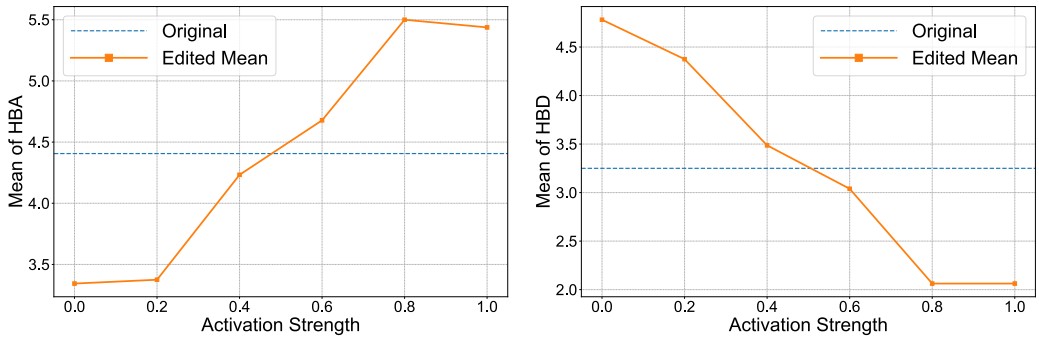

*Figure 16.* Distribution of molecules generated under hydrogen bond acceptor suppression. Hydrogen bonding is evaluated by the count of HBA (**Left**) and HBD (**Right**).

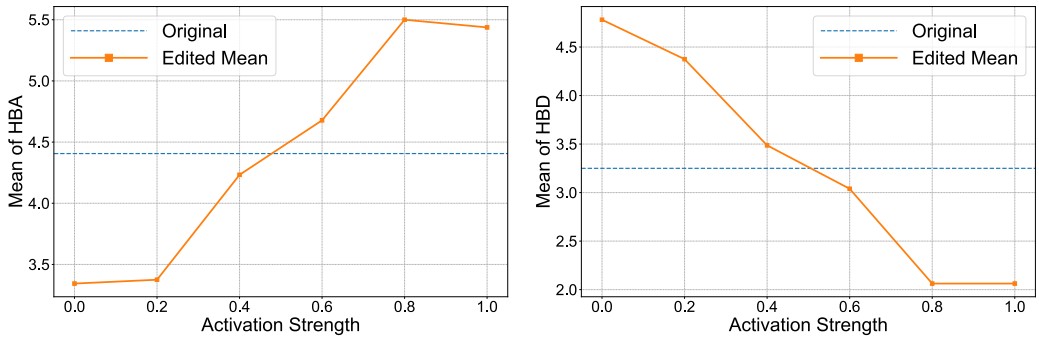

*Figure 17.* Tuning activation strength enables adjustment of hydrogen bonding, as measured by the count of HBA (**Left**) and HBD (**Right**).

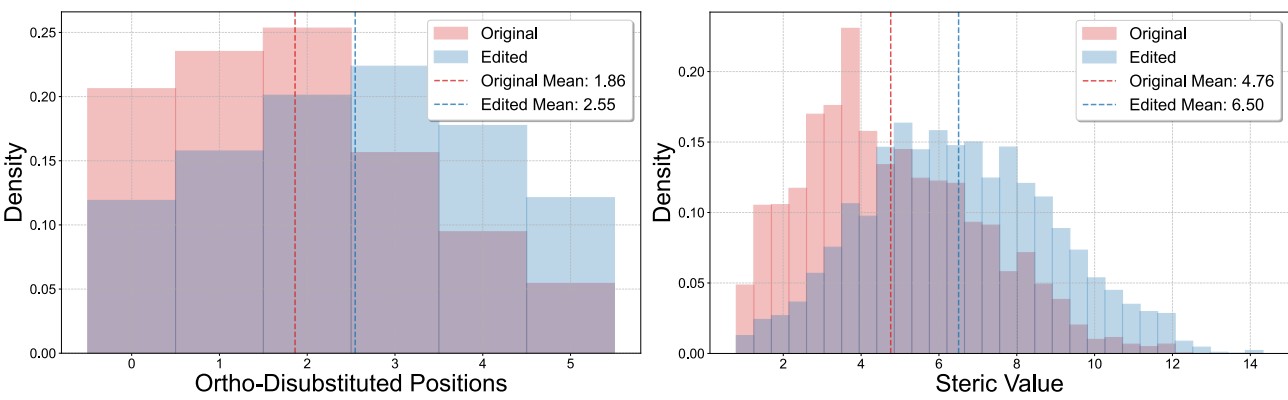

*Figure 18.* Distribution of molecules generated under ortho-disubstituted position amplification. Ortho-disubstituted positions are evaluated by their count (**Left**) and steric value (**Right**), with larger steric value indicating more ortho-disubstituted positions.

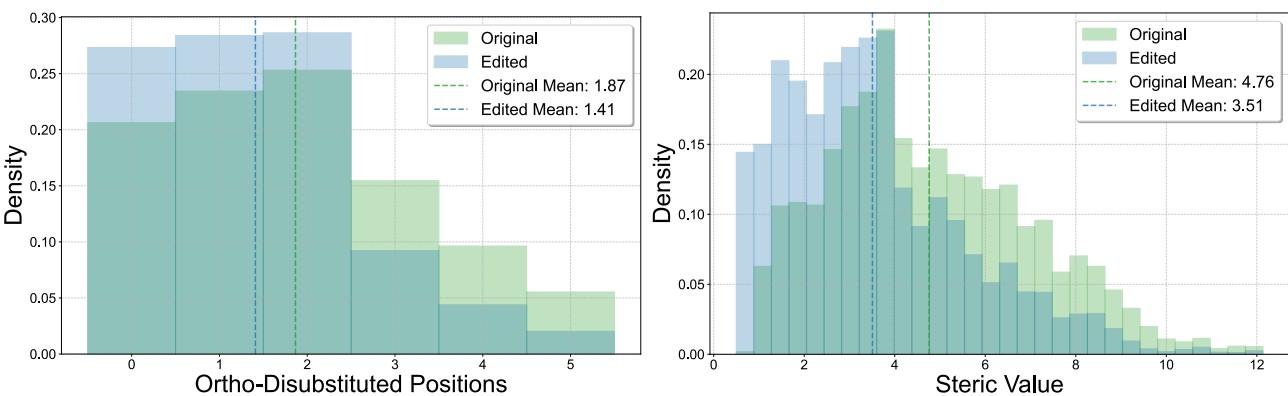

*Figure 19.* Distribution of molecules generated under ortho-disubstituted position suppression. Ortho-disubstituted positions are evaluated by their count (**Left**) and steric value (**Right**), with lower steric value indicating less ortho-disubstituted positions.

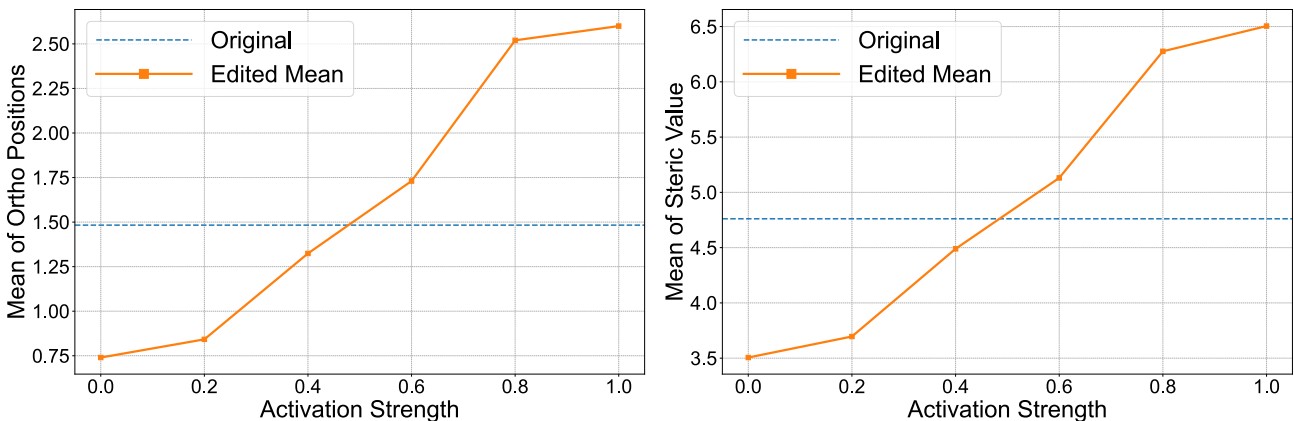

*Figure 20.* Tuning activation strength enables adjustment of ortho-disubstituted positions, as measured by their count (**Left**) and steric value (**Right**).

## H. Bioisosteric Editing

We perform multi-step bioisosteric editing using expert-curated bioisostere pairs to systematically improve water solubility (Figure 21) and lipophilicity (Figure 4) while preserving structural similarity (Lipinski, 1986; Meanwell, 2011; Subbaiah & Meanwell, 2021). Starting from a lead-like scaffold, we design multi-step routes where each step replaces a single atom or functional group with a bioisostere chosen to maintain electronic distribution, size, hydrogen-bonding capacity, and polarity. Solubility-directed edits decrease LogP and increase polar features (HBA/HBD counts, TPSA), while lipophilicity-directed edits increase LogP with TPSA reduction. Scaffold and pharmacophore similarity remain above 60% throughout, minimizing structural perturbation and retaining binding-relevant motifs. This approach leverages bioisosteric principles to precisely modulate molecular properties with strong structural continuity, enabling incremental property optimization while maintaining synthetic and biological plausibility.

Across both water solubility and lipophilicity optimization routes, expert-curated bioisosteres enable controllable property tuning while preserving molecular scaffold. For lipophilicity, introducing hydrophobic or electron-modulating motifs (*e.g.,* –OH⇒–OCH$_3$, –COOH⇒–COOCH$_3$, –NH$_2$⇒–N(CH$_3$)$_2$, Ar–H⇒Ar–Cl, pyridyl⇒phenyl) consistently increases LogP and moderately reduces TPSA, while maintaining key structural features. For water solubility, polarity- and H-bond–enhancing edits (*e.g.,* –SH⇒–OH, Cl⇒CN, ketone⇒sulfone, ester⇒amide, –S–⇒–O–) lower LogP and increase TPSA, HBA, and HBD. In all cases, scaffold and pharmacophore similarity remain above 60%, limiting conformational drift and retaining binding-relevant features. These results demonstrate that our bioisosteric editing strategy enables practical, property-driven lead optimization while maintaining synthetic feasibility and structural integrity.

SpaRE produces stable property optimization with minimal structural disruption: all of the five-step routes achieve effective property improvements while maintaining ≥60% scaffold/pharmacophore similarity at each step. Its precise edits make LogP, TPSA, HBA, and HBD changes directly attributable. Leveraging an expert-curated bioisostere library ensures edits remain synthetically tractable and biologically plausible, with high control success rates and low computational costs (*i.e.,* within five steps). Visualizations of the four routes are shown in Figure 24 (lipophilicity) and Figure 25 (water solubility), with QED in Figure 22 (lipophilicity) and Figure 23 (water solubility).

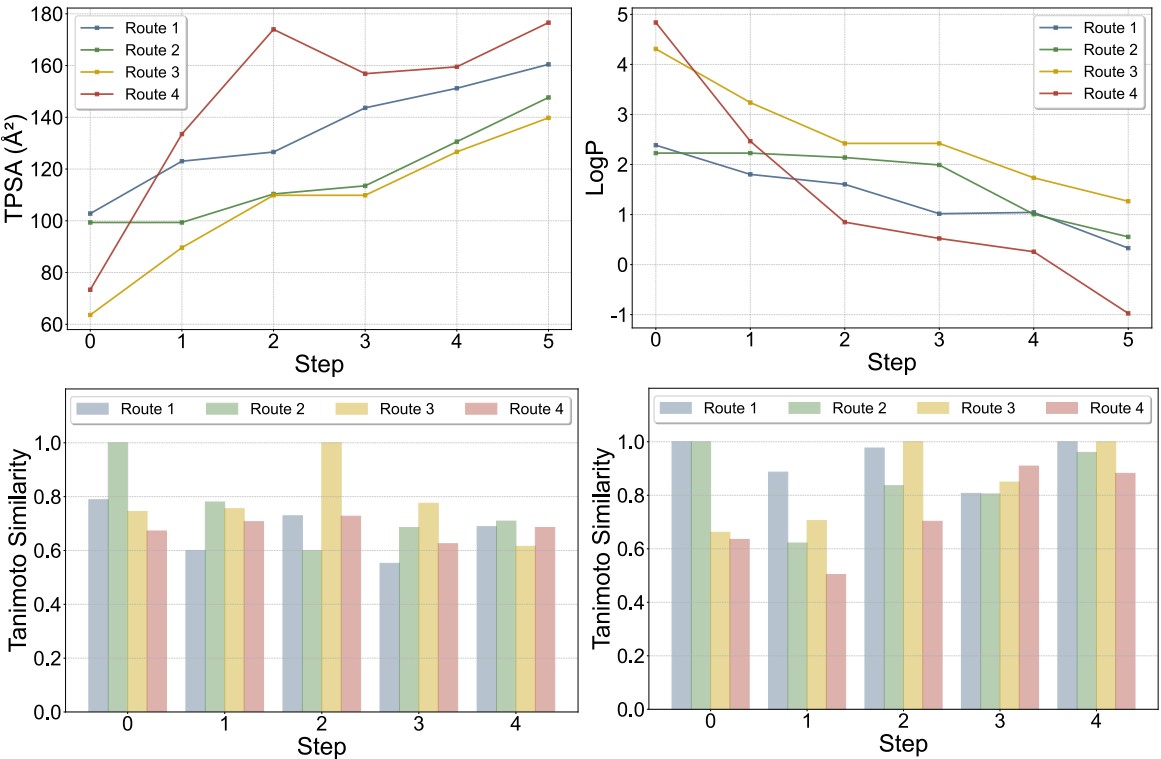

*Figure 21.* The edited molecules optimize (**Top**) water solubility while maintaining (**Bottom**) high scaffold and pharmacophore similarity, showing SpaRE's precision in the optimization.

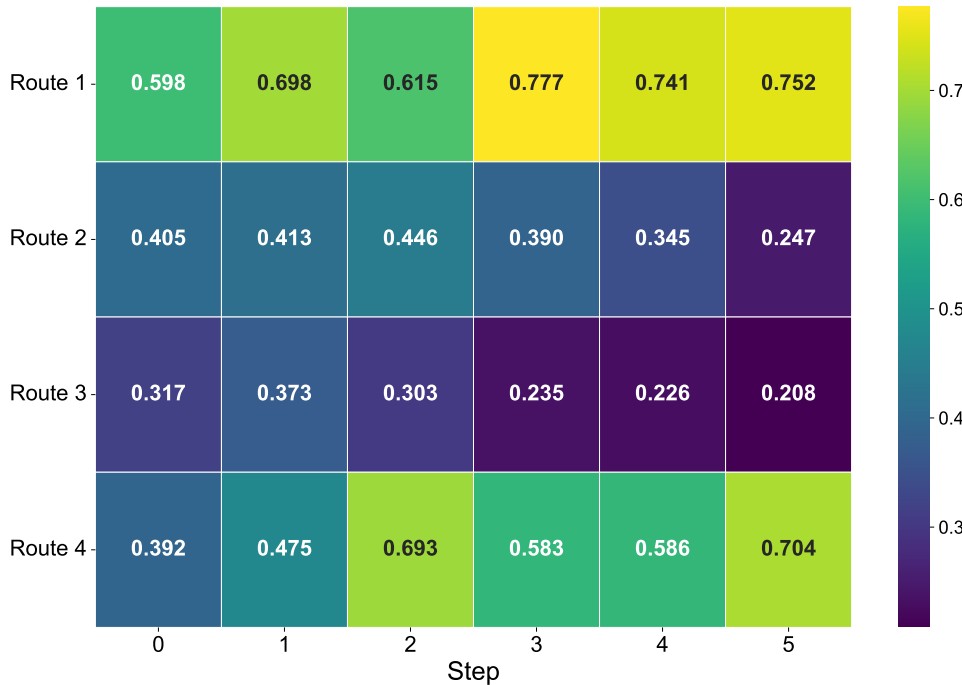

*Figure 22.* Quantitative estimate of drug-likeness for lipophilicity improvement. Two of the four routes achieve favorable drug-like properties (*i.e.,* QED $\geq$ 0.6).

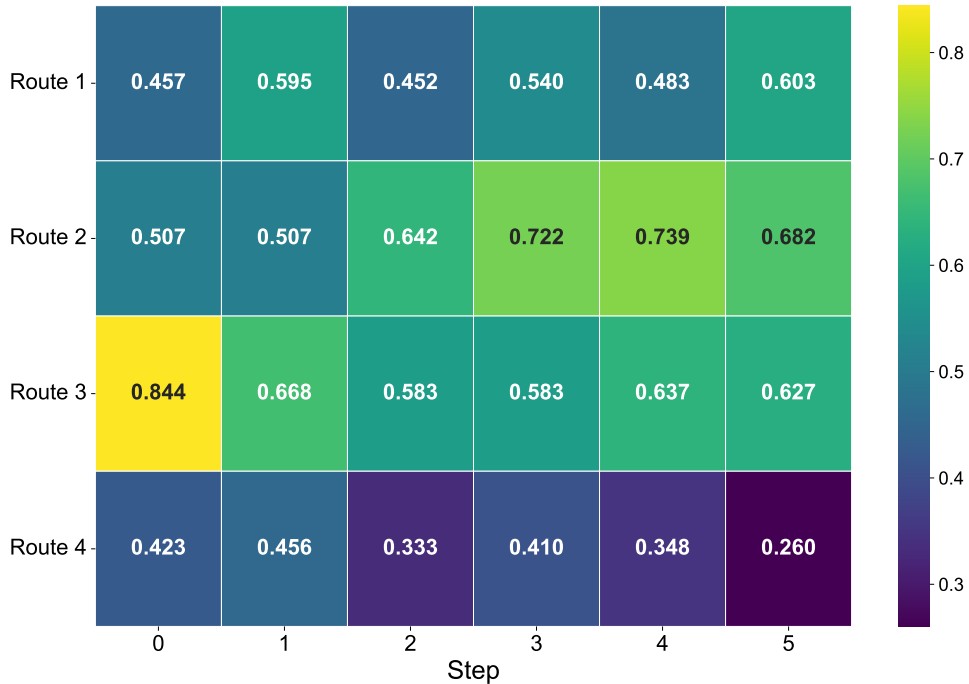

*Figure 23.* Quantitative estimate of drug-likeness for water solubility improvement. Three of the four routes achieve favorable drug-like properties (*i.e.,* QED $\geq$ 0.6).

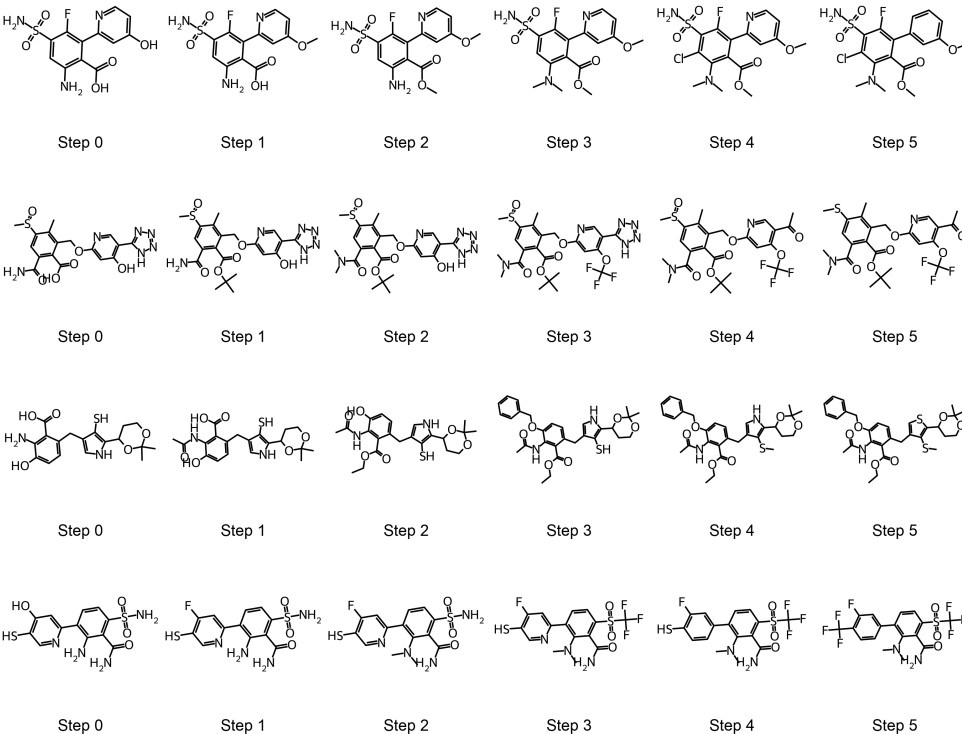

*Figure 24.* Visualization of four stepwise bioisosteric editing routes for lipophilicity improvement (Routes 1–4, top to bottom).

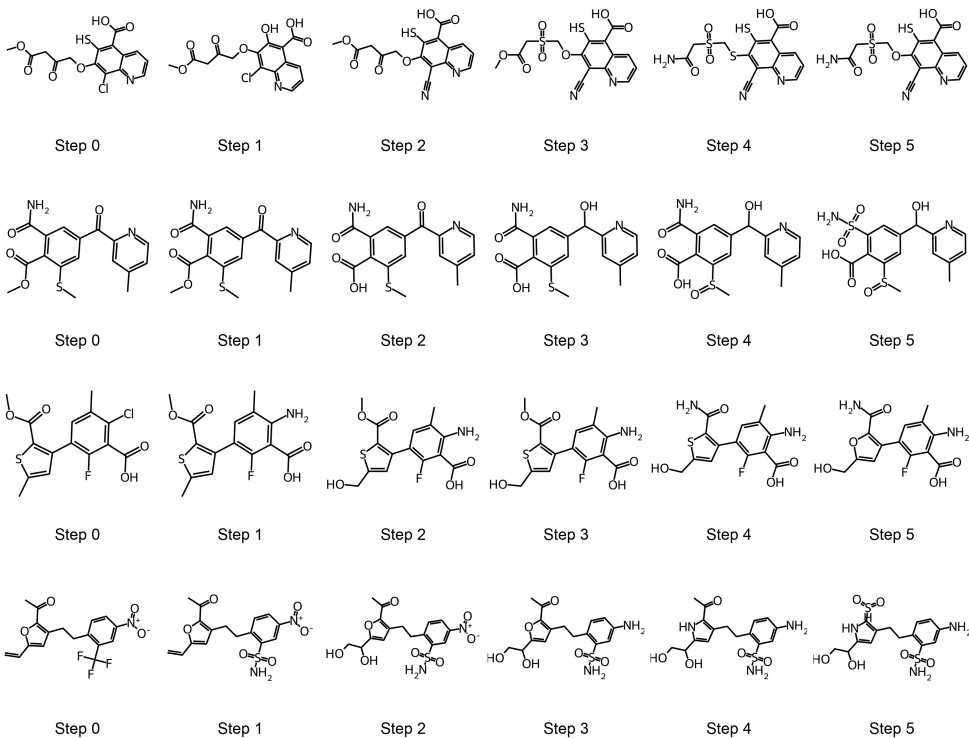

*Figure 25.* Visualization of four stepwise bioisosteric editing routes for water solubility improvement (Routes 1–4, top to bottom).

# I. Complex Editing Routes Planning

We report per-step SA and RA in Table 11 and Table 12. We transform source–target pairs via MCTS with similarity-based rewards, exploring edit routes under chemical priors (validity, SA Score, Tanimoto similarity) until RDKit graph isomorphism confirms an exact match. SpaRE consistently finds per-step synthesizable routes: SA stays manageable; RA is high, dips on hard edits, then recovers, indicating robust transformations. Most targets are reached in 3–5 steps, showing efficient search. Enforcing per-step constraints enables feasible edits and accelerates route discovery from accessible starting molecules. Following Liu et al. (2024); Cheng et al. (2024), we design five groups of edits as follows:

- **Group 1:** Skeletal editing of arenes and heteroarenes,

- **Groups 2 & 3:** Skeletal editing of indoles with fluoroalkyl N-triftosylhydrazones,

- **Groups 4 & 5:** Peripheral editing of indoles with fluoroalkyl N-triftosylhydrazones.

*Table 11.* SA score for 15 representative molecular editing routes. The ID is formatted as "GroupID_RouteID" (*i.e.,* route #1 from group 1 is denoted as 1_1).

| ID | Original | Step 1 | Step 2 | Step 3 | Step 4 | Step 5 | Step 6 | Step 7 | Step 8 | Avg. |
|----|----------|--------|--------|--------|--------|--------|--------|--------|--------|------|
| 1_1 | 2.18 | 1.55 | 2.36 | 3.40 | 1.70 | – | – | – | – | 2.24 |
| 1_2 | 2.05 | 1.60 | 2.21 | 3.35 | 3.80 | 1.96 | – | – | – | 2.50 |
| 1_3 | 1.66 | 1.29 | 1.85 | 3.07 | 1.53 | – | – | – | – | 1.88 |
| 2_1 | 2.02 | 2.20 | 2.17 | 2.86 | 3.61 | 3.42 | – | – | – | 2.71 |
| 2_2 | 2.02 | 2.20 | 2.17 | 2.86 | 3.61 | 3.74 | 3.55 | 3.55 | 3.60 | 3.03 |
| 2_3 | 2.02 | 2.20 | 2.17 | 2.86 | 3.61 | 3.72 | 3.90 | 4.25 | 3.66 | 3.15 |
| 3_1 | 2.02 | 2.20 | 2.17 | 3.12 | 3.17 | – | – | – | – | 2.53 |
| 3_2 | 1.87 | 2.06 | 2.04 | 3.02 | 3.40 | 4.05 | 4.21 | 4.34 | 3.31 | 3.14 |
| 3_3 | 2.02 | 2.20 | 2.17 | 3.12 | 3.49 | 3.60 | 3.22 | 3.35 | – | 2.90 |
| 4_1 | 1.74 | 2.16 | 2.55 | 2.42 | 2.23 | 2.29 | – | – | – | 2.23 |
| 4_2 | 1.74 | 2.07 | 2.47 | 2.81 | 2.68 | 2.43 | – | – | – | 2.37 |
| 4_3 | 1.74 | 2.16 | 2.55 | 2.63 | 2.39 | – | – | – | – | 2.29 |
| 5_1 | 1.74 | 3.03 | 3.67 | 3.59 | – | – | – | – | – | 3.01 |
| 5_2 | 1.76 | 2.76 | 3.47 | 3.54 | – | – | – | – | – | 2.88 |
| 5_3 | 1.87 | 3.42 | 3.89 | 3.73 | – | – | – | – | – | 3.23 |

*Table 12.* RA score for 15 representative molecular editing routes. The ID is formatted as "GroupID_RouteID" (*i.e.,* route #1 from group 1 is denoted as 1_1).

| ID | Original | Step 1 | Step 2 | Step 3 | Step 4 | Step 5 | Step 6 | Step 7 | Step 8 | Avg. |
|----|----------|--------|--------|--------|--------|--------|--------|--------|--------|------|
| 1_1 | 0.99 | 0.99 | 0.96 | 0.54 | 0.98 | – | – | – | – | 0.89 |
| 1_2 | 0.99 | 0.99 | 0.91 | 0.76 | 0.45 | 0.99 | – | – | – | 0.85 |
| 1_3 | 0.99 | 0.99 | 0.98 | 0.78 | 0.97 | – | – | – | – | 0.94 |
| 2_1 | 0.99 | 0.99 | 0.99 | 0.95 | 0.92 | 0.60 | – | – | – | 0.91 |
| 2_2 | 0.99 | 0.99 | 0.99 | 0.95 | 0.92 | 0.88 | 0.40 | 0.40 | 0.37 | 0.76 |
| 2_3 | 0.99 | 0.99 | 0.99 | 0.95 | 0.92 | 0.71 | 0.28 | 0.05 | 0.26 | 0.68 |
| 3_1 | 0.99 | 0.99 | 0.99 | 0.94 | 0.78 | – | – | – | – | 0.94 |
| 3_2 | 0.99 | 0.99 | 0.99 | 0.95 | 0.90 | 0.48 | 0.34 | 0.13 | 0.62 | 0.71 |
| 3_3 | 0.99 | 0.99 | 0.99 | 0.94 | 0.88 | 0.81 | 0.76 | 0.65 | – | 0.88 |
| 4_1 | 0.98 | 0.99 | 0.99 | 0.96 | 0.97 | 0.98 | – | – | – | 0.98 |
| 4_2 | 0.98 | 0.99 | 0.99 | 0.99 | 0.97 | 0.98 | – | – | – | 0.98 |
| 4_3 | 0.98 | 0.99 | 0.99 | 0.95 | 0.96 | – | – | – | – | 0.97 |
| 5_1 | 0.99 | 0.39 | 0.16 | 0.54 | – | – | – | – | – | 0.52 |
| 5_2 | 0.99 | 0.78 | 0.39 | 0.25 | – | – | – | – | – | 0.60 |
| 5_3 | 0.99 | 0.84 | 0.34 | 0.43 | – | – | – | – | – | 0.65 |

## J. Molecular Editing Routes Optimization

Other than water solubility (physicochemical), we optimize the structural properties, specifically the count of rotatable bonds and $sp^3$ carbon atoms. In particular, increasing the number of rotatable bonds enhances molecular flexibility, which can facilitate binding to biological targets. A higher fraction of $sp^3$ carbons increases molecular three-dimensionality, thus improving pharmacokinetic properties and drug-likeness. Optimizing these characteristics facilitates the design of molecules that are not only soluble but also possess favorable physicochemical properties for medical applications. The distributions of generated molecules with respect to improvements in rotatable bonds and $sp^3$ carbon atoms are shown in Figure 26 and Figure 27, respectively. Visualizations of representative generated molecules optimized for water solubility, rotatable bonds, and $sp^3$ carbon atoms are shown in Figure 28, Figure 29, and Figure 30, respectively.

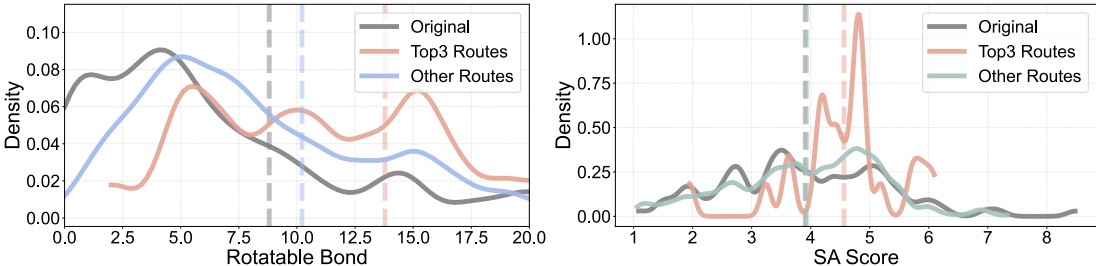

*Figure 26.* The distribution of generated molecules following editing route optimization to increase the number of rotatable bonds.

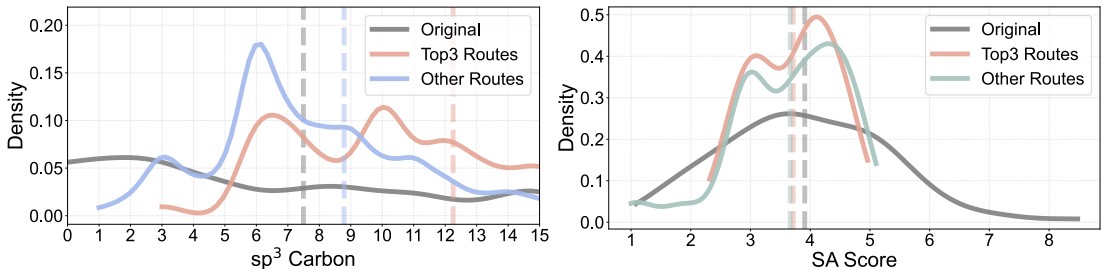

*Figure 27.* The distribution of generated molecules following editing route optimization to increase the number of $sp^3$ carbon atoms.

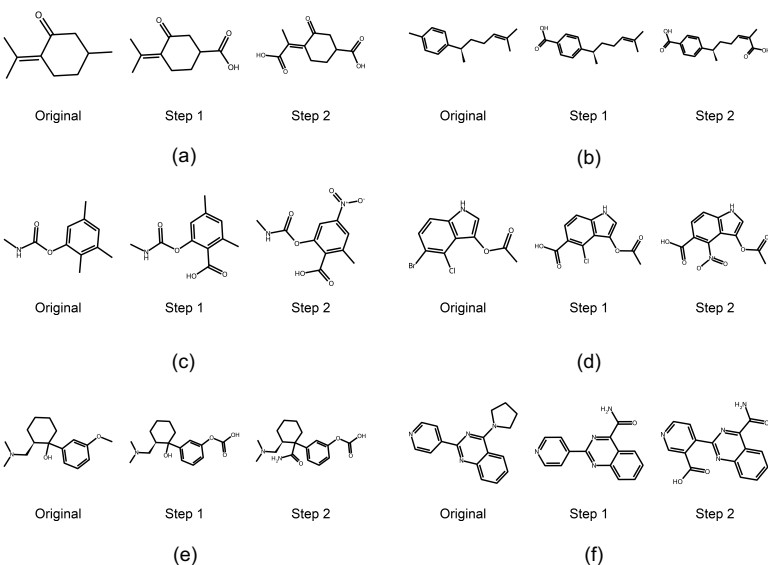

*Figure 28.* Visualization of molecules with favorable synthetic accessibility and improved water solubility. Introducing carboxylate anion, nitro group, and primary amide at specific sites leads to improved water solubility.

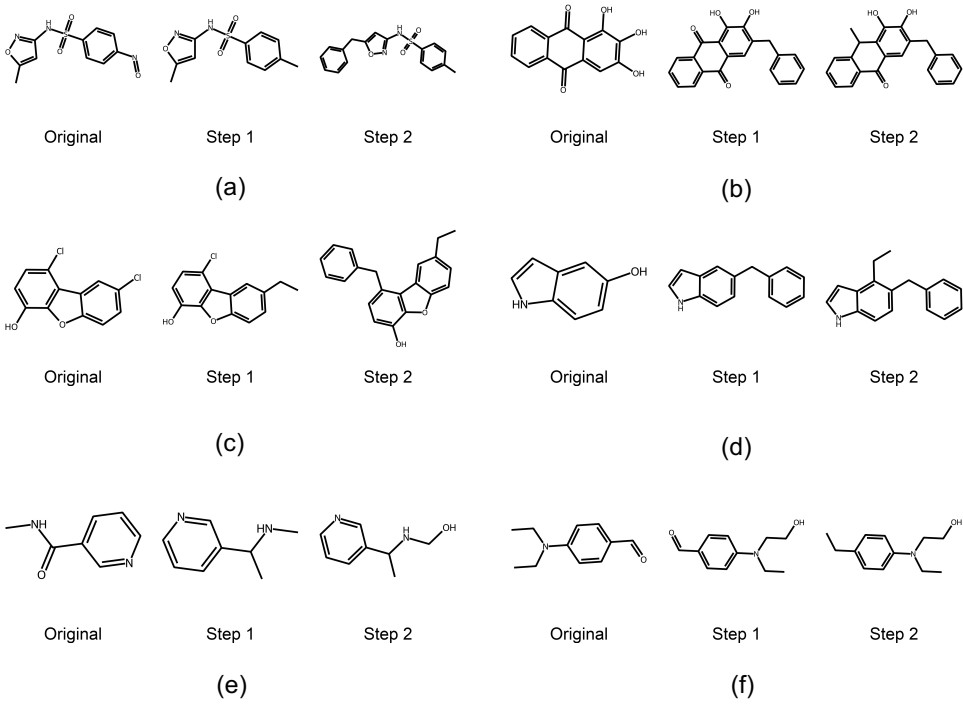

*Figure 29.* Visualization of molecules with favorable synthetic accessibility and increased rotatable bonds. Introducing benzyl, methyl, or ethyl groups at specific sites increases the number of rotatable bonds.

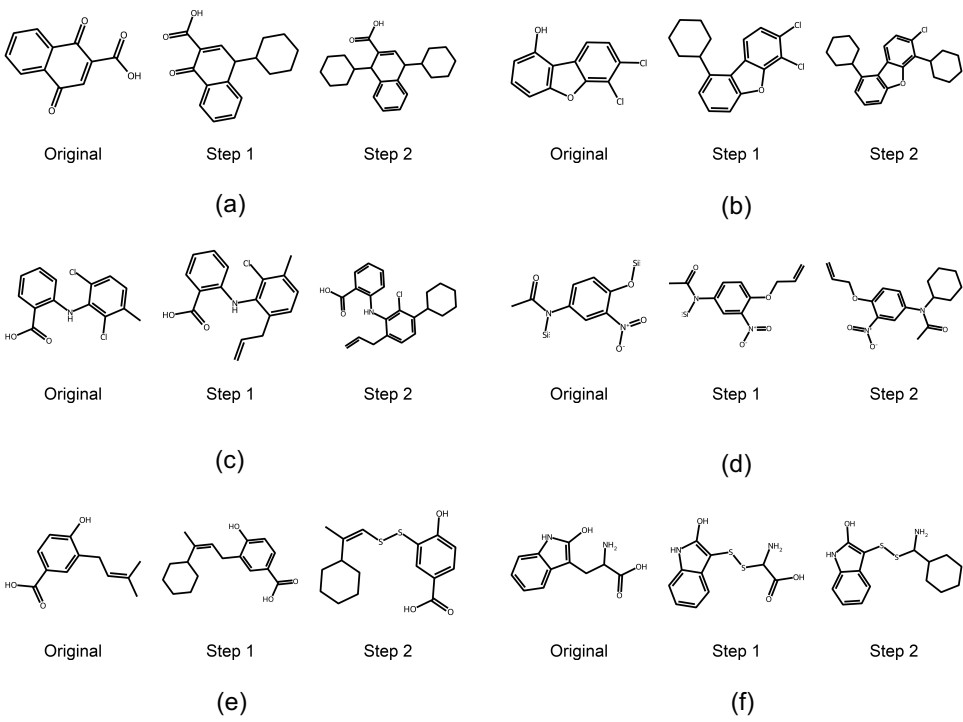

*Figure 30.* Visualization of molecules with favorable synthetic accessibility and increased sp$^3$ carbon count. Introducing a cyclohexyl group, an allyl-cyclohexyl group, or disulfide linkages at specific sites increases the number of sp$^3$ carbon atoms.

# K. Multi-Target Drug Discovery

To enable controllable drug discovery of dual kinase inhibitors, we combine SpaRE with MCTS for guided exploration of the molecular space. The optimization task is defined by four constraints: GSK3$\beta$ activity $\geq 0.5$, JNK3 activity $\geq 0.5$, QED $> 0.6$, and SA score $< 4$. Technically, SpaRE generates chemically valid molecules and refines them at each step, guided by the current molecular state and property targets. MCTS organizes the search process by treating each molecule as a node and each feasible edit as a possible action. At each iteration, the search tree is expanded by evaluating candidate edits according to a composite reward function that reflects both property improvements and constraint satisfaction. Activity changes for both GSK3$\beta$ and JNK3 are assigned a reward of $+1$ or $-1$ for incremental improvements or degradations, with a $+3$ bonus when crossing the final $0.5$ threshold. QED receives $+2$ if above $0.6$, otherwise $+1$ or $-1$ depending on the direction of change; SA score is similarly rewarded, with $+2$ for values below $4$. To encourage consistent progress, a momentum bonus is applied for consecutive property improvements, and a minor penalty is imposed for direction reversals. The MCTS framework is tuned for effective exploration, employing $1000$ optimization iterations, a maximum edit depth of $50$, and progressive widening to balance breadth and depth. Node expansion is triggered after $5$ visits, while virtual loss and $\varepsilon$-greedy strategies ensure parallelism and exploratory diversity. Selection of edits follows an upper confidence bound (UCB) policy (Auer, 2002), which incorporates a bonus to prioritize promising directions. During each iteration, SpaRE generates a set of candidate molecules through targeted and property-driven edits. The resulting molecules are evaluated against the kinase activity, drug-likeness, and synthetic accessibility constraints. Rewards are computed and back-propagated through the search tree, dynamically guiding the search toward regions of chemical space with the highest likelihood of yielding constraint-satisfying molecules. The procedure iterates until a molecule fulfilling all criteria is discovered or the search budget is exhausted. The integration of SpaRE's controllable generation with MCTS-based search enables efficient navigation of the molecular landscape for multi-target drug design and produces potent kinase inhibitors that simultaneously satisfy complex property requirements.

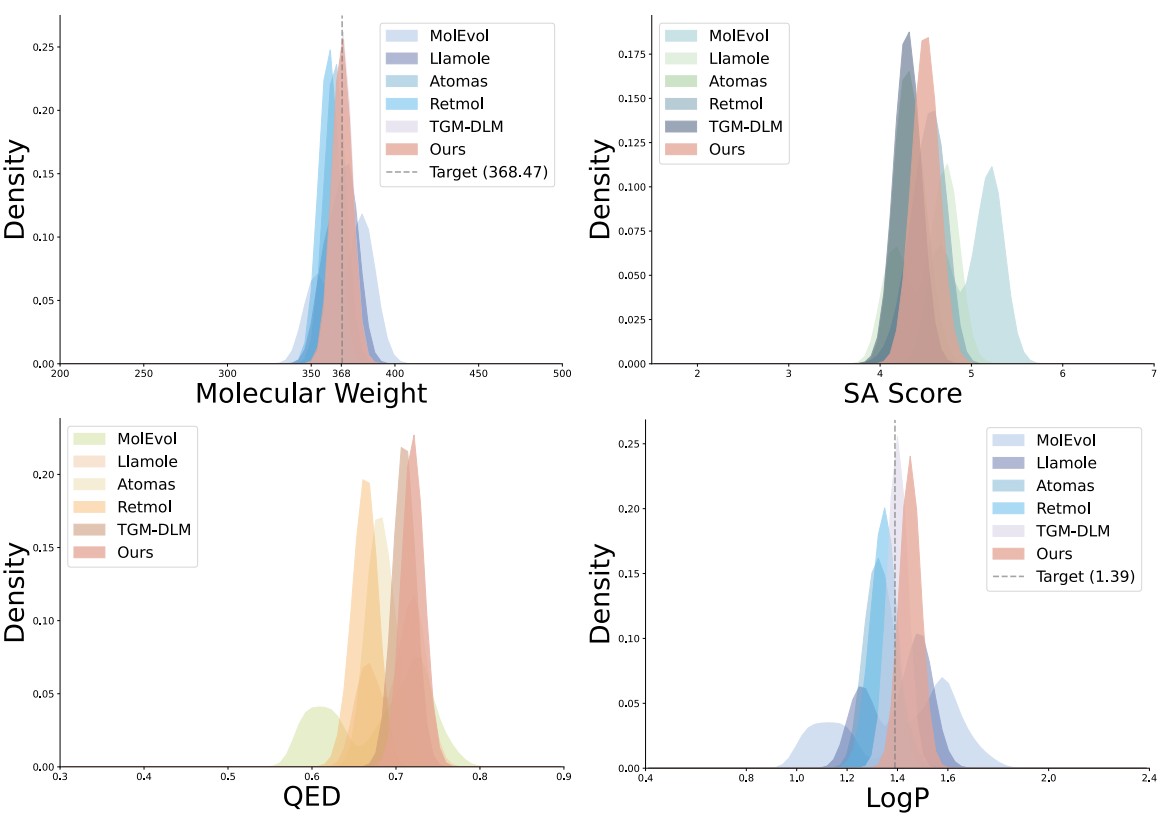

*Figure 31.* Distribution of generated molecules for Perindopril. The hard constraints on target molecular weight and LogP are precisely satisfied, while the soft constraints, QED and SA score, are optimized more effectively than baseline models.

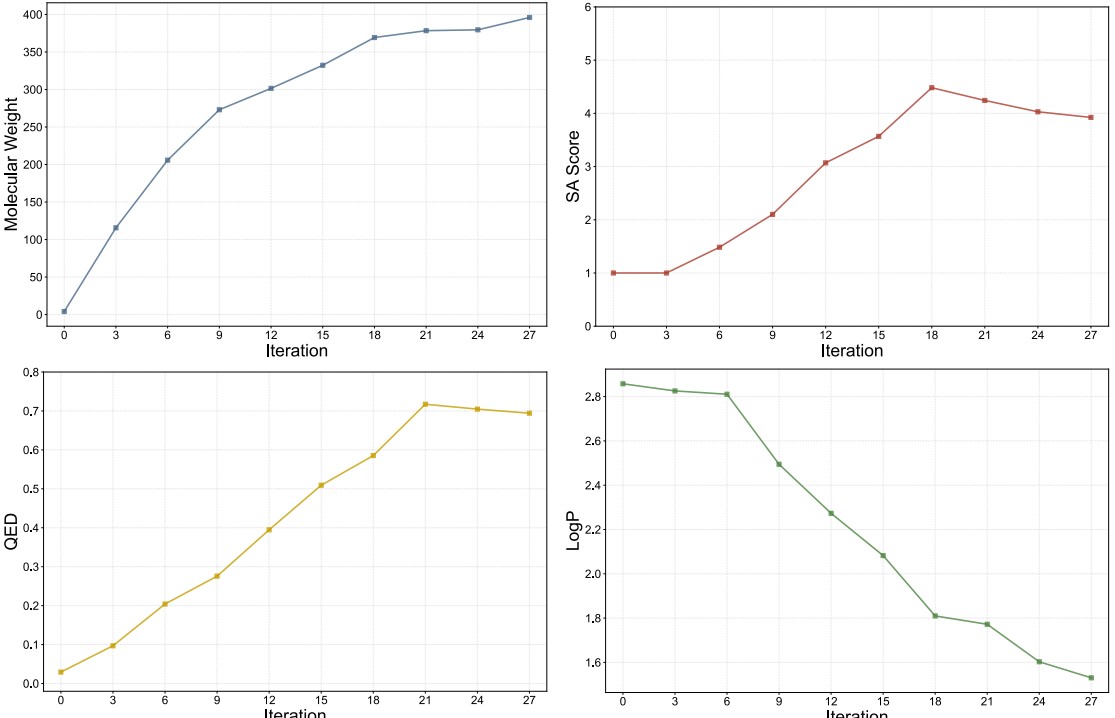

*Figure 32.* Iterative convergence of molecular properties during the optimization process for Perindopril.

For the Perindopril optimization task, we applied SpaRE and MCTS to generate molecules meeting property requirements: target MW of $368.47\pm0.5$ Da, target LogP of $1.39$, maximized QED, and minimized SA score (Todd & Fitton, 1991). In this study, LogP and MW are required to be as close as possible to their target values (which we refer to as "hard constraints") to maintain specific drug-like properties. At each step, SpaRE performs chemically valid edits based on the current molecular state and property objectives, while MCTS explores alternative editing routes guided by a carefully designed reward function. Specifically, the reward assigns $+2$ if MW is within the target range (otherwise $+1$ or $-1$ depending on distance), LogP is rewarded proportionally to its proximity to $1.39$, QED gains $+1$ for increases and $-0.5$ for decreases, and SA score is rewarded for reductions. Momentum bonuses are added for consecutive improvements, encouraging efficient convergence. As illustrated in Figure 31, our method generates molecules with MW and LogP distributions sharply centered around the target values, and outperforms all baselines. Notably, SpaRE achieves both higher QED and lower SA score (In this case, these are "soft constraints," meaning there are no explicit target values.), indicating good drug-likeness and synthetic accessibility. The optimization trajectory (Figure 32) further demonstrates that key properties rapidly converge toward their targets within a limited number of iterations. QED consistently increases while SA score decreases, and MW and LogP quickly stabilize around the desired values. These results highlight the effectiveness of our controllable editing and reward-driven search in generating high-quality Perindopril analogs and balancing multiple objectives more accurately than existing approaches.

For the Aripiprazole optimization task, the model generates molecules that satisfy the following constraints: target MW of $448.39 \pm 0.5$ Da, target LogP of $4.98$, maximized QED, and minimized SA score. At each optimization iteration, SpaRE proposes valid molecular edits based on the current state and target properties, while MCTS explores editing paths using a reward scheme analogous to the Perindopril task. The reward assigns $+2$ if MW falls within the target range, LogP is rewarded based on its proximity to $4.98$, QED is encouraged to increase, and SA score is encouraged to decrease, with additional momentum bonuses for consecutive improvements. As shown in Figure 33, our method produces molecules whose MW and LogP distributions are centered around the desired values, surpassing other baselines. SpaRE achieves higher QED and lower SA score for the generated molecules and highlights improvements in both drug-likeness and synthetic accessibility. Property optimization traces (Figure 34) show that MW and LogP rapidly converge to their targets, while QED steadily improves and SA score decreases throughout the process. These findings demonstrate that our controllable generation method facilitates efficient multi-target drug discovery and yields compound analogs exhibiting a favorable balance of pharmaceutical properties.

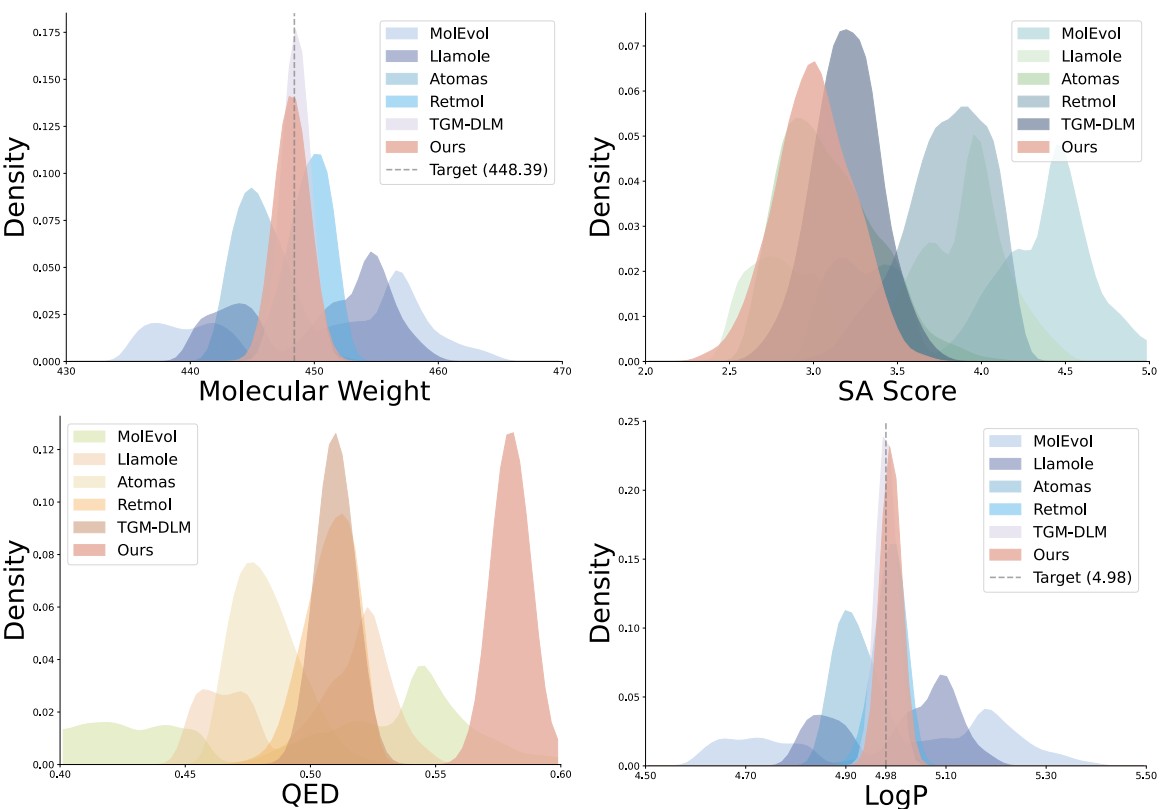

*Figure 33.* Distribution of generated molecules for Aripiprazole. The hard constraints on target molecular weight and LogP are precisely satisfied, while the soft constraints, QED and SA score, are optimized more effectively than baseline models.

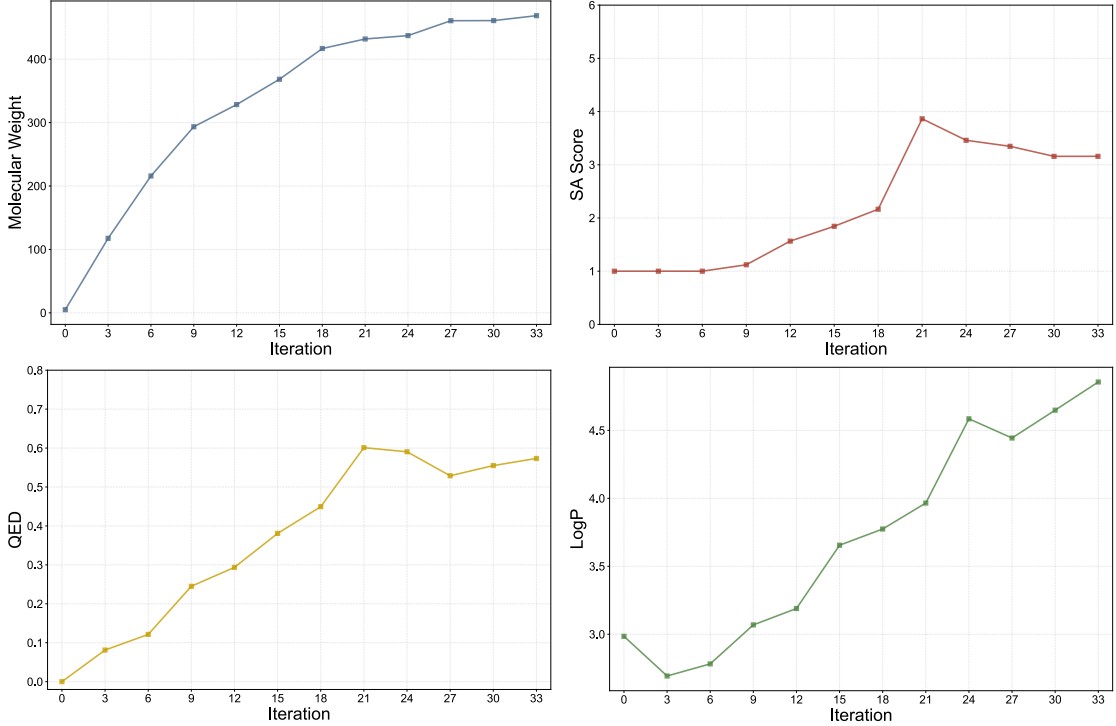

*Figure 34.* Iterative convergence of molecular properties during the optimization process for Aripiprazole.

# L. Complex Constraint Optimization for Drug Design

To comprehensively validate the effectiveness of our method, we evaluate it on property optimization tasks for oral drugs, drugs satisfying Lipinski's Rule of Five (Lipinski, 2004), and sublingual drugs. These representative tasks are chosen because they address critical challenges in drug discovery. Oral drugs and sublingual drugs each demand distinct physicochemical properties related to their routes of administration, while Lipinski's Rule of Five encapsulates widely accepted guidelines for drug-likeness and bioavailability. By optimizing for these different tasks, we can test whether our approach can generate molecules that meet complex property constraints. The distributions of generated molecules for each task are shown in Figure 35 (along with Figure 5), Figure 36, and Figure 38, respectively. The convergence of molecules during optimization iterations is illustrated in Figure 5, Figure 37, and Figure 39. Further details are provided below.

To optimize molecules with desirable oral drug properties, we integrate SpaRE with MCTS. Specifically, the oral drug constraints include $1 \leq \text{LogP} \leq 3$, $MW < 500$ Da, $\text{TPSA} \leq 140$ Å$^2$, $\text{HBD} \leq 5$, $\text{HBA} \leq 10$, aromatic rings $\leq 3$, SA score $< 5$, and QED $> 0.7$. At each step, SpaRE receives the current molecule state and the targeted property improvements, and generates valid molecules that are likely to move it toward satisfying these constraints. MCTS is employed to efficiently explore the space of possible edits where each state corresponds to a molecular structure, and each action is a feasible edit. The reward function assigns $+1$ when an edit moves a property closer to its constraint, $-1$ when it moves away, and $+2$ for satisfying a constraint, while continuous properties use distance-based rewards and discrete properties reward reductions when above limits. The total reward is calculated as $R = \sum_i w_i \times r_i$, where $w_i = 1.0$ for hard constraints (*e.g.,* LogP, TPSA) and $w_i = 0.5$ for soft constraints (*e.g.,* SA score, QED), with an additional $+5$ bonus for satisfying all constraints and a $-0.5$ penalty per consecutive violation. MCTS is configured with 500 iterations per molecule, a maximum depth of 20 edits, UCB exploration constant $c = 1.414$, rollout depth of 5 steps, node expansion after 10 visits, batch evaluation of 32 molecules, and early stop when all constraints are met. The optimization proceeds as follows: starting from a given molecule, at each iteration, SpaRE performs property-driven edits, properties are evaluated, rewards are computed, and promising nodes are selected and expanded via UCB. Simulations are conducted for value estimation, and rewards are back-propagated. The process terminates when either the maximum number of edits is reached or a molecule meeting all constraints is found. Overall, integrating controllable molecule generation with MCTS enables fine-grained and efficient generation and optimization of oral drug candidates.

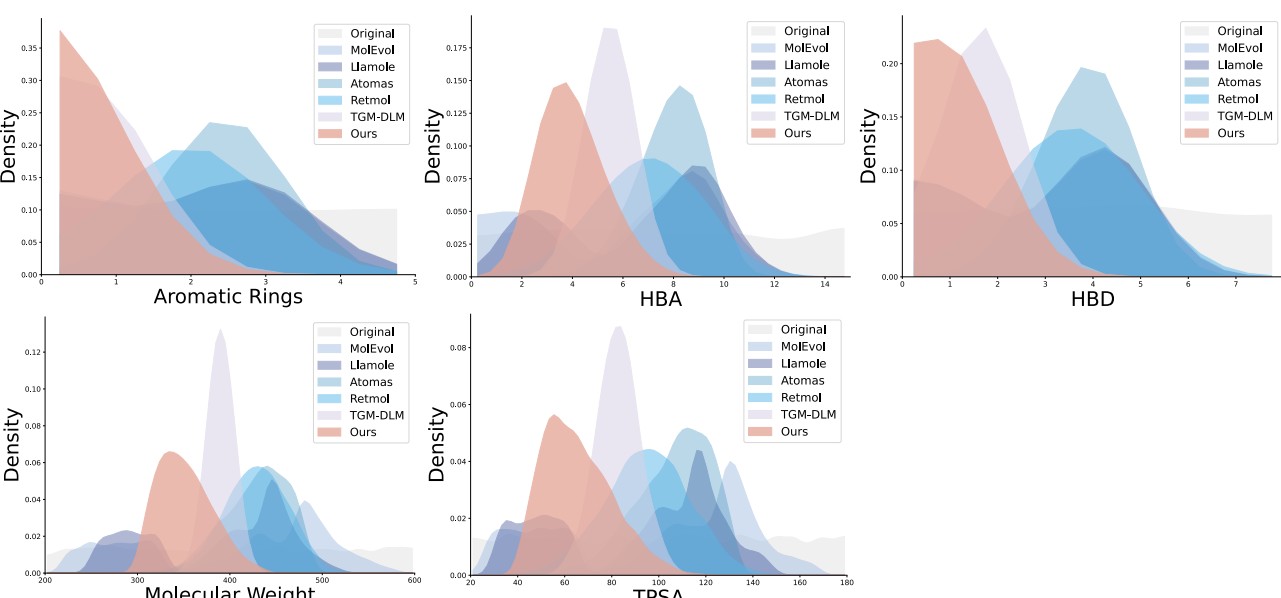

*Figure 35.* Distribution of generated molecules for oral drug optimization, showing that five property constraints (with the other three illustrated in Figure 5) are precisely optimized within their defined ranges, with SpaRE surpassing all baselines.

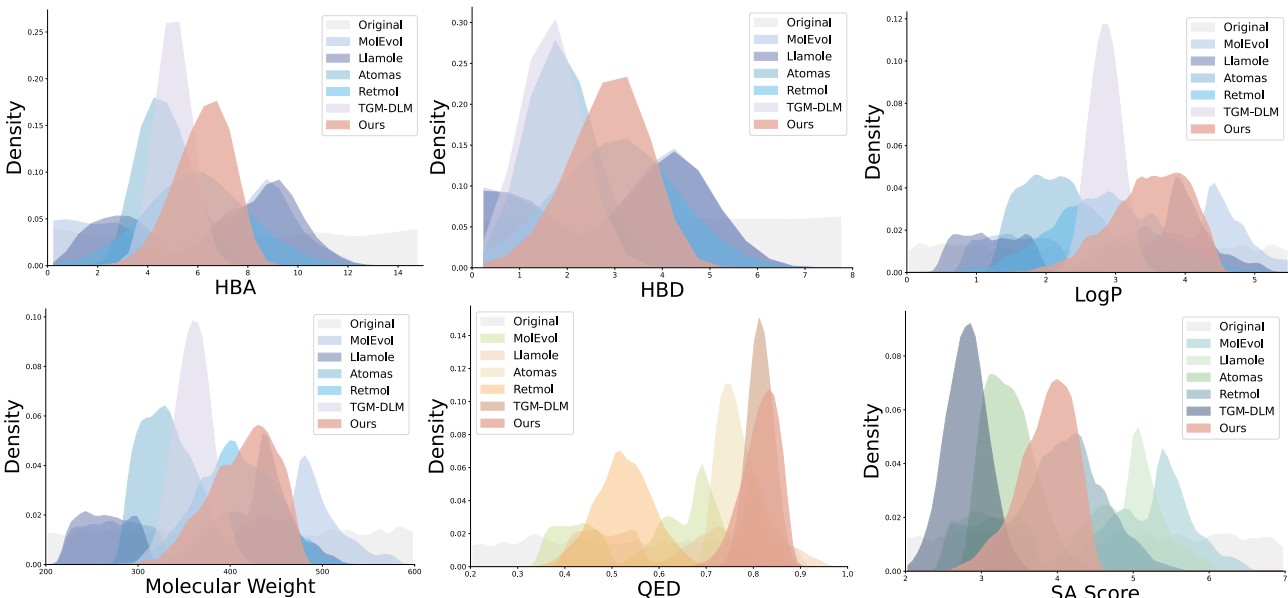

*Figure 36.* Distribution of generated molecules under Lipinski's Rule of Five, showing that six property constraints are precisely optimized within their defined ranges, with SpaRE surpassing all baselines.

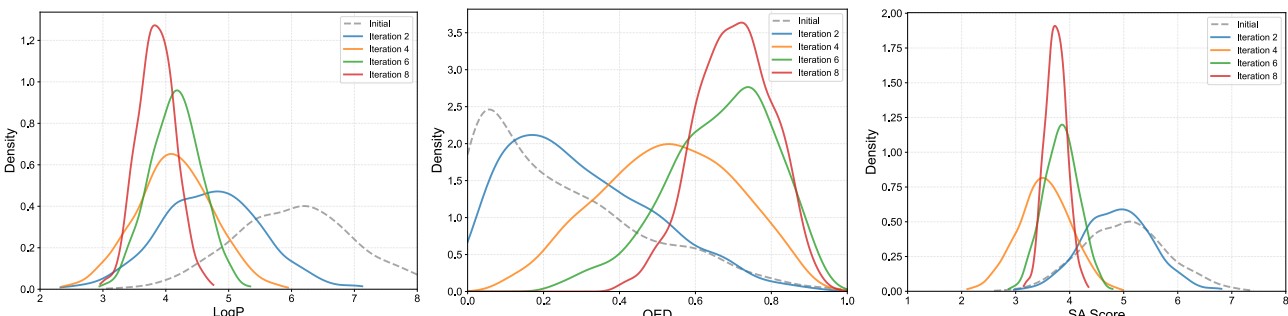

*Figure 37.* Iterative convergence of molecular properties during the optimization process under Lipinski's Rule of Five.

We further demonstrate the generalizability of our controllable molecule generation framework by targeting two representative drug-like property profiles. Specifically, we consider (1) Lipinski's Rule of Five requires molecules to satisfy HBA $\leq 10$, HBD $\leq 5$, MW $\leq 500$ Da, and LogP $\leq 5$, together with two additional practical objectives: SA score $< 5$ and QED $> 0.7$; and (2) sublingual drug requirements, which include $2 \leq$ LogP $\leq 5$, MW $< 400$ Da, TPSA $\leq 90$ Å$^2$, HBD $\leq 3$, HBA $\leq 6$, aromatic rings $< 2$, SA score $< 5$ and QED $> 0.7$. In both settings, MCTS explores the molecular space by applying property-driven edits, with a reward function that integrates all relevant criteria. This approach facilitates the discovery of candidate molecules that simultaneously satisfy multiple requirements. As shown in Figure 36 and Figure 37, our method outperforms baseline approaches in Lipinski's Rule of Five task, achieving more concentrated distributions for HBA, HBD, MW, and LogP within the target ranges, as well as higher QED and lower SA scores. The convergence curves further validate that our approach efficiently guides the optimization process within a few steps, with all properties rapidly shifting toward the desired ranges. Similarly, for the sublingual drug task (Figure 38, Figure 39), our method generates molecules that better match the requirements on aromatic rings, HBA, HBD, LogP, TPSA, and MW compared to baselines. The property distributions of the generated molecules are more tightly centered within the desired ranges, and our approach achieves notable improvements in both QED and SA score. During optimization iterations, property constraints converge quickly toward their targets, proving the efficiency of our strategy in tackling drug optimization tasks.

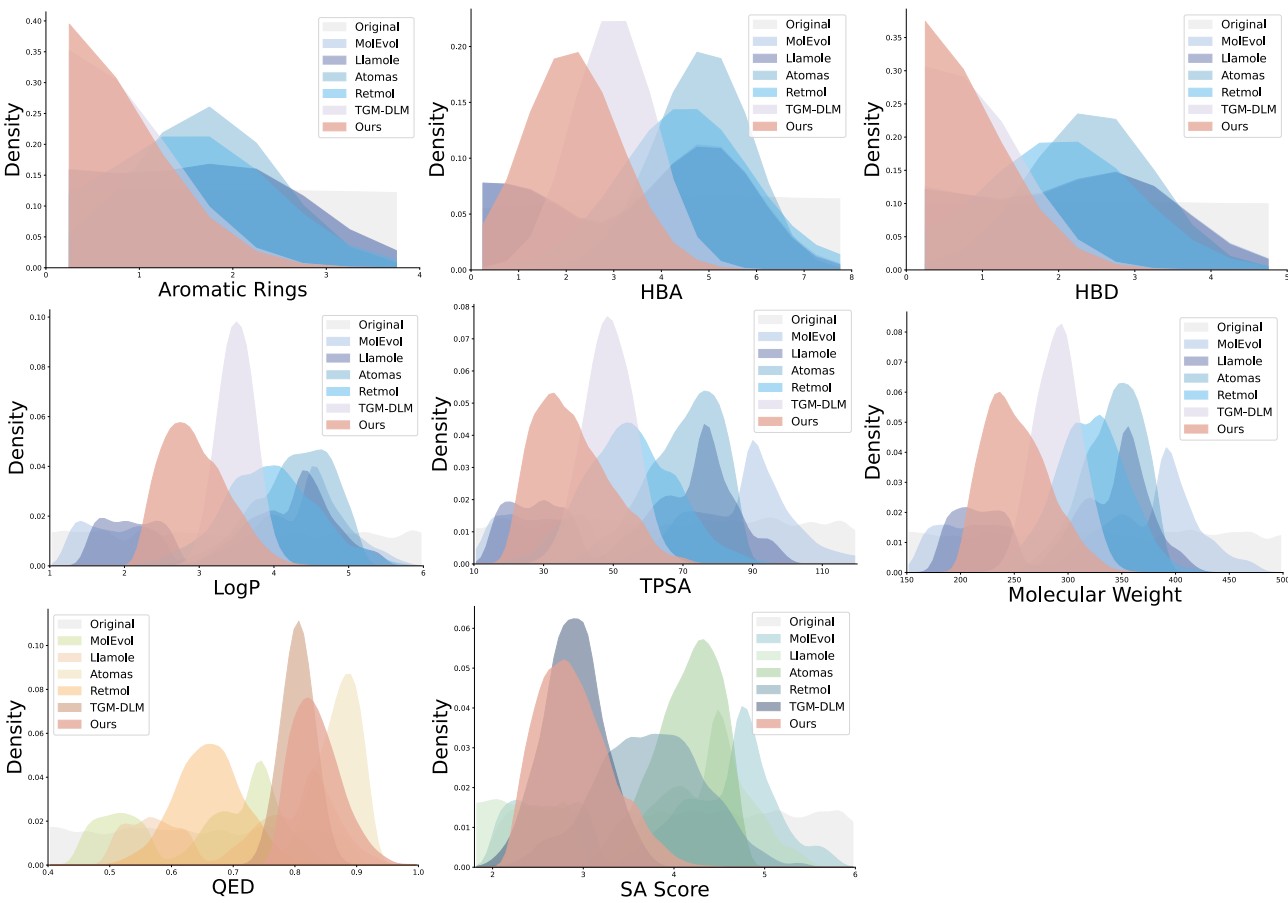

*Figure 38.* Distribution of generated molecules for sublingual drug optimization, showing that eight property constraints are steered toward their target ranges, with SpaRE outperforming all baselines.

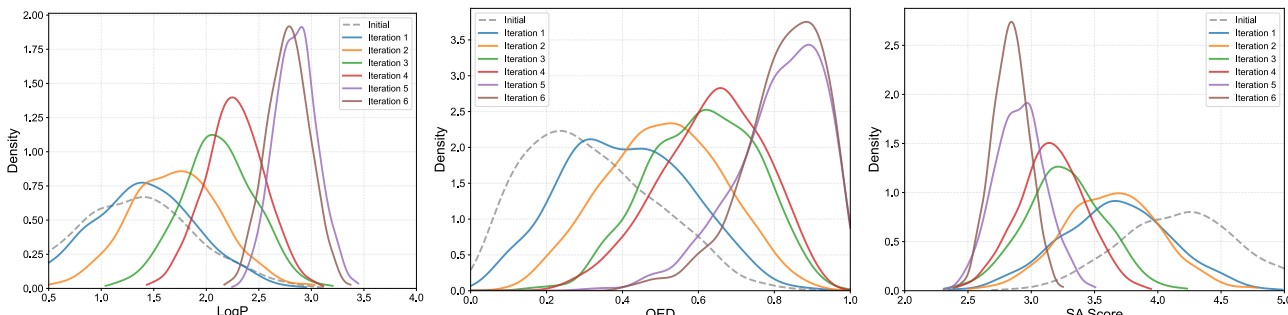

*Figure 39.* Iterative convergence of molecular properties during the optimization process for sublingual drug design.

## M. Additional Results on Constrained Multi-Objective Molecule Generation

In this section, we report detailed results for multi-objective molecule generation, including oral drug design, sublingual drug optimization, and multi-target drug discovery. We evaluate generation quality, constraint satisfaction rate, and computational efficiency, with results in Table 13, Table 14, and Table 15. As shown in these tables, SpaRE consistently achieves the best constraint satisfaction rate and generation quality, while maintaining comparable computational efficiency.

*Table 13.* Additional multi-target drug discovery results on constraint satisfaction rate (%, ↑) and generation quality (%, ↑). **Best** and second-best results are marked in bold and underlined, respectively.

| Method | Constraint Satisfaction Rate | | | | | Generation Quality | | | |
| --- | --- | --- | --- | --- | --- | --- | --- | --- | --- |
| | GSK3$\beta \geq 0.5$ | JNK3$\geq 0.5$ | QED$> 0.6$ | SA$< 4$ | All | Success | Validity | Novelty | Diversity |
| TGM-DLM | 72.4 | 69.2 | 90.6 | 98.4 | 44.7 | 96.1 | 98.4 | 77.2 | 66.9 |
| MolEvol | 57.1 | 61.3 | 65.6 | 73.9 | 17.0 | 90.1 | 93.5 | 82.4 | 67.3 |
| RetMol | 74.1 | 86.6 | 92.3 | 93.3 | 55.3 | 95.9 | 97.2 | 87.6 | 72.6 |
| Atomas | 72.2 | 80.8 | 90.7 | 93.4 | 49.5 | 94.1 | 97.7 | 85.2 | 70.9 |
| Llamole | 70.3 | 73.7 | 89.0 | 93.5 | 43.1 | 88.2 | 98.3 | 91.1 | 67.1 |
| SpaRE (**Ours**) | **85.6** | **89.2** | **96.1** | **99.8** | **83.2** | **99.8** | **100.0** | **95.1** | **74.4** |

*Table 14.* Quantitative success rate (%, ↑) for oral drug design. **Best** and second-best results are indicated in bold and underlined, respectively.

| Method | LogP ([1,3]) | MW ($< 500$) | TPSA ($\leq 140$) | HBD ($\leq 5$) | HBA ($\leq 10$) | Arom. Rings ($\leq 3$) | SA ($< 5$) | QED ($> 0.7$) | All |
| --- | --- | --- | --- | --- | --- | --- | --- | --- | --- |
| Original | 57.2 | 74.8 | 75.0 | 63.9 | 68.1 | 60.5 | 60.6 | 32.8 | 1.9 |
| MolEvol | 65.4 | 84.1 | 88.4 | 91.3 | 92.1 | 79.0 | 62.1 | 22.6 | 1.6 |
| Llamole | 88.3 | 98.3 | 96.9 | 91.9 | 91.5 | 77.8 | 78.8 | 56.8 | 21.7 |
| Atomas | 90.3 | 97.6 | 95.8 | 96.3 | 96.0 | 99.1 | 91.5 | 98.4 | 67.4 |
| RetMol | 95.8 | 98.9 | **100.0** | 89.6 | 93.7 | 93.9 | 96.2 | 0.5 | 0.2 |
| TGM-DLM | **100.0** | 96.5 | 94.2 | 97.1 | 96.6 | 94.8 | 95.4 | 97.9 | 78.5 |
| SpaRE (**Ours**) | **100.0** | **100.0** | **100.0** | **100.0** | **100.0** | **100.0** | **100.0** | **100.0** | **100.0** |

*Table 15.* Runtime for each method (seconds per molecule satisfying the specified constraints, ↓). **Best** and second-best results are indicated in bold and underlined, respectively.

| Model | Oral Drug Design | Sublingual Drug Optimization | Multi-Target Drug Discovery |
| --- | --- | --- | --- |
| TGM-DLM | 4.43 | 7.07 | 17.84 |
| MolEvol | 2.40 | 3.61 | 8.98 |
| RetMol | 10.12 | 15.99 | 40.12 |
| Llamole | 43.11 | 64.02 | 161.77 |
| Atomas | **1.92** | **2.83** | 7.01 |
| SpaRE (**Ours**) | 2.18 | 3.50 | **6.67** |

# N. Interpretability Analysis

In this section, we present interpretations of the most significant features identified by linear probing in Figure 8, as well as the LLM-interpreted features in Table 17. The corresponding prompts are provided below. Additionally, Figure 40 presents the input–output activation patterns, highlighting the input words that most strongly activate the selected output token.

*Table 16.* Feature interpretations from the fitted linear model indicate that these features are statistically significant for the prediction of solubility.

| Feature ID | Interpretation |
|---|---|
| 1592 | Polarity |
| 7329 | Hydrophilic Groups |
| 38472 | Ionic Nature |
| 29104 | Nonpolar |
| 12945 | Amphiphilic |
| 23847 | Lattice Energy |
| 6758 | Hydrogen Bonds |
| 40123 | Crystal Structure |
| 15683 | Weakly Acidic |

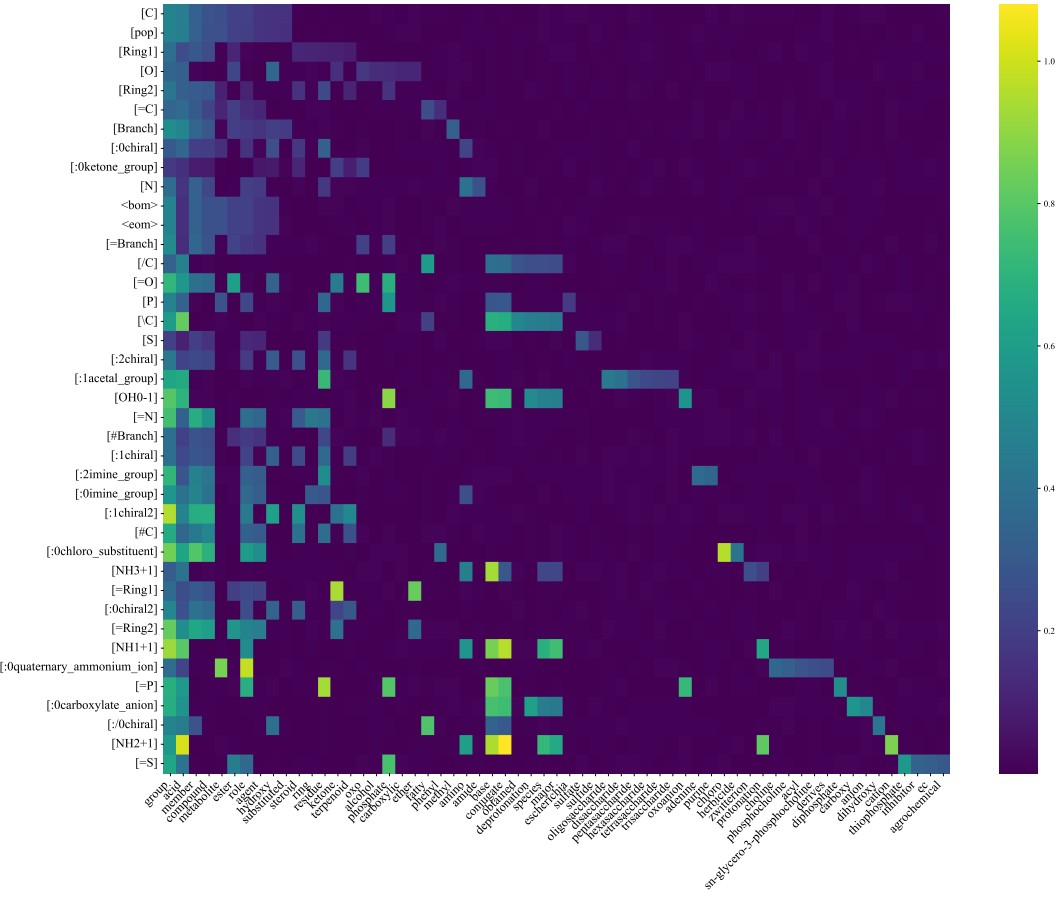

*Figure 40.* The activations for input–output patterns (from LLM layer 10), where the *X*- and *Y*-axes represent the input and output of the LLM, respectively. We focus on the most frequently generated output token and identify the input tokens that most strongly activate it.

We carefully design prompts for LLMs (specifically, Gemini 2.5 Pro (Comanici et al., 2025)) to generate interpretations of the activated latent features. By summarizing shared patterns in the molecules that most strongly activate these features, the LLM provides human-interpretable explanations of these latent features. The designed prompt is shown below.

---

### Prompt Used for LLM-Based Interpretations

**Objective**:
To interpret a specific feature from a text-to-molecule generation model by analyzing the text prompts that activate it most strongly.

**Context**:
I am analyzing the internal workings of a text-to-molecule generation model. This model takes a natural language text prompt and generates a corresponding molecular structure. I have identified a specific internal feature (*e.g.,* a neuron or a latent vector dimension) and have found the top 500 text prompts that cause the highest activation for this feature. My goal is to understand what chemical concept this feature has learned to encode. Your role is to act as an expert in medicinal chemistry and cheminformatics. Please analyze the semantic content of these text prompts to hypothesize the precise chemical property, structural motif, or therapeutic concept that this feature represents.

**Input Data**:
Below are the top 500 activating text prompts for this feature, ranked from highest to lowest activation.

**Your Task & Required Output Format**:
Analyze Semantic Commonalities: Carefully read all prompts. Identify the common keywords, phrases, and underlying chemical or biological concepts being described. Look for patterns related to molecular structure, function, physical properties, or therapeutic use.

Hypothesize Feature Identity: Based on your analysis, propose a concise name and a clear, one-sentence description for the concept this feature has learned.

Provide a Detailed Rationale: Elaborate on your hypothesis. Explain why you believe the common theme is what the feature represents. Crucially, you must reference specific words or phrases from the provided prompts to justify your reasoning and show how they collectively point to your conclusion.

**Please Structure Your Response Exactly as Follows**:
Feature Name: A short, descriptive name for the feature, *e.g.,* "Kinase Hinge–Binding Motif" or "High Lipophilicity and Low Solubility."

Summary: A single sentence summarizing the feature's function, *e.g.,* "This feature represents the concept of a molecule designed as an ATP–competitive kinase inhibitor that forms key hydrogen bonds with the hinge region."

Detailed Rationale: A detailed paragraph explaining the common patterns observed in the text prompts. For example: "The prompts consistently point towards kinase inhibition. Prompt 1 explicitly mentions 'a potent inhibitor of EGFR', and prompt 5 names 'a tyrosine kinase'. The mechanism is specified in prompt 2 ('binds to the hinge region') and prompt 3 ('ATP–competitive inhibition'). Furthermore, the prompts allude to the necessary structural components for this function, such as 'a pyrimidine core' (prompt 1) and 'a hydrogen bond donor–acceptor pattern' (prompt 4), which are classic elements of hinge–binders. The convergence of these functional and structural descriptions strongly suggests the feature has learned to encode the specific concept of a kinase hinge–binder."

---

We also provide the prompt used to interpret features from the fitted linear model with the LLM (Gemini 2.5 Pro (Comanici et al., 2025) is also used for this task), as shown below.

---

### Prompt Used for LLM-Based Interpretations

**Objective**:
To interpret a specific feature from a text-to-molecule generation model by analyzing the text prompts that activate it most strongly, with a focus on water solubility characteristics.

**Context**:
I am analyzing the internal workings of a text-to-molecule generation model. This model takes a natural language text prompt and generates a corresponding molecular structure. I have identified a specific internal feature (*e.g.,* a neuron or a latent vector dimension) and have found the top 100 text prompts that cause the highest activation for this feature. My goal is to understand what chemical concept this feature has learned to encode, particularly as it relates to water solubility and aqueous behavior.

**Your role**:
Your role is to act as an expert in medicinal chemistry and cheminformatics with specialized knowledge in molecular solubility. Please analyze the semantic content of these text prompts to hypothesize the precise solubility-related property, structural motif affecting water solubility, or hydrophilic/hydrophobic concept that this feature represents.

**Input Data**:
Below are the top 100 activating text prompts for this feature, ranked from highest to lowest activation.

**Your Task & Required Output Format**:
1. Analyze Semantic Commonalities: Carefully read all prompts. Identify the common keywords, phrases, and underlying chemical concepts being described. Look for patterns related to:

- Water solubility descriptors (*e.g.,* "water-soluble", "aqueous", "hydrophilic", "hydrophobic")

- Functional groups affecting solubility (*e.g.,* "hydroxyl groups", "charged residues", "polar substituents")

- Physicochemical properties related to solubility (*e.g.,* "LogP", "polar surface area", "hydrogen bonding")

- Formulation or delivery aspects (*e.g.,* "oral bioavailability", "aqueous formulation", "membrane permeability")

2. Hypothesize Feature Identity: Based on your analysis, propose a concise name and a clear, one-sentence description for the solubility-related concept this feature has learned.

3. Provide a Detailed Rationale: Elaborate on your hypothesis. Explain why you believe the common theme is what the feature represents. Crucially, you must reference specific words or phrases from the provided prompts to justify your reasoning and show how they collectively point to your conclusion.

**Please Structure Your Response Exactly as Follows**:
Feature Name: [A short, descriptive name for the feature, *e.g.,* "High Aqueous Solubility Enhancer" or "Hydrophilic Functional Group Pattern"]

Summary: [A single sentence summarizing the feature's function, *e.g.,* "This feature represents molecules with multiple polar functional groups that significantly enhance water solubility through hydrogen bonding networks."]

Detailed Rationale: [A detailed paragraph explaining the common patterns observed in the text prompts. For example: "The prompts consistently emphasize water solubility enhancement. Prompt 1 explicitly mentions 'highly water-soluble

---

compound', while prompt 3 specifies 'improved aqueous solubility'. The structural basis for this property is revealed through references to specific functional groups: prompt 2 mentions 'multiple hydroxyl groups', prompt 4 describes 'polar substituents on the aromatic ring', and prompt 5 notes 'ionizable groups at physiological pH'. Additionally, the functional consequences are highlighted with phrases like 'oral bioavailability' (prompt 1) and 'suitable for IV formulation' (prompt 3). The convergence of these structural features and functional outcomes strongly suggests this feature encodes the concept of hydrophilic molecular modifications that enhance water solubility."]

Note: We acknowledge that LLM-based interpretation is not guaranteed to be *fully faithful*, but it offers a useful qualitative view of how chemically meaningful concepts may be organized in latent space. More importantly, it suggests structured reasoning patterns corresponding to chemically meaningful concepts in LLMs' latent representations. As shown in Section 4.1, these common patterns across diverse latent features can be extracted from representations and controlled by our method.

*Table 17.* LLM-derived interpretations of latent features, summarizing commonalities and reporting each feature's category, concept, and detailed interpretation. Feature groups are not necessarily mutually exclusive, and a latent feature may contribute to multiple higher-level concepts.

| Feature ID | Category | Concept | Interpretation |
|---|---|---|---|
| 21, 2737, 10881, 13815 | Functional Group | Peroxide (R-O-O-R') bond | Recognizes the peroxide (R-O-O-R') bond, a feature implying reactivity or instability. |
| 1883, 4374, 10168, 13616, 14625 | Structural Class | Scaffold: Piperidine ring | Identifies the piperidine ring as a common saturated heterocyclic scaffold. |
| 6643, 6749, 7729, 9916, 14095 | Structural Class | Scaffold: Oxazole/Thiazole ring | Recognizes oxazole or thiazole rings as key five-membered aromatic scaffolds. |
| 4580, 12785 | Structural Class | Scaffold: Thiophene ring | Identifies the thiophene ring as a sulfur-containing, core aromatic scaffold. |
| 327, 5576, 11937, 12190, 12674 | Functional Group | Sulfone (R-S(=O)$_2$-R') | Recognizes the sulfone group (R-S(=O)$_2$-R'), a stable and often electron-withdrawing moiety. |
| 1478, 5676, 13132 | Stereochemistry | Center of symmetry/meso compounds | Understands the concept of meso compounds: molecules with chiral centers that are achiral overall. |
| 7017, 8234, 11031 | Functional Group | Epoxide (oxirane) ring | Recognizes the strained and highly reactive epoxide (oxirane) ring. |
| 2107, 5036, 10626, 14729 | Functional Group | N-oxide | Recognizes the N-oxide feature for tuning the electronics and solubility of heterocycles. |
| 2839, 2872, 13586 | Stereochemistry | Helical chirality | Models helical chirality, a stereochemical feature arising from a molecule's screw-shaped structure. |
| 14381, 14624 | Chemical Interaction | Hydrogen bond donor/acceptor arrays | Specifies a defined pattern of hydrogen bond donors/acceptors to guide intermolecular interactions. |
| 7, 77, 12777 | Topology | Branched vs linear chain isomerism | Differentiates between branched and linear topologies of an aliphatic chain. |
| 9182, 13412 | Stereochemistry | Chiral centers with defined (S) stereochemistry | Defines a specific (S) configuration at a chiral center to ensure precise stereochemistry. |
| 8123, 13412, 14112 | Physicochemical Properties | Enhanced solubility through polar group addition | Guides generation by introducing polar groups to enhance water solubility. |
| 1601, 8808, 11765 | Conformation | Acyclic conformation: gauche/trans preference | Understands the energetic preference for gauche vs. trans conformations in acyclic chains. |
| 10113, 14412 | Structural Class | Saturated carbocyclic systems | Generates saturated carbocyclic systems (*e.g.*, cyclohexane), focusing on sp$^3$-rich structures. |
| 6, 14, 15015 | Physicochemical Properties | Aromaticity | Recognizes and prioritizes aromatic structures with planar, conjugated systems. |
| 99, 4567, 13987 | Physicochemical Properties | Target: High lipophilicity | Guides generation toward highly lipophilic molecules, targeting a LogP > 4. |
| 1001, 7002, 12311 | Conformation | Molecular shape: Planar/flat geometry | Enforces a planar geometry on the molecule or a significant portion, common for conjugated systems. |
| 8113, 13987 | Reactivity | Redox-active moieties | Recognizes and incorporates moieties capable of undergoing redox reactions (*e.g.*, quinones, thiols). |
| 556, 9123, 14411 | Structural Class | Low chirality/achiral design | A design constraint that guides the generation of achiral or low-chirality molecules for simplified synthesis. |

# O. Generalizability to Out-of-Distribution Data

In this section, we evaluate the generalizability of the learned concepts. We test the original SAEs on both local and global control across two unseen datasets, PubChemSTM (Liu et al., 2023a) and MolTextNet (Zhu et al., 2025), which are out-of-distribution for the pretrained SAEs. Local control results on PubChemSTM and MolTextNet are reported in Table 18 and Table 19. Global solubility control is shown in Figure 41 and Figure 42, respectively. We select representative state-of-the-art baselines spanning GNN-based, diffusion-based, and autoregressive models. Across both datasets, SpaRE shows strong OOD performance and achieves better generation quality and controllability than baselines. Global controls also transfer well to unseen data, indicating that SAEs learn meaningful concepts that generalize across datasets.

*Table 18.* Local molecule generation on the PubChemSTM dataset (Liu et al., 2023a). Quality and controllability are reported as percentages, while synthesizability and efficiency are reported as numerical values. **Best** and second-best results are indicated in bold and underlined, respectively.

| MODEL | VALID | UNIQUENESS | NOVELTY | ATOM STA | COMPLETENESS | SUCCESS RATE | SA SCORE |
|---|---|---|---|---|---|---|---|
| MolEvol | 92.96 | 79.61 | 84.82 | 81.04 | 97.89 | 31.53 | 4.79 |
| LDMol | 95.02 | 82.27 | **91.14** | **94.96** | 98.62 | 21.62 | **3.91** |
| TGM-DLM | 97.30 | **82.78** | 84.26 | 93.52 | 96.39 | 24.63 | 4.19 |
| RetMol | 93.58 | 78.03 | 73.84 | 82.75 | 98.23 | 28.34 | 4.92 |
| Llamole | 94.17 | 78.59 | 75.71 | 89.25 | 94.71 | 19.49 | 4.39 |
| SpaRE (**Ours**) | **100.00** | 78.87 | 90.16 | 89.85 | **99.21** | **96.77** | 3.98 |

*Table 19.* Local molecule generation on the MolTextNet dataset (Zhu et al., 2025). Quality and controllability are reported as percentages, while synthesizability and efficiency are reported as numerical values. **Best** and second-best results are indicated in bold and underlined, respectively.

| MODEL | VALID | UNIQUENESS | NOVELTY | ATOM STA | COMPLETENESS | SUCCESS RATE | SA SCORE |
|---|---|---|---|---|---|---|---|
| MolEvol | 91.19 | 72.48 | 93.62 | 89.59 | 92.92 | 42.57 | 4.78 |
| LDMol | 96.48 | 70.48 | 88.52 | 86.38 | 99.64 | 33.49 | **3.79** |
| TGM-DLM | 95.15 | 75.91 | 94.92 | 84.92 | 97.44 | 36.84 | 4.08 |
| RetMol | 96.31 | 73.65 | 82.42 | 87.78 | 94.50 | 22.30 | 4.22 |
| Llamole | 96.87 | 69.70 | 90.76 | 91.13 | 91.58 | 43.76 | 4.49 |
| SpaRE (**Ours**) | **100.00** | **77.25** | **96.70** | **93.98** | **99.85** | 91.43 | 3.85 |

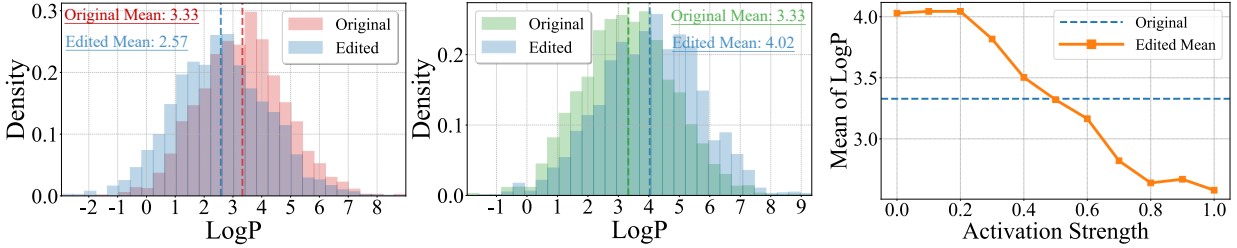

*Figure 41.* Distribution of molecules generated under solubility control on the PubChemSTM dataset (Liu et al., 2023a): (**Left**) amplification, (**Middle**) suppression, and (**Right**) controllable tuning by varying activation strength.

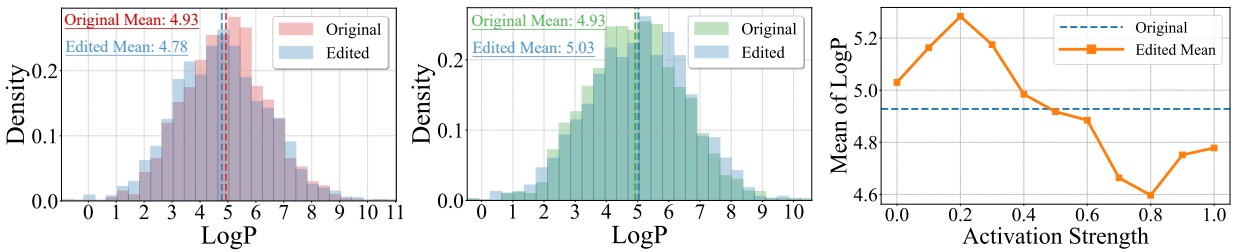

*Figure 42.* Distribution of molecules generated under solubility control on the MolTextNet dataset (Zhu et al., 2025): (**Left**) amplification, (**Middle**) suppression, and (**Right**) controllable tuning by varying activation strength.

## P. Additional Results on Water Solubility Prediction

To further examine whether the selected sparse features are meaningfully related to the controlled property, we conduct additional experiments on water solubility prediction. Specifically, we use randomly selected features for this task to verify that our linear probe identifies interpretable and chemically meaningful features. Results are reported in Table 20, Table 21, and Table 22, with the best results in **bold**. We find that sparse features enable more precise control than dense ones, yielding higher success rates and smoother global control. They also outperform random baselines, and the leave-one-out ablation suggests that a small subset of sparse features carries much of the predictive signal.

*Table 20.* Solubility prediction using ten randomly selected latent features, matched to the dimensionality of the linear probe in Figure 8).

| METHOD | ACCURACY (%, ↑) | AUC (%, ↑) |
|---|---|---|
| Random 1 | 53 | 47 |
| Random 2 | 54 | 48 |
| Random 3 | 50 | 52 |
| Random 4 | 52 | 49 |
| Random 5 | 48 | 55 |
| **Ours** | **88** | **92** |

*Table 21.* Ablation study on solubility prediction features via leave-one-out removal.

| SETTING | # REMOVED | ACCURACY (%, ↑) | AUC (%, ↑) |
|---|---|---|---|
| *w/o* x1 | 1 | 85 | 88 |
| *w/o* x1-x2 | 2 | 78 | 83 |
| *w/o* x1-x3 | 3 | 76 | 79 |
| *w/o* x1-x4 | 4 | 72 | 75 |
| *w/o* x1-x5 | 5 | 70 | 72 |
| *w/o* x1-x6 | 6 | 67 | 64 |
| *w/o* x1-x7 | 7 | 63 | 60 |
| *w/o* x1-x8 | 8 | 57 | 55 |
| *w/o* x1-x9 | 9 | 54 | 52 |
| *w/o* x1-x10 | 10 | 50 | 48 |
| **Ours** | - | **88** | **92** |

*Table 22.* Random directions matched in norm for increasing solubility (*i.e.,* measured by LogP) and the number of rings. $\Delta$ indicates the change in value.

| DIRECTION | $\Delta$LOGP (↓) | $\Delta$RING COUNT (↑) | LOGP SUCCESS RATE (%, ↑) | RING SUCCESS RATE (%, ↑) |
|---|---|---|---|---|
| Random 1 | - 0.13 | + 0.03 | 44.81 | 47.16 |
| Random 2 | + 0.10 | - 0.05 | 38.97 | 47.70 |
| Random 3 | - 0.02 | - 0.19 | 39.14 | 39.97 |
| Random 4 | - 0.07 | - 0.18 | 47.29 | 42.54 |
| Random 5 | + 0.07 | - 0.20 | 44.01 | 41.56 |
| **Ours** | **- 0.99** | **+ 1.12** | **92.43** | **91.60** |

## Q. Limitations and Future Work

- The efficacy of SpaRE is fundamentally tied to the capability of the underlying LLM. If the base model has not learned a robust concept, the SAE cannot extract it, and imperfect disentanglement may cause unintended edits to other molecular properties. Future work could involve applying SpaRE to more powerful foundation models and exploring advanced autoencoder designs to achieve more precise control.

- The current global control scheme relies on curated positive and negative exemplar sets, which can be a bottleneck for properties that are difficult to define with binary examples. Future research could focus on automatically building these examples, potentially by training a model to map property descriptions directly to contrastive sets.

- The scope of this work is limited to the field of molecular science, focusing on the study of molecular structures, properties, and interactions. While the principles and techniques discussed may hold relevance for other scientific disciplines, their exploration and validation are reserved for future studies.

