# OpenReview forum: "Controllable Molecule Generation via Sparse Representation Editing: An Interpretability-Driven Perspective"
_ICML.cc/2026/Conference — ICML 2026 regular_

### Official Review · Reviewer_KT48 · 2026-03-06

**Soundness:** 2
**Presentation:** 2
**Significance:** 3
**Originality:** 1
**Overall Recommendation:** 4
**Confidence:** 4

**Summary:**

The paper proposes SpaRE, which employs a sparse autoencoder on hidden activations of a pretrained molecular LLM to obtain disentangled sparse features aligned with certain useful chemical concepts. Then, by steering these sparse feature activations during inference, the method enables control to modify local substructures or global molecular properties from a given starting molecule.
It is shown that many of the emerged sparse properties can be interpreted with actual chemical properties. Experiments for site-specific editing, property steering, and multi-objective molecular optimization show consistent and efficient control accuracy with maintained validity. The authors conclude that sparse latent representations provide an interpretable and effective interface for controllable molecule design.

**Compliance With Llm Reviewing Policy:**

Affirmed.

**Key Questions For Authors:**

- When one performs global property control(Scheme 2), if all sparse features are controlled by the direction of ($\bar{h}^+-\bar{h}^-$), isn't it effectively equivalent to modifying dense feature in the first place? To justify the necessity of sparse features, the authors should compare the similar strategy into dense features without SAE(e.g., obtaining $\bar{h}^+$ and $\bar{h}^-$ with dense features and feed their difference into $z$) for the experiments on global property control(e.g., Figure 3).

**Limitations:**

yes

**Strengths And Weaknesses:**

**Strength**

Presentation:
- The authors have conducted extensive experiments across diverse scenarios(local/global property steering, multi-step editing route planning, and constrained multi-objective drug design), consistently demonstrating that steering sparse features can reliably modify the desired aspects of a molecule while preserving unrelated structural components.

Significance:
- Since state-of-the-art molecule generative models yet show suboptimal comprehension and alignment ability for complex chemical conditions, presenting practical molecule modification methods with interpretable control can expand these molecular generative models' real-world applicability.
---

**Weakness**

Originality & Soundness:
- The motivation, the methodology of using a sparse autoencoder, and the analyses of the interpretability of its features in this paper are mostly taken from Anthropic's research on monosemanticity of LLMs[1][2] (yet it's not discussed as a previous work of LLM sparse feature extraction, or even cited).

  While these approaches have been empirically successful, the emergence of sparse features that are interpretable and allow easy control of the model’s output is still not always guaranteed.

Presentation:
- The main benchmark results against the baselines(e.g., Table 1–2) seem to be an unfair comparison; since most baselines are not molecule editing models, they must generate molecules from scratch using only given text prompts, when SpaRE only needs to edit an already given molecule.

  The authors should rather include results on applying SpaRE to other different chemical LLMs beyond MolXPT, and demonstrate its consistent improvement in controlability.

- Regarding the global property control plot in Figure 3 and others, the authors should present a scatter plot of the input vs. output property for each molecule input to exclude the possibility of individual failure cases.
---
[1] Bricken, et al., "Towards Monosemanticity: Decomposing Language Models With Dictionary Learning", Transformer Circuits Thread, 2023.

[2] Templeton, et al., "Scaling Monosemanticity: Extracting Interpretable Features from Claude 3 Sonnet", Transformer Circuits Thread, 2024.

---

> ### Author Rebuttal · Authors · 2026-03-30
>
> We sincerely thank the reviewer for the careful assessment and constructive suggestions. Below, we address each concern point by point:
>
> **W1: Originality & Soundness.**
> We apologize for the omission and will properly discuss these works in the revision.
> - While inspired by prior work, we aim to shift interpretability analysis into actionable control, specifically addressing the lack of fine-grained controllability in molecular design. By disentangling latent representations, we first identify and construct molecular concepts that guide generation. We then develop two domain-specific control schemes: local control via hook retrieval and global control via contrastive guidance, tailored to the distinct nature of molecular substructures (single-token) and properties (multi-token), respectively. We transform interpretability into inference-time guidance for highly fine-grained controllable molecule generation under complex constraints. Moreover, by integrating SpaRE with MCTS, we establish an effective framework for various real-world challenges (e.g., oral drug design and synthesis planning), unlocking its scientific utility.
> - We acknowledge that the guarantees are not universal. However, empirical validation across diverse real-world tasks (e.g., drug design, bioisosteric editing) shows its practical applicability and value. Establishing theoretical guarantees remains a valuable direction for future work.
>
> **W2: Comparison of baselines and generalizability.**
> - We thank the reviewer for this important concern. Local control targets site-specific generation, requiring each method to *de novo* generate molecules that satisfy both text instructions and precise atom/group constraints. To the best of our knowledge, no existing method directly supports such highly user-specified, site-specific control. SpaRE is thus designed to address this gap. To ensure fairness, we benchmark against latest representative baselines using identical inputs, protocols, and constraint definitions, and will clarify this setup in the revision.
> - We appreciate the suggestion to evaluate generalizability. Applying SpaRE to diverse LLMs yields consistent performance in both local and global control, confirming its fine-grained controllability. Results are shown below.
>
> `Local control`
> | Model | Valid | Unique | Novel | Atom Sta | Complete | Success Rate | SA Score |
> |-|:-:|:-:|:-:|:-:|:-:|:-:|:-:|
> | BioT5 | 100.00 | 82.17 | 90.32 | 96.00 | 99.74 | 98.01 | 4.09 |
> | 3D-MoLM | 99.41 | 81.04 | 87.06 | 93.49 | 98.88 | 96.48 | 3.85 |
> | MolT5-Large | 99.88 | 80.76 | 91.22 | 95.97 | 99.71 | 97.74 | 3.67 |
> | Ours | 100.00 | 81.60 | 92.10 | 97.24 | 99.66 | 98.92 | 3.78 |
>
> `Global control of solubility (original mean: 2.57)`
> | Activation | Ours | BioT5 | 3D-MoLM | MolT5 |
> |:-:|:-:|:-:|:-:|:-:|
> | 0.0 | 4.55 | 4.46 | 4.37 | 4.34 |
> | 0.2 | 4.30 | 4.21 | 4.15 | 4.25 |
> | 0.4 | 3.82 | 3.60 | 3.64 | 3.71 |
> | 0.6 | 2.17 | 2.23 | 2.25 | 2.28 |
> | 0.8 | 0.77 | 1.03 | 0.99 | 1.05 |
> | 1.0 | 0.55 | 0.71 | 0.82 | 0.68 |
>
> `Global control of ring count (original mean: 2.26)`
> | Activation | Ours | BioT5 | 3D-MoLM | MolT5 |
> |:-:|:-:|:-:|:-:|:-:|
> | 0.0 | 1.37 | 1.47 | 1.50 | 1.48 |
> | 0.2 | 1.47 | 1.54 | 1.60 | 1.56 |
> | 0.4 | 1.69 | 1.77 | 1.80 | 1.76 |
> | 0.6 | 2.52 | 2.46 | 2.41 | 2.45 |
> | 0.8 | 3.19 | 3.07 | 2.98 | 3.04 |
> | 1.0 | 3.32 | 3.18 | 3.07 | 3.15 |
>
> **W3: Scatterplot in Figure 3.**
> We appreciate this suggestion. We generated a scatterplot to investigate failure cases, which confirms SpaRE's effectiveness: edited molecules consistently show increased solubility (points below $y=x$) and higher ring counts (points above $y=x$). The figure demonstrates robust global control with minimal failures. We will include this figure in our revision.
>
> **Q1: Sparse vs. dense features.**
> We appreciate this insightful comment. We leverage semantically disentangled concepts for interpretability. We consider dense features to be inherently polysemantic and entangled, posing risks of spurious correlations and inaccurate control. Yet, we fully agree on the need for empirical validation. Our comparison with a dense-feature baseline reveals inferior controllability in both local and global tasks, where identical activation strengths often fail to produce comparable effects. We attribute this to noisy, entangled dimensions that interfere with target properties, degrading performance compared to cleaner sparse representations.
>
> `Local control`
> | Method | Success Rate |
> |-|:-:|
> | Dense Features | 93.76 |
> | Ours | 98.92  |
>
> `Global control of solubility`
> | Activation | Ours (Mean LogP) | Dense (Mean LogP) |
> |:-:|:-:|:-:|
> | 0.0 | 3.53 | 3.31 |
> | 0.2 | 3.10 | 2.88 |
> | 0.4 | 2.75 | 2.63 |
> | 0.6 | 2.53 | 2.57 |
> | 0.8 | 2.30 | 2.42 |
> | 1.0 | 1.54 | 1.87 |
>
> `Global control of ring count`
> | Activation | Ours (Mean Rings) | Dense (Mean Rings) |
> |:-:|:-:|:-:|
> | 0.0 | 1.50 | 1.72 |
> | 0.2 | 1.72 | 1.94 |
> | 0.4 | 1.95 | 2.07 |
> | 0.6 | 2.26 | 2.18 |
> | 0.8 | 2.55 | 2.33 |
> | 1.0 | 3.10 | 2.87 |

---

> > ### Author Rebuttal · Reviewer_KT48 · 2026-04-01
> >
> > I thank the authors for their responses to address my concerns. I confirm that I have read the author's response and will accordingly update my review if necessary.

---

> > > ### Author Response · Authors · 2026-04-01
> > >
> > > We sincerely appreciate your thoughtful feedback and helpful suggestions. We are glad to learn that our rebuttal has addressed your concerns. Please do not hesitate to reach out if further clarification is needed. Should you find our responses satisfactory, we would be grateful if you would consider updating your score accordingly to support this work.

---

### Official Review · Reviewer_F2fx · 2026-03-09

**Soundness:** 3
**Presentation:** 3
**Significance:** 3
**Originality:** 3
**Overall Recommendation:** 5
**Confidence:** 3

**Summary:**

This paper proposes a framework for controllable molecule generation. The method trains sparse autoencoders on the hidden states of language models to extract features, enabling local editing of target fragments and adjustment of global properties during the inference stage. The authors evaluate the method on several downstream tasks, including site specific molecule generation, bioisosteric editing, and multi objective drug optimization under complex constraints.

**Compliance With Llm Reviewing Policy:**

Affirmed.

**Final Justification:**

The authors’ response has fully addressed the issues I raised in my initial review. I have raised my initial score.

**Key Questions For Authors:**

see weaknesses

**Limitations:**

yes

**Strengths And Weaknesses:**

Strengths:
1. Investigating controllable molecule generation addresses a highly meaningful problem in the field.
2. The designed experimental tasks are rich and possess strong practical application value.
3. The overall writing and organization of the paper are clear and easy to follow.
4. Attempting to use large language models to interpret neuron activations represents an interesting exploration.

Weaknesses:
1. It is recommended to evaluate the latest large language models, such as Gemini 3 Pro, and discuss whether they can already perform such tasks well. This will help further demonstrate the necessity of developing a specialized model.
2. The initial molecules before optimization in Section 4.3 are not clearly stated.
3. For the complex constraint optimization task in oral drug design, please calculate and provide the success rate of each method in satisfying all properties simultaneously. A quantitative success rate table can more intuitively demonstrate the advantages of your method.
4. It is advised to add error bars to Figure 6 to more rigorously evaluate the stability and statistical significance of the model performances.

---

> ### Author Rebuttal · Authors · 2026-03-30
>
> We sincerely thank the reviewer for the careful assessment and constructive suggestions. Below, we address each concern point by point:
>
> **W1: Evaluations of latest LLMs.**
> We appreciate this valuable suggestion. As recommended, we evaluated three state-of-the-art LLMs (GPT-5.4, Claude Opus 4.6, and Gemini 3.0 Pro) on controllable molecule generation. Our results demonstrate that SpaRE outperforms these models in both generation quality (validity, atom stability, and novelty) and controllability, achieving a significantly higher control success rate. Notably, SpaRE avoids the high reasoning latency and computational overhead inherent to advanced LLMs, delivering superior efficiency. This underscores the necessity of task-specific models over general-purpose LLMs for fine-grained controllable molecule generation. Detailed results are presented below and will be included in the revised manuscript.
>
> `Evaluation of latest general-purpose LLMs`
> | Model | Validity | Uniqueness | Novelty | Atom Sta | Completeness | Success Rate | SA Score | Time |
> |-------|:-----:|:----------:|:-------:|:--------:|:------------:|:------------:|:--------:|:----:|
> | GPT-5.4 | 88.60 | 80.83 | 83.79 | 92.11 | 95.41 | 56.98 | 3.97 | 45525.25 |
> | Claude Opus 4.6 | 91.02 | 79.60 | 84.77 | 89.74 | 95.46 | 66.82 | 4.02 | 5688.72 |
> | Gemini 3.0 Pro | 92.05 | 77.53 | 84.07 | 91.52 | 96.91 | 57.27 | 3.86 | 21125.61 |
> | Ours | 100.00 | 81.60 | 92.10 | 97.24 | 99.66 | 98.92 | 3.78 | 12.19 |
>
>
> **W2: Details about the dataset.**
> We apologize for the ambiguity regarding the data. The initial molecules used in Section 4.3 were randomly sampled from the ChEBI-20 dataset. Specifically, we selected 5,000 molecules that initially violated at least three of the specified constraints. We then optimized these molecules to satisfy all constraints required for multi-target drug discovery and oral drug design.
>
> **W3: Quantitative success rate table.**
> This is an excellent suggestion that strengthens our empirical analysis. We have computed the success rates (expressed as percentages) for each method. As shown below, SpaRE outperforms all baselines, achieving a 100% control success rate in multi-constraint molecule optimization. Moreover, these complex drug optimization objectives are achieved in only eight iterations, demonstrating our method's efficiency. The detailed results are summarized below:
>
> `Table of quantitative success rate`
> | Method | LogP ([1,3]) | MW (<500) | TPSA (≤140) | HBD (≤5) | HBA (≤10) | Arom. Rings (≤3) | SA (<5) | QED (>0.7) | All |
> |--------|:------------:|:---------:|:-----------:|:--------:|:---------:|:---------------:|:-------:|:----------:|:---:|
> | Original | 57.2 | 74.8 | 75.0 | 63.9 | 68.1 | 60.5 | 60.6 | 32.8 | 1.9 |
> | MolEvol | 65.4 | 84.1 | 88.4 | 91.3 | 92.1 | 79.0 | 62.1 | 22.6 | 1.6 |
> | Llamole | 88.3 | 98.3 | 96.9 | 91.9 | 91.5 | 77.8 | 78.8 | 56.8 | 21.7 |
> | Atomas | 90.3 | 97.6 | 95.8 | 96.3 | 96.0 | 99.1 | 91.5 | 98.4 | 67.4 |
> | RetMol | 95.8 | 98.9 | 100.0 | 89.6 | 93.7 | 93.9 | 96.2 | 0.5 | 0.2 |
> | TGM-DLM | 100.0 | 96.5 | 94.2 | 97.1 | 96.6 | 94.8 | 95.4 | 97.9 | 78.5 |
> | Ours | 100.0 | 100.0 | 100.0 | 100.0 | 100.0 | 100.0 | 100.0 | 100.0 | 100.0 |
>
> **W4: Adding error bars to Figure 6.**
> Thank you for this thoughtful comment. As detailed in our experimental setup (Appendix D), each experiment was repeated ten times with different random seeds to ensure statistical robustness. Following your suggestion, we have computed the standard deviation for each method and will update Figure 6 to include error bars. Additionally, to ensure consistency across the paper, we will also add error bars to other relevant figures in the revised manuscript. The computed mean and standard deviation for Figure 6 are shown below:
>
> `Mean and standard deviation`
> | Model | GSK3$\beta$ | JNK3 | QED | SA Score | LogP |
> |--------|-------|------|-----|----------|------|
> | TGM-DLM | 0.65 ± 0.05 | 0.63 ± 0.03 | 0.77 ± 0.04 | 2.50 ± 0.14 | 2.94 ± 0.15 |
> | MolEvol | 0.55 ± 0.03 | 0.57 ± 0.03 | 0.65 ± 0.04 | 3.55 ± 0.29 | 2.52 ± 0.17 |
> | RetMol | 0.66 ± 0.03 | 0.78 ± 0.04 | 0.79 ± 0.07 | 2.95 ± 0.18 | 3.16 ± 0.11 |
> | Atomas | 0.65 ± 0.05 | 0.72 ± 0.03 | 0.77 ± 0.06 | 2.95 ± 0.20 | 2.84 ± 0.15 |
> | Llamole | 0.63 ± 0.04 | 0.66 ± 0.05 | 0.76 ± 0.05 | 2.94 ± 0.21 | 2.52 ± 0.11 |
> | Ours | 0.64 ± 0.03 | 0.67 ± 0.04 | 0.78 ± 0.04 | 2.24 ± 0.15 | 3.24 ± 0.25 |

---

> > ### Author Rebuttal · Reviewer_F2fx · 2026-04-04
> >
> > Thank the authors for their responses. The authors’ response has fully addressed the issues I raised in my initial review. I will raise my score accordingly.

---

> > > ### Author Response · Authors · 2026-04-04
> > >
> > > We sincerely thank the reviewer for their constructive feedback and encouragement. We are delighted that our responses have fully addressed your concerns, and we truly appreciate your support in helping us improve our work.

---

### Official Review · Reviewer_HRKP · 2026-03-11

**Soundness:** 4
**Presentation:** 3
**Significance:** 3
**Originality:** 3
**Overall Recommendation:** 4
**Confidence:** 3

**Summary:**

This paper proposes SpaRE (Sparse Representation Editing), a lightweight method for fine-grained controllable molecular generation during inference. By editing interpretable features learned via SAE in the model’s representation space, SpaRE enables precise control over both local substructure insertion and global property modulation without modifying model parameters.

**Compliance With Llm Reviewing Policy:**

Affirmed.

**Key Questions For Authors:**

I am willing to increase the score if the author can clearly explain the following questions.

(1) How is the generalization ability of SAE on other models, such as when changing the parameter scale or applying it to other models?

(2) Since molecular properties often depend on 3D geometry, could the authors comment on whether the learned feature directions remain valid under conformational variation?

(3)Could the authors clarify what methodological novelty their approach provides beyond applying SAE to a different molecular generation model?

**Limitations:**

yes

**Strengths And Weaknesses:**

Strengths：

(1) Fine-grained controllable molecular generation with an LLM and an sparse autoencoders (SAE), more accurate molecular design than existing methods.

(2) The method is validated on various real-world tasks, demonstrating its feasibility and effectiveness for practical applications.

(3) The use of SAE to decompose LLM representations into chemically semantic features helps interpret the properties of generated molecules and improves the transparency of the approach.

Weaknesses：

(1) The authors claim their proposed method is lightweight and efficient, but they have not provided a clear complexity analysis, and I have concerns about its performance in practical applications.

(2) I have concerns about the novelty of this work because the proposed method is based on SAE design, which has recently been applied to interpretability analysis of the protein language model [1]. Could the authors elaborate on whether the performance improvement of the proposed method is not solely due to the use of SAE, but rather to a redesign of the problem of controlled molecular editing?

[1] Adams E, Bai L, Lee M, et al. From mechanistic interpretability to mechanistic biology: Training, evaluating, and interpreting sparse autoencoders on protein language models[J]. bioRxiv, 2025.

---

> ### Author Rebuttal · Authors · 2026-03-30
>
> We sincerely thank the reviewer for the careful assessment and constructive suggestions. Below, we address each concern point by point:
>
> **W1: Complexity analysis.**
> - We thank the reviewer for this important point. A key advantage is that SpaRE separates one-time offline concept construction from online (e.g., inference-time) control.
> - In the offline stage, sparse features are extracted via SAE through linear operations between the hidden dimension $d$ and sparse dimension $m$, incurring a cost of $O(d \cdot m)$.  Manipulating only the $s$ sparsely activated features costs $O(d \cdot s)$, where $s \ll m$. In our setting, $d = 1{,}024$ and $m = 40{,}960$ (40$×$ expansion), with training achieving over 99.7% sparsity. Local control typically identifies ~15 latents per atom/group, while global control identifies ~30 latents per property.
> - In the online stage, SpaRE does not re-run SAE at every generative step. Instead, concept vectors are precomputed in the offline stage and injected via a forward hook during online generation. As stated in Appendix C: "The edit is a precomputed vector in activation space injected via a forward hook, no SAE forward pass is needed." The additional inference-time overhead thus reduces to a lightweight hidden-state edit: $O(d)$ for local control at a single token step, or $O(T \cdot d)$ for global control over $T$ token length. For comparison, the dominant cost remains the base Transformer forward pass at around $O(L \cdot d^2)$ for $L$ layers (in our model, $L = 24$). SpaRE thus adds only marginal overhead relative to the underlying model, avoiding iterative sampling and auxiliary conditioning modules.
> - Therefore, once the concept vectors are built, SpaRE enables highly efficient local and global control. We will distinguish costs across these two stages in the revision.
>
> **W2 & Q3: Novelty explanation.**
> - We thank the reviewer for this insightful question. While Adams et al. primarily use SAEs for interpretability analysis, their notion of steering is limited to probing feature–family relationships and does not enable sequence generation or functional design. In this work, we shift SAEs from post-hoc interpretability analysis to actionable control, using latent representations to directly drive controllable molecule generation.
> - SpaRE is proposed to address the lack of granular control in molecular design, where SAE is used as a tool to identify concept-aligned latent features. Building on this, we develop two domain-specific schemes: local control via hook retrieval and global control via contrastive guidance, tailored to the distinct nature of molecular substructures (single-token) and properties (multi-token), respectively. Thus, we shift interpretability into inference-time guidance to address the granularity challenge in controllable molecule generation. Moreover, integrating SpaRE with MCTS extends its applicability to various real-world tasks (e.g., drug design and synthesis planning), showing that performance gains and practical value stem from the control-oriented framework rather than the use of SAEs alone.
>
> **Q1: Generalizability.**
> - We appreciate the suggestion. Experiments across 3 LLMs confirm SpaRE's consistent fine-grained controllability in both local and global tasks. We hypothesize that feature sparsity, concept alignment, and layer selectivity vary across backbones, offering a promising direction for future work. Due to space constraints, please refer to `W2` of `Reviewer KT48` for detailed results.
> - Appendix E details our hyperparameter sensitivity study (expansion factors and layers), while Appendix N reveals OOD generalizability. It confirms that SAEs learn transferable concepts across datasets.
>
> **Q2: Validity under conformational variation.**
> - This is a very important question. Though LLMs operate on text representations and do not explicitly encode 3D geometry, we empirically observe (Appendix J) that SpaRE can control features such as $sp^3$ carbons and rotatable bonds, which are closely related to 3D structure. We hypothesize this is because (1) training on GEOM-DRUGS and GEOM-QM9 implicitly exposes the model to conformational diversity, encoding compressed 3D information (geometry-associated representation), and (2) LLMs can capture underlying geometric correlations through extensive sequence-based learning.
> - However, since SpaRE operates in the text space rather than directly on 3D coordinates, we cannot claim invariance to conformational changes. Yet, most control targets in this work (e.g., solubility, substructures) are primarily determined by 2D topology and are less sensitive to geometry. Thus, SpaRE remains highly effective for 2D-dependent properties while offering potential utility for 3D-sensitive ones. We anticipate that integrating SpaRE with models explicitly capturing 3D geometry could further enhance robustness for learning 3D-specific concepts. A systematic study of 3D validity remains a valuable direction for future work.

---

> > ### Author Rebuttal · Reviewer_HRKP · 2026-04-03
> >
> > Thank you for the author's detailed response. The response has largely resolved my concerns, and I will consider updating my score.

---

> > > ### Author Response · Authors · 2026-04-03
> > >
> > > We sincerely appreciate your thoughtful feedback and constructive suggestions. We are glad that our rebuttal has addressed your concerns. Please feel free to reach out if any further clarification is needed. If you find our responses satisfactory, we would be deeply grateful if you could consider updatinf your score accordingly to support this work.

---

### Official Review · Reviewer_jaee · 2026-03-12

**Soundness:** 3
**Presentation:** 3
**Significance:** 3
**Originality:** 3
**Overall Recommendation:** 4
**Confidence:** 4

**Summary:**

This paper proposes SpaRE, an inference-time framework for controllable molecule generation using sparse representation editing inside a molecular LLM. The model is first adapted to Group SELFIES, after which layer-wise sparse autoencoders are trained to obtain sparse latent features that the authors interpret as chemically meaningful concepts. SpaRE then uses two control mechanisms: a local control scheme that edits a chosen token position to encourage a target atom or functional group, and a global control scheme that derives a contrastive steering direction for distributed molecular properties such as solubility, aromaticity, or ring statistics. The paper evaluates the method on site-specific editing, global property control, editing-route planning, and multi-objective drug design (combined with MCTS), and it also includes interpretability analyses based on linear probing and LLM-assisted feature descriptions.

**Compliance With Llm Reviewing Policy:**

Affirmed.

**Final Justification:**

The authors provided more results which resolved my concerns. I will keep my current positive score (4). The reason I cannot give a higher score is because of the relatively weaker theoretical justification (compared to the presented experiments).

**Key Questions For Authors:**

See Weaknesses.

**Limitations:**

yes

**Strengths And Weaknesses:**

**Strengths**

1. The paper addresses a compelling problem: achieving fine-grained, inference-time control over molecular generation without the need to retrain the underlying generator. The core contribution, which adapts sparse autoencoder-based representation editing from the mechanistic interpretability literature into the domain of controllable molecular generation, is genuinely novel. The decomposition into local and global control modes is well-motivated and intuitive. I also appreciate that the method remains lightweight at inference time and avoids introducing a heavyweight conditioning module.

2. The experimental coverage is commendably broad, spanning local editing, global property control, route planning, and multi-objective optimization, and further supplemented by ablation studies over layer selection and expansion factor. The reported improvements on local control tasks are substantial: on ChEBI-20, for instance, the method achieves a 98.92% success rate compared to 62.58% for the strongest baseline (Table 1), and on GEOM-DRUGS it reaches 96.77% against 57.84% (Table 7). The interpretability perspective is also a welcome addition. While I do not consider it fully substantiated at this stage (see below), the effort to ground steering vectors in chemically meaningful latent structure, rather than treating controllability as a purely black-box intervention, adds a valuable dimension to the work.

**Weaknesses**

1. My primary concern relates to potential evaluation leakage. As described in the implementation details, local concept vectors are derived from the dataset, and global concept vectors are explicitly constructed by averaging over the entire dataset, encompassing both positive and negative examples. Crucially, the paper never makes an unambiguous statement that these concept representations are computed exclusively from training-split examples, with no overlap with validation or test instances. This distinction matters significantly: if evaluation molecules contribute to the construction of the concept vectors, the controllability metrics could be meaningfully inflated. As it stands, this remains ambiguous and needs clarification.

2. The selected baselines span quite different model families, including graph-editing methods, diffusion models, and autoregressive text generators, and several of these were not originally designed for the specific site-specific replacement protocol employed here. Similarly, in the route-planning and multi-objective experiments, the proposed system is augmented with MCTS, yet it is unclear whether competing baselines are afforded an equivalent search procedure, an equivalent edit budget, comparable oracle access, or similar computational resources. Given the magnitude of the reported improvements, this absence of protocol-level detail is a serious concern. Without it, it is difficult to disentangle how much of the performance gain genuinely stems from sparse representation editing as opposed to favorable task construction, additional search capacity, or model-family mismatch.

3. Appendices A and B are framed as formal proofs, but in practice they rest on a set of strong assumptions that are themselves unproven: positive alignment between the steering vector and the relevant unembedding direction, approximate orthogonality with respect to unrelated token dimensions, and the persistence of property-aligned logit shifts across the generation trajectory. Under these conditions, the derivations follow, but the assumptions do the heavy lifting. As such, the results would be more accurately characterized as conditional guarantees rather than rigorous proofs.

4. The linear-probe result for solubility is suggestive, and the use of LLM-generated feature descriptions is an interesting methodological choice, but neither is sufficient to establish that the edited features are uniquely concept-aligned or causally faithful. Stronger validation would meaningfully strengthen these claims. For example, comparisons against dense hidden-state baselines, random-feature controls, random steering directions matched in norm, or targeted concept-ablation studies that quantify both specificity and collateral impact on unrelated molecular properties would all help substantiate the interpretability narrative more rigorously.

---

> ### Author Rebuttal · Authors · 2026-03-30
>
> We sincerely thank the reviewer for the careful assessment and constructive suggestions. Below, we address each concern point by point:
>
> **W1: Concept vector construction.**
> We agree that this point is critical. All local and  global concepts are constructed strictly from the training split, with no overlap with validation or test data. We will clarify this explicitly in the revision to remove any ambiguity.
>
> **W2: Baseline comparison.**
> - We thank the reviewer for this concern. We aim to benchmark against representative SOTA methods for controllable/constraint-aware molecule generation. To the best of our knowledge, no prior approach directly supports fine-grained, site-specific generation. SpaRE is thus designed to address this gap, enabling user-specified atom/group control at specific positions. To ensure fairness, we used identical inputs, metrics, and constraint definitions across all methods, and will explicitly state these protocols in the revision.
> - For route-planning and multi-objective experiments, we leverage MCTS to efficiently explore the edit space under complex constraints. It is a standard practice also used by our baselines like *Llamole* (A* search), *RetMol* (retrieval database), and *MolEvol* (evolutionary search). Crucially, fine-grained control stems from precise latent representation manipulation, while MCTS solely enhances search efficiency. We will detail search/edit budget and computational resources in the revision, confirming they are comparable across all methods.
>
> **W3: Theoretical claims.**
> We appreciate this important point. We acknowledge that the results in Appendices A and B represent conditional guarantees based on stated assumptions rather than formal proofs. We will revise the wording to clarify this distinction and explicitly discuss these assumptions in the revision.
>
> **W4: Empirical validation.**
> We thank the reviewers for this insightful suggestion. We agree that the current evidence supports concept alignment but does not yet establish full causal faithfulness. We conducted all the suggested experiments. Our findings reveal that sparse features enable more precise control than dense ones, yielding higher control success rates and smoother global control. They significantly outperform random baselines, while ablation studies showcase their causal importance. Results are given below.
>
> `Local control`
> | Method | Success Rate |
> |-|:-:|
> | Dense Feature | 93.76 |
> | Ours | 98.92  |
>
> `Global control of solubility`
> | Activation | Ours (Mean LogP) | Dense Feature (Mean LogP) |
> |:-:|:-:|:-:|
> | 0.0 | 3.53 | 3.31 |
> | 0.2 | 3.10 | 2.88 |
> | 0.4 | 2.75 | 2.63 |
> | 0.6 | 2.53 | 2.57 |
> | 0.8 | 2.30 | 2.42 |
> | 1.0 | 1.54 | 1.87 |
>
> `Global control of ring count`
> | Activation | Ours (Mean Rings) | Dense Feature (Mean Rings) |
> |:-:|:-:|:-:|
> | 0.0 | 1.50 | 1.72 |
> | 0.2 | 1.72 | 1.94 |
> | 0.4 | 1.95 | 2.07 |
> | 0.6 | 2.26 | 2.18 |
> | 0.8 | 2.55 | 2.33 |
> | 1.0 | 3.10 | 2.87 |
>
> `Randomly-selected ten latent features (same as our linear probe) to predict solubility`
> | Method | Accuracy (%, ↑) | AUC (%, ↑) |
> |-|-|-|
> | Random1 | 53 | 47 |
> | Random2 | 54 | 48 |
> | Random3 | 50 | 52 |
> | Random4 | 52 | 49 |
> | Random5 | 48 | 55 |
> | Ours | 88 | 92 |
>
> `Random direction matched in the same norm (increase solubility and rings)`
> | Direction | ΔLog (↓) | ΔRing Count (↑) | LogP Success Rate (%, ↑ ) | Ring Success Rate (%, ↑ ) |
> |-|:-:|:-:|:-:|:-:|
> | Random1 | -0.13 | +0.03 | 44.81 | 47.16 |
> | Random2 | +0.10 | -0.05 | 38.97 | 47.70 |
> | Random3 | -0.02 | -0.19 | 39.14 | 39.97 |
> | Random4 | -0.07 | -0.18 | 47.29 | 42.54 |
> | Random5 | +0.07 | -0.20 | 44.01 | 41.56 |
> | Ours | -0.99 | +1.12 | 92.43 | 91.60 |
>
> `Ablation study on solubility prediction features via leave-one-out removal`
> | Setting | # Removed | Accuracy (%, ↑) | AUC (%, ↑) |
> |-|:-:|:-:|:-:|
> | w/o x1 | 1 | 85 | 88 |
> | w/o x1-x2 | 2 | 78 | 83 |
> | w/o x1-x3 | 3 | 76 | 79 |
> | w/o x1-x4 | 4 | 72 | 75 |
> | w/o x1-x5 | 5 | 70 | 72 |
> | w/o x1-x6 | 6 | 67 | 64 |
> | w/o x1-x7 | 7 | 63 | 60 |
> | w/o x1-x8 | 8 | 57 | 55 |
> | w/o x1-x9 | 9 | 54 | 52 |
> | w/o x1-x10 | 10 | 50 | 48 |
> | Ours | 0 | 88 | 92 |
>
> `Random latent features for local control`
> | Method | Success Rate |
> |-|:-:|
> | Random1 | 29.84 |
> | Random2 | 30.56 |
> | Random3 | 28.93 |
> | Ours | 98.92 |
>
> `Random latent features for solubility control (original mean: 2.57)`
> | Activation | Ours | Random1 | Random2 | Random3 |
> |:-:|:-:|:-:|:-:|:-:|
> | 0.0 | 4.55 | 2.62 | 2.51 | 2.63 |
> | 0.2 | 3.76 | 2.61 | 2.58 | 2.55 |
> | 0.4 | 3.01 | 2.55 | 2.64 | 2.59 |
> | 0.6 | 2.17 | 2.50 | 2.56 | 2.62 |
> | 0.8 | 1.33 | 2.63 | 2.52 | 2.58 |
> | 1.0 | 0.59 | 2.56 | 2.62 | 2.50 |
>
> `Random latent features for ring count control (original mean: 2.26)`
> | Activation | Ours | Random1 | Random2 | Random3 |
> |:-:|:-:|:-:|:-:|:-:|
> | 0.0 | 1.37 | 2.30 | 2.21 | 2.28 |
> | 0.2 | 1.73 | 2.24 | 2.29 | 2.22 |
> | 0.4 | 2.11 | 2.27 | 2.23 | 2.30 |
> | 0.6 | 2.52 | 2.22 | 2.28 | 2.25 |
> | 0.8 | 2.92 | 2.29 | 2.22 | 2.27 |
> | 1.0 | 3.30 | 2.25 | 2.31 | 2.23 |

---

> > ### Author Rebuttal · Reviewer_jaee · 2026-04-01
> >
> > Thank you for the detailed rebuttal. This response is helpful and does address several of my original concerns. That said, I still have some reservation about the current submission as written. A number of key protocol details and several of the strongest supporting results now appear only in the rebuttal, rather than in the paper itself, especially regarding fairness/comparability in the search-based experiments. So while my confidence in the work has increased, I still feel the paper would benefit from revision before publication. I therefore remain slightly negative overall.

---

> > > ### Author Response · Authors · 2026-04-03
> > >
> > > We sincerely thank the reviewer for recognizing the value of our rebuttal and for the constructive feedback. We are glad that our clarification has addressed several concerns and increased your confidence in the work. Regarding W2, we commit to meticulously incorporating all detailed explanations and results from the rebuttal into the revised manuscript. Specifically, we will provide the following clarifications:
> > > - Regarding the core of our method (covering most experiments), **local control of molecular substructures (e.g., site-specific generation) and global control of molecular properties**, SpaRE **does not use MCTS at any stage**. Instead, the key to fine-grained controllability lies in identifying and constructing effective concept representations to control user-specified atoms/functional groups or overall molecular properties.
> > > - We integrate SpaRE with MCTS only when adapting it to **more efficiently address real-world tasks**, such as route planning, multi-objective optimization, and drug design, aiming to reduce the search space for greater efficiency.
> > > - We conducted additional experiments to evaluate these search-based methods across three aspects: generation quality, constraint satisfaction, and efficiency. All experiments were run under identical hardware and inference configurations. To ensure fair comparison within the same resource constraints, we applied a unified computational budget and stopping criteria. Furthermore, all methods accessed only the same property predictor or scoring oracle, without any additional priors, enhanced search budgets, or superior evaluation interfaces. The results are presented below.
> > >
> > > `Constraint values for molecules optimized for multi-target drug discovery`
> > > | Method   |  GSK3$\beta$ ($≥$ 0.5, ↑) |   JNK3 ($≥$ 0.5, ↑) |    QED ($>$ 0.6, ↑) | SA Score ($<$ 4, ↓) |
> > > | :------- | -----: | -----: | -----: | -------: |
> > > | TGM-DLM  | 0.6487 | 0.6252 | 0.7712 |   2.4997 |
> > > | MolEvol  | 0.5450 | 0.5719 | 0.6524 |   3.5513 |
> > > | RetMol   | 0.6615 | 0.7769 | 0.7852 |   2.9489 |
> > > | Atomas   | 0.6475 | 0.7178 | 0.7722 |   2.9453 |
> > > | Llamole  | 0.6334 | 0.6586 | 0.7591 |   2.9417 |
> > > | Ours | 0.7323 | 0.7805 | 0.8429 |   2.1485 |
> > >
> > > `Constraint satisfaction Rate (%, ↑) for multi-target drug discovery`
> > > | Method   | GSK3$\beta$ $≥$ 0.5 | JNK3 $≥$ 0.5 | QED $>$ 0.6 |  SA $<$ 4 | All |
> > > | :------- | --------: | -------: | ------: | ----: | ----: |
> > > | TGM-DLM  |     72.4 |    69.2 |   90.6 | 98.4 | 44.7 |
> > > | MolEvol  |     57.1 |    61.3 |   65.6 | 73.9 | 17.0 |
> > > | RetMol   |     74.1 |    86.6 |   92.3 | 93.3 | 55.3 |
> > > | Atomas   |     72.2 |    80.8 |   90.7 | 93.4 | 49.5 |
> > > | Llamole  |     70.3 |    73.7 |   89.0 | 93.5 | 43.1 |
> > > | Ours |   85.6   |    89.2 |   96.1 | 99.8 | 83.2
> > >
> > > `Generation quality (%) for molecules optimized for multi-target drug discovery`
> > > | Method           |  Success (↑) |  Validity (↑) |  Novelty (↑) | Diversity (↑) |
> > > | :--------------- | -------: | --------: | -------: | --------: |
> > > | MolEvol          |     90.1 |      93.5 |     82.4 |      67.3 |
> > > | RetMol           |     95.9 |      97.2 |     87.6 |      72.6 |
> > > | TGM-DLM          |     96.1 |      98.4 |     77.2 |      66.9 |
> > > | Atomas           |     94.1 |      97.7 |     85.2 |      70.9 |
> > > | Llamole          |     88.2 |      98.3 |     91.1 |      67.1 |
> > > | Ours | 99.8 | 100.0 | 95.1 |  74.4 |
> > >
> > > `Runtime for each method (seconds per molecule satisfying the specified constraints, ↓)`
> > > | Model   | Oral Drug Optimization | Sublingual Drug Optimization | Multi-Target Drug Discovery |
> > > | - | - | - | - |
> > > | TGM-DLM | 4.43    | 7.07    | 17.84   |
> > > | MolEvol | 2.40    | 3.61    | 8.98    |
> > > | RetMol  | 10.12   | 15.99   | 40.12   |
> > > | Llamole | 43.11   | 64.02   | 161.77  |
> > > | Atomas  | 1.92    | 2.83    | 7.01    |
> > > | Ours    | 2.18    | 3.50    | 6.67    |
> > >
> > > We sincerely thank the reviewer for their time and effort during the review process. Please do not hesitate to reach out if further clarification is needed.

---

### Decision · Program_Chairs · 2026-04-30

**Decision:**

Accept (regular)

**Comment:**

This paper adapts sparse autoencoders, a mechanistic interpretability approach for understanding LLM activations, for controllable generation of molecules using the learned feature space. This builds upon existing LLM interpretability and steering work, but the application to molecular generation is novel and valuable.

Multiple reviewers raised scores after the rebuttal, which largely addressed reviewer concerns, and all reviewers lean accept at this stage. The main caution at this point is that many of the experimental results and details only appear in the rebuttal, not the submission. Please be very thorough in updating the final version of the manuscript!